# FIND YOUR OPTIMAL ASSIGNMENTS ON-THE-FLY: A HOLISTIC FRAMEWORK FOR CLUSTERED FEDERATED LEARNING

## ABSTRACT

Federated Learning (FL) is an emerging distributed machine learning approach that preserves client privacy by storing data on edge devices. However, data heterogeneity among clients presents challenges in training models that perform well on all local distributions. Recent studies have proposed clustering as a solution to tackle client heterogeneity in FL by grouping clients with distribution shifts into different clusters. However, the diverse learning frameworks used in current clustered FL methods make it challenging to integrate various clustered FL methods, gather their benefits, and make further improvements.

To this end, this paper presents a comprehensive investigation into current clustered FL methods and proposes a four-tier framework, namely HCFL, to encompass and extend existing approaches. Based on the HCFL, we identify the remaining challenges associated with current clustering methods in each tier and propose an enhanced clustering method called HCFL$^+$ to address these challenges. Through extensive numerical evaluations, we showcase the effectiveness of our clustering framework and the improved components. Our code will be publicly available.

## 1 INTRODUCTION

Federated Learning (FL) is a privacy-focused distributed machine learning approach. In FL, the server shares the model with clients for local training, and the clients send parameter updates back to the server. The clients will not share their raw data with servers, ensuring privacy. However, the non-iid client data distribution leads to significant performance drops for FL algorithms (McMahan et al., 2016; Li et al., 2018; Karimireddy et al., 2020; 2019). To address data heterogeneity, traditional FL focuses on training a single global model that performs well across all local distributions (Li et al., 2021; 2018; Tang et al., 2022; Guo et al., 2023a). However, relying solely on a global model may not adequately handle the heterogeneous client distributions. As a remedy, clustered FL methods have been proposed to group clients into different clusters based on their local distributions. Numerous studies have demonstrated the superiority of clustered FL methods over single-model FL approaches (Long et al., 2023; Sattler et al., 2020b; Ghosh et al., 2020; Marfoq et al., 2021; Guo et al., 2023b).

**Diverse learning frameworks pose challenges on enhancing the clustered FL.** Despite the success of current clustered FL methods, the use of diverse learning frameworks poses challenges in integrating different algorithms, gathering their advantages, and achieving further improvements. For instance, FedEM (Marfoq et al., 2021) excels in addressing complex mixture distribution scenarios and performs admirably on challenging tasks. However, it necessitates a predefined number of clusters, constraining its practicality. In contrast, adaptive clustering techniques such as CFL (Sattler et al., 2020b) can autonomously determine the number of clusters. Nonetheless, CFL cannot be seamlessly integrated with soft clustering methods like FedEM, thereby limiting its effectiveness in handling complex mixture distribution tasks.

**Consolidating existing methods as a solution.**   To tackle these challenges, our aim is to establish a holistic learning framework for clustered FL methods, allowing us to seamlessly combine their advantages. Once accomplished, we can easily incorporate existing methods such as FedEM and CFL to develop an improved approach (see Table 1).

Therefore, we first revisit and summarize existing clustered FL methods. We then introduce HCFL, a holistic clustered FL algorithm framework with four tiers (as shown in Figure 1). These tiers address the primary tasks in clustering methods: (1) Cluster Learning and Assignment (tiers 1 and 2),

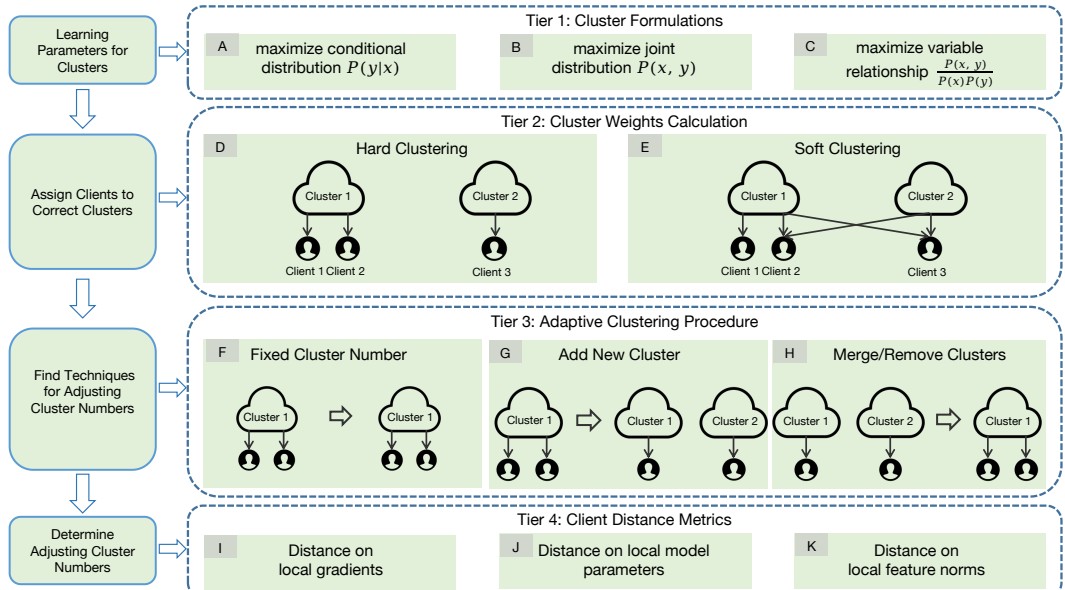

Figure 1: **Overview of the HCFL.** The HCFL encompasses the existing clustered FL algorithms through the design of four tiers, including *cluster formulations*, which maximize conditional distribution, joint distribution, or variable relationships; *cluster weights calculation*, including soft clustering and hard clustering; *adaptive clustering procedure*, including using a predefined number of clusters, automatically adding new clusters, or merge and remove existing clusters; *client distance metrics*, including using distance on clients' local gradients, clients' local model parameters, or clients' local feature norms. The four tiers collaborate to form a comprehensive clustered FL learning process, as shown in the left part of the figure. For instance, CFL can be described by the A, D, G, and J, while A, E, and F cover FedEM.

which assigns clients to optimal clusters and learns cluster-specific parameters; (2) Cluster Number Determination (tiers 3 and 4), which decides the number of clusters. The four tiers within HCFL constitute a comprehensive clustered FL learning process that incorporates existing methods and allows for flexible modifications in each tier (see Algorithm 1). It is evident that the enhanced algorithms exhibit significant performance improvements compared to the original methods, as demonstrated in Table 1.

In light of the HCFL, we have identified the remaining challenges within each tier that were previously overlooked by existing clustered FL, as illustrated in Figure 2 in Section 4.1. We then introduce HCFL$^+$, an enhanced algorithm designed to tackle these remaining challenges. Numerical results confirm that HCFL$^+$ effectively extends existing methods, achieving a superior balance between personalization and generalization while delivering strong performance.

We summarize the contribution of this paper as follows:

- We introduce HCFL, a holistic framework for clustered FL that encompass the existing methods. HCFL enables the integration of existing methods' benefits by adjusting techniques in each tier.
- We identify four remaining challenges in clustered FL algorithms within each tier of HCFL, and introduce an improved algorithm called HCFL$^+$ to address these challenges.
- Extensive experiments on different datasets (CIFAR10, CIFAR100, and Tiny-Imagenet) and various architectures (MobileNet-V2 and ResNet18) demonstrate the effectiveness of our framework and the improved components of HCFL$^+$.

## 2    RELATED WORKS

In the field of Federated Learning, FedAvg serves as the de-facto algorithm, employing local Stochastic Gradient Descent (local SGD) techniques (McMahan et al., 2016; Lin et al., 2020) to reduce communication costs and protect client privacy. However, FL faces significant challenges due to distribution shifts among clients, which can hinder the performance of FL algorithms (Li et al., 2018; Wang et al., 2020; Karimireddy et al., 2020; Jiang & Lin, 2023; Guo et al., 2021). Traditional FL methods primarily focus on improving the convergence speed of global models and incorporate bias reduction techniques (Tang et al., 2022; Guo et al., 2023a; Li et al., 2021; 2018). However, these single-model approaches are often inadequate for handling heterogeneous data distributions,

especially in cases involving concept shifts (Ke et al., 2022; Guo et al., 2023b; Jothimurugesan et al., 2023). To address these challenges, researchers have introduced clustered FL algorithms to enhance the performance of FL algorithms.

Clustered FL groups clients based on their local data distribution, addressing the distribution shift problem. Most methods employ hard clustering with a fixed number of clusters, grouping clients by measuring their similarities (Ghosh et al., 2020; Long et al., 2023; Wang et al., 2022b; Stallmann & Wilbik, 2022). However, hard clustering may not adequately capture complex relationships between local distributions, and soft clustering paradigms have been proposed to address this issue (Marfoq et al., 2021; Wu et al., 2023; Ruan & Joe-Wong, 2022; Guo et al., 2023b). In this paper, we propose a generalized formulation for clustered FL that encompasses current methods and improves them by addressing issues related to intra-client inconsistency and efficiency.

Another line of research focuses on automatically determining the number of clusters. Current methods utilize hierarchical clustering (Sattler et al., 2020b;a; Zhao et al., 2020; Briggs et al., 2020; Zeng et al., 2023; Duan et al., 2021a;b), which measures client dissimilarity using model parameters or local gradient distances. Some papers enhance these distance metrics by employing various techniques, such as eigenvectors (Yan et al., 2023) and local feature norms (Wei & Huang, 2023). FEDCOLLAB (Bao et al., 2023) quantifies client similarity through client discriminators. However, the requirement for discriminators between every client pair in FEDCOLLAB hinders scalability for cross-device scenarios with numerous clients. In this paper, we concentrate on cross-device settings, introducing a holistic adaptive clustering framework enabling cluster splitting and merging. We also present enhanced weight updating for soft clustering and finer distance metrics for various clustering principles. For further discussions on related works, please refer to Appendix C.

## 3  HCFL: REVISITING AND EXTENDING CLUSTERED FL METHODS

Current clustered FL methods typically employ diverse learning frameworks. As a result, existing methods often face challenges in gathering the advantages of different algorithms for potential enhancements. To address this issue, as shown in Figure 1, we introduce the HCFL, consisting of four tiers designed to tackle the primary tasks of clustering methods: (1) Cluster Learning and Assignment (tiers 1 and 2): Identify which clients should belong to the same clusters and learns parameters for each cluster. (2) Cluster Number Determinant (tiers 3 and 4): Decide the number of clusters. As a result, the four tiers of the HCFL form a comprehensive learning process (Algorithm 1), enabling flexible improvements and the integration of advantages from different algorithms (Table 1).

### 3.1  TIERS 1 & 2: THE CLUSTER FORMULATIONS AND CLUSTER WEIGHTS CALCULATION

We introduce the first two tiers: Cluster Formulations and Cluster Weights Calculation. Cluster Formulations defines the objective functions of the clustering methods, aiming to learn the underlying distributions of each cluster. Cluster Weights Calculation orthogonally helps find the suitable clusters for each client, whereas hard clustering assigns each client to one cluster, while soft clustering allows clients to contribute to multiple clusters. We propose the following optimization framework to encompass these two tiers.

**Optimization framework of clustered FL methods.** The clustered FL methods can be expressed as a dual-variable optimization problem that maximizes $\mathcal{L}(\boldsymbol{\Theta}, \boldsymbol{\Omega})$, with $K$ clusters and $M$ data sources represented as $\mathcal{D}_1, \cdots, \mathcal{D}_M$:

$$\mathcal{L}(\boldsymbol{\Theta}, \boldsymbol{\Omega}) = \frac{1}{N} \sum_{i=1}^{M} \sum_{j=1}^{N_i} \log \left( \sum_{k=1}^{K} \omega_{i;k} \mathcal{L}_k(\mathbf{x}_{i,j}, y_{ij}; \boldsymbol{\theta}_k) \right), \quad \text{s.t.} \quad \sum_{k=1}^{K} \omega_{i;k} = 1, \forall i, \quad (1)$$

where $N = \sum_{i=1}^{M} N_i$ and $N_i := |\mathcal{D}_i|$. The parameters to be optimized are clustering weights $\boldsymbol{\Omega} = [\omega_{1;1}, \cdots, \omega_{M,K}]$, and model parameters $\boldsymbol{\Theta} = [\boldsymbol{\theta}_1, \cdots, \boldsymbol{\theta}_K]$.

**Tier 1: Incorporate existing Cluster Formulations.** The existing methods employ clustering to address diverse tasks, which results in the proposal of various formulations. Our Algorithm 1 encompasses the existing clustering formulations by selecting $\mathcal{L}_k(\mathbf{x}_{i,j}, y_{ij}; \boldsymbol{\theta}_k)$ as follows:

- $\mathcal{P}_{\boldsymbol{\theta}_k}(y_{i,j}|\mathbf{x}_{i,j})$. Most existing methods (Marfoq et al., 2021; Ghosh et al., 2020; Long et al., 2023) can be recovered using this conditional distribution (likelihood functions).
- $\mathcal{P}_{\boldsymbol{\theta}_k}(\mathbf{x}_{i,j}, y_{i,j})$. FedGMM (Wu et al., 2022) uses this joint probability. [1]

---

[1] In FedGMM (Wu et al., 2023), $\boldsymbol{\theta}_k$ is split into $[\boldsymbol{\theta}_{k_1}, \boldsymbol{\nu}_{k_2}]$, and it uses $\mathcal{L}_k(\mathbf{x}_{i,j}, y_{i,j}; \boldsymbol{\theta}_k) = \mathcal{P}_{\boldsymbol{\theta}_{k_1}}(y_{i,j}|\mathbf{x}_{i,j})\mathcal{P}_{\boldsymbol{\nu}_{k_2}}(\mathbf{x}_{i,j})$ to model the joint probability.

---

**Algorithm 1** HCFL: holistic Algorithm Framework of clustered FL.

---

**Require:** Number of communication rounds $T$, initial number of clusters $K^0$, initial parameters $\phi^0$, and $\mathbf{\Theta}^0$.
**Ensure:** Number of clusters $K^T$, trained parameters $\phi^T$, and $\mathbf{\Theta}^T$.

1: **for** $t = 0, \cdots, T-1$ **do**
2:     Sample a subset of clients $\mathcal{S}^t$, and send $\mathbf{\Theta}^{t+1}$ to the clients.
3:     **for** Client $i$ in $\mathcal{S}^t$ **do**
4:        Local updates by solving (1).               ▷ Tiers 1 and 2
5:        Upload local gradients $\mathbf{g}_{i;k}^{t+1}$ to the server.
6:     $\boldsymbol{\theta}_k^{t+1} = \boldsymbol{\theta}_k^t - \eta_g \sum_{i \in \mathcal{S}^t} \mathbf{g}_{i;k}^{t+1}, \forall k$.
7:     Calculate distance matrix $\mathbf{D}^t$, and $\mathbf{D}_k^t$ for each cluster $k$.          ▷ Tier 4
8:     **if** Detect cluster $k_s$ need to be split **then**              ▷ Tier 3
9:        Split clients in cluster $k_s$ into two sub-clusters $\mathcal{S}_{s,1}$ and $\mathcal{S}_{s,2}$ based on $\mathbf{D}_{k_s}^t$ or $\mathbf{D}^t$.
10:        $\boldsymbol{\theta}_{k_s}^{t+1} = \boldsymbol{\theta}_k^t - \eta_g \sum_{i \in \mathcal{S}_{s,1}} \mathbf{g}_{i;k}^{t+1}$.
11:        $\boldsymbol{\theta}_{K^t+1}^{t+1} = \boldsymbol{\theta}_k^t - \eta_g \sum_{i \in \mathcal{S}_{s,2}} \mathbf{g}_{i;k}^{t+1}$.
12:        Update $\omega_{i;k}$ for corresponding clients.
13:     **if** Detect cluster $k_d$ need to be deleted **then**            ▷ Tier 3
14:        Delete cluster $k_d$.
15:        Update $\omega_{i;k}$ for corresponding clients.
16:     Update $K^{t+1}$ by the current number of clusters.

---

- $\frac{\mathcal{P}_{\boldsymbol{\theta}_k}(\mathbf{x}_{i,j}, y_{i,j})}{\mathcal{P}_{\boldsymbol{\theta}_k}(\mathbf{x}_{i,j})\mathcal{P}_{\boldsymbol{\theta}_k}(y_{i,j})}$. FedRC (Guo et al., 2023b) relies on correlations between variables $\mathbf{x}$ and $y$.

**Tier 2: Incorporate existing Cluster Weights Calculation.** Various methods employ distinct mechanisms for calculating clustering weights $\omega_{i;k}$. The choice of $\omega_{i;k}$, with either binary values $\omega_{i;k} \in \{0, 1\}$ or continuous values $\omega_{i;k} \in [0, 1]$, characterizes the dynamic clustering procedure.

- Hard clustering methods employ binary values $\omega_{i;k} \in 0, 1$. In these methods, $\omega_{i;k}$ is determined using heuristic techniques, such as parameter distance (Long et al., 2023; Zeng et al., 2023; Sattler et al., 2020b) or local loss function values (Ghosh et al., 2020).
- Soft clustering approaches permit $\omega_{i;k} \in [0, 1]$, determined by maximizing $\mathcal{L}(\mathbf{\Theta}, \mathbf{\Omega})$(Marfoq et al., 2021; Guo et al., 2023b; Wu et al., 2023), or by normalizing local loss values(Ruan & Joe-Wong, 2022). Soft clustering methods do not assume separated clients' local distributions and can thus handle complex scenarios, such as mixture distributions (Marfoq et al., 2021; Wu et al., 2023).

## 3.2 TIERS 3 & 4: THE ADAPTIVE CLUSTERING PROCEDURE AND DISTANCE METRICS

Tiers 3 and 4 illustrate the techniques for Cluster Number Determination. In detail, the adaptive clustering procedures automatically adjust the number of clusters, while distance metrics control the adaptive clustering procedures, determining whether clusters should split or merge. The HCFL allows for different techniques at each tier, enhancing flexibility in choosing the optimal adaptive clustering methods or converting methods that rely on fixed cluster numbers to adaptive ones.

**Tier 3: Adaptive clustering procedures demonstrate how to modify cluster numbers.** To automatically determine the number of clusters, current approaches can be categorized into two orthogonal methods: (1) Splitting clusters to increase the number of clusters (Sattler et al., 2020b;a). (2) Merging clusters to reduce the number of clusters (Zeng et al., 2023). We unify these approaches at tier 3.

**Tier 4: Client distance metrics dictate when cluster numbers should be adjusted.** The client's distances are utilized to determine whether the current number of clusters should be adjusted. For instance, when the distance within a cluster is large, the cluster will divide into sub-clusters. Conversely, if the distances between two clusters are small, these two clusters should be merged. Existing clustering methods use various metrics such as cosine similarity of local gradients (Sattler et al., 2020b), gradients from a globally shared network (Zeng et al., 2023), and local feature norms (Wei & Huang, 2023).

## 4 HCFL$^+$: TACKLING REMAINING CHALLENGES IN CLUSTERED FL

Section 3 introduces a holistic clustering framework with four tiers to encompass existing methods. However, each tier still presents challenges that current methods cannot address. In this section, we

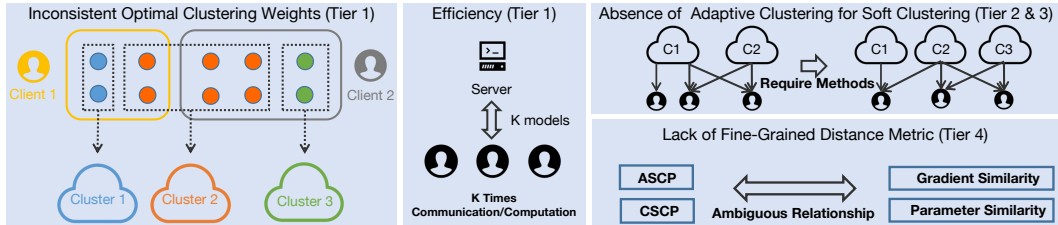

Figure 2: **Remaining challenges in clustered FL methods.** We identify four key issues in clustered FL algorithms: (1) inconsistent intra-client clustering weights, (2) efficiency concerns, (3) the absence of adaptive clustering for soft clustering methods, and (4) the lack of fine-grained distance metrics for various clustering principles. Clustering principles ASCP and CSCP differ in their approach as follows: ASCP assigns clients with any shifts into different clusters, while CSCP only assigns clients with concept shifts to different clusters."

outline four key remaining challenges in Figure 2 and introduce HCFL$^+$ to tackle them. Due to space constraints, we summarize the improved algorithm in Algorithm 2.

### 4.1 REMAINING CHALLENGES OF THE CLUSTERING IN FL

In this subsection, we identify four remaining challenges of the HCFL. We categorize these challenges by tiers in the HCFL, as shown in Figure 2. The details are provided below.

**Challenges on tier 1: Inconsistent intra-client clustering weights and efficiency concerns.** These challenges can be addressed by improving the clustering formulations.

- **Inconsistent intra-client clustering weights.** Existing approaches use the same clustering weights $\omega_{i;k}$ for all the samples belonging to client $i$ (Sattler et al., 2020b; Ghosh et al., 2020; Marfoq et al., 2021; Guo et al., 2023b). However, they overlook cases where the optimal clustering weights of different samples within the same client can be inconsistent, implying that $\omega_{i,j_1;k} \neq \omega_{i,j_2;k}$ for certain samples $(\mathbf{x}_{i,j_1}, y_{i,j_1})$ and $(\mathbf{x}_{i,j_2}, y_{i,j_2})$. See our example here[2].
- **Efficiency.** The current clustered FL methods (Marfoq et al., 2021; Long et al., 2023; Guo et al., 2023b) require K-fold higher communication or computation costs, hindering overall algorithm efficiency during deployment.

**Challenges on tiers 2 & 3: The absence of adaptive clustering for soft clustering methods.** Current adaptive clustering methods primarily address hard clustering (Sattler et al., 2020b;a; Zeng et al., 2023). Hence, there exists a gap between research and practice, as there is a need to automatically determine the number of clusters for soft clustering methods (Marfoq et al., 2021; Guo et al., 2023b).

**Challenges on tier 4: Lack of fine-grained distance metrics for various clustering principles.** The clustering principles determine which clients should be assigned to the same clusters. Existing clustering methods may use different clustering principles, as described by ASCP and CSCP below:
- ASCP (Any Shift Type Clustering Principle): clients with any distribution shifts are placed into separate clusters (Marfoq et al., 2021; Wu et al., 2023).
- CSCP (Concept Shift Only Clustering Principle): only clients with concept shifts are assigned to separate clusters (Guo et al., 2023b).

As discussed in Section 3.2, client distances determine whether the current number of clusters should be changed, aligning with the role of clustering principles: if the current number of clusters cannot meet the requirements of the clustering principles, the cluster number should be adjusted. Consequently, we advocate for distance metrics to be closely tied to distribution shifts, ultimately aligning with clustering principles. Unfortunately, existing distance metrics, such as those based on local gradients or local model parameters (Sattler et al., 2020b; Zeng et al., 2023; Long et al., 2023; Yan et al., 2023), cannot establish a clear link to distribution shifts. As a result, current methods struggle to satisfy diverse and detailed clustering principles.

### 4.2 IMPROVE TIER1: INCONSISTENCY AND EFFICIENCY AWARE OBJECTIVE FUNCTIONS

To address tier 1 challenges, specifically, (i) inconsistent intra-client clustering weights and (ii) efficiency, we propose an extension of the objective function (Eq. (1)), which is defined as $\mathcal{L}(\phi, \Theta, \Omega, \tilde{\Omega})$ and includes the parameters $\phi$, $\Theta$, $\Omega$, and $\tilde{\Omega}$.

---

[2]When client $i$'s local distribution is a mixture of two distributions, namely, $(\mathbf{x}_{i,j_1}, y_{i,j_1})$ sampled from the first distribution and $(\mathbf{x}_{i,j_2}, y_{i,j_2})$ from the second distribution, the optimal clustering weights for $(\mathbf{x}_{i,j_1}, y_{i,j_1})$ and $(\mathbf{x}_{i,j_2}, y_{i,j_2})$ should be distinct.

- **Global shared feature extractor $\phi$, and cluster-specific predictors $\Theta = [\theta_1, \cdots, \theta_K]$.** Dividing the feature extractor $\phi$ and the predictors $\{\theta_k\}$ reduces communication and computation costs since the predictors are lightweight architectures, like linear classifier layers.
- **Sample-wise clustering weights $\Omega = [\omega_{1,1;1} \cdots, \omega_{M,N_M;K}]$ for enhanced training stage and client-wise clustering weights $\tilde{\Omega} = [\tilde{\omega}_{1;1} \cdots, \tilde{\omega}_{M;K}]$ for testing stage** [3]. We employ sample-specific clustering weights $(\omega_{i,j;k})$ during training to ensure that data samples from the same clients can contribute to different cluster models, resolving the issue of inconsistent intra-client clustering weights. Furthermore, during testing, when test-time label information is unavailable, we utilize client-specific weights $(\tilde{\omega}_{i;k})$ for each client and cluster.
- **Enhance the optimization of $\omega_{i,j;k}$ by regularizing the distance between $\tilde{\omega}_{i;k}$ and $\omega_{i,j;k}$.** Motivated by the intuition that "if data from the same clients have similar distributions, the corresponding clustering weights should be similar", we encourage $\tilde{\omega}_{i;k}$ and $\omega_{i,j;k}$ to be close to each other.

The following objective function is designed to meet our requirements.

$$\mathcal{L}(\phi, \Theta, \Omega, \tilde{\Omega}) = \underbrace{\frac{1}{N} \sum_{i=1}^{M} \sum_{j=1}^{N_i} \log \left( \sum_{k=1}^{K} \omega_{i,j;k} \mathcal{L}_k(\mathbf{x}_{i,j}, y_{ij}; \phi, \theta_k) \right)}_{\mathcal{A}_1} - \underbrace{\mu \sum_{i=1}^{M} \sum_{j=1}^{N_i} \left( \sum_{k=1}^{K} \tilde{\omega}_{i;k} \log \frac{\tilde{\omega}_{i;k}}{\omega_{i,j;k}} \right)}_{\mathcal{A}_2} \quad (2)$$

$$\text{s.t.} \sum_{k=1}^{K} \omega_{i,j;k} = 1, \forall i, j, \sum_{k=1}^{K} \tilde{\omega}_{i;k} = 1, \forall i, \quad \tilde{\Omega} = \arg\min_{\tilde{\Omega}} \left| \max_{\Omega} \mathcal{L}(\phi, \Theta, \Omega, \tilde{\Omega}) - \mathcal{L}(\phi, \Theta, \tilde{\Omega}, \tilde{\Omega}) \right|, \quad (3)$$

where $\mathcal{A}_1$ term is extended from (1) by using the global shared feature extractor $\phi$ and the sample-wise weights $\omega_{i,j;k}$. $\mathcal{A}_2$ focuses on regularizing the difference between the sample-wise clustering weights $\omega_{i,j;k}$ and the client-wise clustering weights $\tilde{\omega}_{i;k}$. The $\mu$ controls the strength of this regularization. We obtain $\tilde{\omega}_{i;k}$ by solving (3), where we aim to minimize the impact of replacing $\omega_{i,j;k}$ with $\tilde{\omega}_{i;k}$.

**Optimization of the proposed objective function.** Different from heuristic methods used in most studies to optimize (2) [4]. we aim to introduce a more interpretable approach here. In this approach, we maximize the objective functions (Eq. (2)) to obtain optimization steps. Specifically, we can update $\tilde{\omega}_{i;k}$, $\omega_{i,j;k}$, $\theta_k$, and $\phi$ by (4)–(7).

$$\gamma_{i,j;k}^{t+1} = \frac{\omega_{i,j;k}^{t} \mathcal{L}_k(\mathbf{x}_{i,j}, y_{ij}; \phi^t, \theta_k^t)}{\sum_{n=1}^{K} \omega_{i,j;n}^{t} \mathcal{L}_k(\mathbf{x}_{i,j}, y_{ij}; \phi^t, \theta_n^t)}, \quad \tilde{\gamma}_{i,j;k}^{t+1} = \frac{\tilde{\omega}_{i;k}^{t} \mathcal{L}_k(\mathbf{x}_{i,j}, y_{ij}; \phi^t, \theta_k^t)}{\sum_{n=1}^{K} \omega_{i;n}^{t} \mathcal{L}_k(\mathbf{x}_{i,j}, y_{ij}; \phi^t, \theta_n^t)}, \quad (4)$$

$$\tilde{\omega}_{i;k}^{t+1} = \frac{1}{N_i} \sum_{j=1}^{N_i} \tilde{\gamma}_{i,j;k}^{t+1}, \quad \omega_{i,j;k}^{t+1} = \frac{\gamma_{i,j;k}^{t+1}}{1 + \mu N} + \frac{\mu N}{1 + \mu N} \tilde{\omega}_{i;k}^{t+1} = \tilde{\mu} \gamma_{i,j;k}^{t+1} + (1 - \tilde{\mu}) \tilde{\omega}_{i;k}^{t+1}, \quad (5)$$

$$\theta_k^{t+1} = \theta_k^t - \eta \sum_{i=1}^{M} \sum_{j=1}^{N_i} \frac{\gamma_{i,j;k}^{t+1}}{\mathcal{L}_k(\mathbf{x}_{ij}, y_{ij}, \phi^t, \theta_k^t)} \nabla_{\theta_k} \mathcal{L}_k(\mathbf{x}_{ij}, y_{ij}, \phi^t, \theta_k^t), \quad (6)$$

$$\phi^{t+1} = \phi^t - \eta \sum_{i=1}^{M} \sum_{j=1}^{N_i} \sum_{k=1}^{K} \frac{\gamma_{i,j;k}^{t+1}}{\mathcal{L}_k(\mathbf{x}_{ij}, y_{ij}, \phi^t, \theta_k^t)} \nabla_{\phi} \mathcal{L}_k(\mathbf{x}_{ij}, y_{ij}, \phi^t, \theta_k^{t+1}), \quad (7)$$

where $\gamma_{i,j;k}$ and $\tilde{\gamma}_{i,j;k}$ are intermediate results for calculating $\omega_{i,j;k}$ and $\tilde{\omega}_{i;k}$. More detailed proofs can be found in Appendix A. $\tilde{\mu} = \frac{1}{1+\mu N}$ serves as a hyperparameter to control the strength of the penalty term in Equation (2).

**Theoretical Results on Linear Representation Learning Case.** We examine the convergence of a linear representation learning problem, as extended from the settings in Collins et al. (2021); Tziotis et al. (2022). We assume that the clustering weights, denoted as $\omega_{i,j;k}$, are obtained in each communication round. We assume that local data $\mathbf{x}_{i,j} \in \mathbb{R}^d$, and the global shared feature extractor is parameterized by $\mathbf{B} \in \mathbb{R}^{d \times c}$. For each underlying cluster $k$, we define $\theta_k \in \mathbb{R}^c$, and the labels for data $\mathbf{x}_{i,j}$ belonging to cluster $k$ are given by $y_{i,j} = (\theta_k^*)^T (\mathbf{B}^*)^T \mathbf{x}_{i,j} + z_k$, where $z_k \sim \mathcal{N}(0, \sigma^2)$ captures the heterogeneous between $K$ underlying clusters. The global empirical risk is defined as the mean square error

$$\min_{\mathbf{B}, \Theta} \frac{1}{2N} \sum_{i=1}^{M} \sum_{j=1}^{N_i} \left( y_{i,j} - \sum_{k=1}^{K} \omega_{i,j;k} \theta_k^T \mathbf{B}^T \mathbf{x}_{i,j} \right)^2, \quad (8)$$

---

[3]Experiments on the effectiveness of sample-wise clustering weights in Figures 4(a) and 4(b).

[4]IFCA (Ghosh et al., 2020) sets $\omega_{i,j;k_{i,\min}} = 1, \forall j$ when $k_{i,\min} = \arg\min_k \mathbb{E}_{D_i}[f_{i;k}(\mathbf{x}_{i,j}, y_{i,j}, \phi, \theta_k)]$, where $f_{i;k}$ is the local loss function. FeSEM (Long et al., 2023) sets $k_{i,\min} = \arg\min_k \|\theta_k - \theta_i\|_2$, where $\theta_k, \theta_i$ represents the model parameters of cluster $k$ and client $i$, respectively.

where $\boldsymbol{\Theta} = [\boldsymbol{\theta}_1, \cdots, \boldsymbol{\theta}_K]$. Then we can derive the convergence of HCFL$^+$ given the following theorem. Detailed definitions, assumptions proofs, and discussions refer to Appendix B.

**Theorem 4.1** (Convergence of HCFL$^+$). *Under Assumption 1- 4, when we have $N \geq \frac{K^2}{d+c}$, and* $\min_k \hat{N}_k \geq \mathcal{C} \frac{c^3(1+\sigma^2)^4 \log(M)}{E_0^2} \min\left\{\frac{1}{\kappa^2}, \bar{\sigma}_{\min}^2\right\}$ *for some constant $\mathcal{C}$, we have*

$$dist(\hat{\mathbf{B}}^{t+1}, \hat{\mathbf{B}}^*) \leq dist(\hat{\mathbf{B}}^t, \hat{\mathbf{B}}^*)(1 - c_{min} + \frac{57}{200}c_{max})(1 - \frac{1}{2}c_{max})^{-1/2} + (\frac{7}{100}c_{max})(1 - \frac{1}{2}c_{max})^{-1/2}, \quad (9)$$

*with the probability at least* $1 - \exp(-90(d+c)) - \exp(-90c^2 \log(M))$. *Here* $\hat{N}_k = \sum_{i=1}^M \sum_{j=1}^{N_i} \omega_{i,j;k}$, $E_0 = 1 - dist^2(\hat{\mathbf{B}}^0, \hat{\mathbf{B}}^*)$, $c_{min} = \eta K \frac{\min_k \hat{N}_k}{N} \bar{\sigma}_{\min,*}^2 E_0$, *and* $c_{max} = \eta K \frac{\max_k \hat{N}_k}{N} \bar{\sigma}_{\min,*}^2 E_0$.

### 4.3 IMPROVE TIERS 2 & 3: ADAPTIVE CLUSTERING FOR SOFT CLUSTERING PARADIGMS

Given the limitations of existing adaptive clustering methods, we have extended the clustering weight update mechanisms to incorporate soft clustering and have verified its effectiveness in Figures 4(c) and 4(d). The overall process is summarized in Algorithms 4 and 5. In Algorithm 4, the clustering weights are adjusted after splitting cluster $k$ into two sub-clusters, denoted by $k_1$ and $k_2$. Then we set $\omega_{i,j,k_1} = \omega_{i,j,k_2} = \omega_{i,j,k}/2$ for all $i$ and $j$. In Algorithm 5, the clustering weights are updated when removing cluster $k$. For all $k' \neq k$, we modify $\omega_{i,j;k'}$ as $\omega_{i,j;k'} = \frac{\omega_{i,j;k'}}{\sum_{n \neq k} \omega_{i,j;n}}$.

We use the hyperparameter $\rho$ to control cluster splitting. As evidenced in Table 1, a higher $\rho$ results in fewer clusters, signifying enhanced generalization but reduced personalization. In detail, the cluster $k$ will split if the following condition is met:

$$\max(\mathbf{D}_k) - \text{mean}(\mathbf{D}_k) \geq \rho, \quad (10)$$

where $\mathbf{D}_k$ is the distance matrix of cluster $k$. We identify the need for cluster removal when the cluster no longer receives the highest clustering weights from any clients. Additional details about the enhanced adaptive process can be found in Algorithm 2.

### 4.4 IMPROVE TIER4: FINE-GRAINED DISTANCE METRIC DESIGN

Due to the page limitations, we include most of the details about the method design and practical implements in Appendix D. As discussed in Section 4.1, various algorithms may group clients into different clusters based on different clustering principles. Therefore, in this section, we design the following fine-grained distance metrics for these different clustering principles:

$$\mathbf{D}_{i,j}^k = \begin{cases} \max\{d_c, d_{lf}\} \mathbb{E}_{D_i}\left[\tilde{\mathcal{L}}_k(\mathbf{z}, y; \boldsymbol{\theta}_k)\right] \mathbb{E}_{D_j}\left[\tilde{\mathcal{L}}_k(\mathbf{z}, y; \boldsymbol{\theta}_k)\right], & \text{ASCP}, \\ d_c \mathbb{E}_{D_i}\left[\tilde{\mathcal{L}}_k(\mathbf{z}, y; \boldsymbol{\theta}_k)\right] \mathbb{E}_{D_j}\left[\tilde{\mathcal{L}}_k(\mathbf{z}, y; \boldsymbol{\theta}_k)\right], & \text{CSCP}, \end{cases} \quad (11)$$

where dist is the cos-similarity, $d_c = \max_y\left\{\text{dist}\left(\mathbb{E}_{D_i}\left[\mathcal{P}(\mathbf{z}|\mathbf{x}, y; \boldsymbol{\phi})\right], \mathbb{E}_{D_j}\left[\mathcal{P}(\mathbf{z}|\mathbf{x}, y; \boldsymbol{\phi})\right]\right)\right\}$, and $d_{lf} = \text{dist}\left(\mathbb{E}_{D_i}\left[\mathcal{P}(\mathbf{z}|\mathbf{x}; \boldsymbol{\phi})\right], \mathbb{E}_{D_j}\left[\mathcal{P}(\mathbf{z}|\mathbf{x}; \boldsymbol{\phi})\right]\right)$. The distances above become large only when the following conditions occur together: (1) Large values of $d_c$ indicate concept shifts between clients $i$ and $j$; (2) Large $d_{lf}$ indicate significant feature and label distribution differences. (2) Large values of $\mathbb{E}_{D_i}\left[\tilde{\mathcal{L}}_k(\mathbf{z}, y; \boldsymbol{\theta}_k)\right] \mathbb{E}_{D_j}\left[\tilde{\mathcal{L}}_k(\mathbf{z}, y; \boldsymbol{\theta}_k)\right]$ indicate incorrect clustering weights with high confidence. The effectiveness of the above distance metrics design is evidenced in Table 2.

## 5 NUMERICAL RESULTS

In this section, we evaluate the performance of HCFL$^+$ and other clustered FL methods. Additional experiment results, including hyper-parameter ablation studies, different model architectures, and additional scenarios, can be found in Appendix E.

### 5.1 DATASETS AND EXPERIMENT SETTINGS

**Diverse distribution shifts scenarios.** We establish clients with three types of distribution shifts. For label distribution shifts, we employ LDA with $\alpha = 1.0$, as introduced by Yoshida et al. (2019); Hsu et al. (2019); Reddi et al. (2021). For feature distribution shifts, we adopt the methodology from CIFAR10-C and CIFAR100-C creation (Hendrycks & Dietterich, 2019). Regarding concept shift, we draw inspiration from Guo et al. (2023b); Jothimurugesan et al. (2023), and selectively swap labels based on the parameter $\beta$. For example, with $\beta = 0.1$ for CIFAR10, two labels per concept are swapped, while the remaining eight labels remain unchanged. By default, we create three concepts in the experiments. More details about the construction of scenarios are included in Appendix E.1.

Table 1: **Performance of the adaptive clustering methods** on CIFAR10, CIFAR100, and Tiny-Imagenet datasets. For each algorithm, we present the best Validation and Test accuracies. For clustering methods that require a fixed number of clusters, we set $K = 3$. The hyperparameters $\text{tol}_1$, $\text{tol}_2$, $\alpha^*(0)$, $\tau$, and $\rho$ in adaptive clustering methods govern the balance between personalization and generalization, as well as the cluster number. For instance, lower $\tau$ in StoCFL or lower $\rho$ in HCFL$^+$ indicate improved personalization and reduced generalization. $K^T$ denotes the cluster number in the final training round, where a larger $K^T$ suggests enhanced personalization and reduced generalization. We emphasize the best results in **bold** and the worst results in blue.

| Algorithm | CIFAR10, $\beta = 0.2$ | | | CIFAR100, $\beta = 0.2$ | | | Tiny-Imagenet, $\beta = 0.2$ | | |
|---|---|---|---|---|---|---|---|---|---|
| | Val | Test | $K^T$ | Val | Test | $K^T$ | Val | Test | $K^T$ |
| FedAvg | 49.19 ±2.15 | 45.42 ±2.42 | 1 | 26.01 ±1.15 | 27.87 ±2.12 | 1 | 38.83 ±0.20 | 39.07 ±0.44 | 1 |
| FeSEM | 45.30 ±0.40 | 29.01 ±0.79 | 3 | 26.37 ±0.64 | 24.50 ±0.28 | 3 | 37.10 ±0.80 | 30.00 ±2.02 | 3 |
| IFCA | 34.46 ±2.06 | 23.18 ±2.55 | 3 | 26.99 ±3.89 | 26.20 ±1.56 | 3 | 38.52 ±0.30 | 29.92 ±0.30 | 3 |
| FedEM | 66.49 ±0.69 | 53.64 ±1.61 | 3 | 29.75 ±0.47 | 24.18 ±0.03 | 3 | 42.00 ±0.74 | 39.25 ±0.31 | 3 |
| FedRC | 63.65 ±2.95 | 59.41 ±0.19 | 3 | 34.56 ±0.79 | **37.62** ±0.16 | 3 | 38.93 ±0.18 | 39.73 ±0.04 | 3 |
| CFL | | | | | | | | | |
| $\text{tol}_1 = 0.4, \text{tol}_2 = 1.6$ | 61.55 ±1.74 | 46.88 ±0.35 | 6 | 35.05 ±0.35 | 24.84 ±2.50 | 4 | 37.41 ±1.87 | 30.25 ±0.55 | 3 |
| $\text{tol}_1 = 0.4, \text{tol}_2 = 0.8$ | 65.06 ±3.34 | 45.74 ±4.01 | 9 | 36.98 ±3.37 | 22.00 ±1.88 | 5 | 40.36 ±3.55 | 28.82 ±0.71 | 4 |
| $\text{tol}_1 = 0.2, \text{tol}_2 = 0.8$ | 58.92 ±2.09 | 55.02 ±0.97 | 4 | 37.73 ±7.68 | 31.47 ±0.09 | 3 | 35.74 ±0.57 | 34.41 ±1.92 | 1 |
| ICFL | | | | | | | | | |
| $\alpha^*(0) = 0.85$ | 77.59 ±0.04 | 57.38 ±1.91 | 98 | 52.73 ±1.03 | 32.77 ±0.28 | 100 | 64.72 ±0.30 | 34.73 ±0.39 | 87 |
| $\alpha^*(0) = 0.98$ | 60.58 ±1.07 | 61.18 ±0.78 | 14 | 41.49 ±4.11 | 33.57 ±1.56 | 40 | 53.05 ±2.57 | 35.09 ±0.25 | 42 |
| StoCFL | | | | | | | | | |
| $\tau = 0.05$ | 59.79 ±1.34 | 57.35 ±0.92 | 15 | 29.97 ±0.47 | 31.40 ±2.16 | 4 | 31.85 ±0.08 | 31.39 ±0.87 | 1 |
| $\tau = 0.10$ | 70.84 ±1.58 | 51.72 ±0.07 | 54 | **69.76** ±2.57 | 9.42 ±0.07 | 89 | **67.48** ±1.53 | 13.03 ±0.67 | 91 |
| HCFL$^+$ (FeSEM) | | | | | | | | | |
| $\rho = 0.05$ | **87.77** ±1.11 | 41.85 ±4.11 | 58 | 69.25 ±0.69 | 14.24 ±1.93 | 67 | 60.44 ±0.86 | 23.14 ±1.46 | 32 |
| $\rho = 0.1$ | 85.08 ±0.11 | 43.34 ±0.94 | 44 | 62.32 ±0.23 | 16.67 ±2.97 | 38 | 52.18 ±2.90 | 32.97 ±1.27 | 14 |
| $\rho = 0.3$ | 79.31 ±3.95 | 47.62 ±2.90 | 17 | 44.49 ±1.57 | 28.03 ±0.85 | 8 | 45.76 ±0.09 | 36.08 ±1.25 | 4 |
| HCFL$^+$ (FedEM) | | | | | | | | | |
| $\rho = 0.05$ | 82.45 ±0.13 | 57.73 ±1.70 | 22 | 60.36 ±1.47 | 22.95 ±1.44 | 40 | 63.41 ±0.05 | 34.24 ±0.33 | 33 |
| $\rho = 0.1$ | 84.64 ±1.47 | 60.90 ±0.61 | 16 | 62.98 ±0.42 | 26.17 ±1.22 | 34 | 59.88 ±0.11 | 37.17 ±0.37 | 20 |
| $\rho = 0.3$ | 83.67 ±0.72 | 62.43 ±0.71 | 10 | 50.72 ±2.97 | 32.13 ±0.18 | 9 | 45.53 ±0.53 | 38.64 ±0.23 | 3 |
| HCFL$^+$ (FedRC) | | | | | | | | | |
| $\rho = 0.05$ | 69.16 ±0.65 | 67.37 ±0.42 | 8 | 39.20 ±0.31 | 34.38 ±0.64 | 11 | 43.78 ±0.31 | 38.75 ±0.54 | 10 |
| $\rho = 0.1$ | 71.67 ±0.83 | 68.64 ±0.76 | 8 | 39.56 ±0.14 | 34.62 ±0.78 | 8 | 44.26 ±0.10 | 38.82 ±0.77 | 6 |
| $\rho = 0.3$ | 69.33 ±0.24 | **69.67** ±1.27 | 3 | 39.97 ±0.21 | **36.50** ±0.28 | 4 | 42.60 ±0.21 | **40.65** ±0.36 | 3 |

**Baselines.** We use FedAvg (McMahan et al., 2016) as a single-model FL example. We consider the most recently published clustered FL methods as our baselines. For clustered FL with fixed cluster number, we select IFCA (Ghosh et al., 2020), FedEM (Marfoq et al., 2021), FeSEM (Long et al., 2023), and FedRC (Guo et al., 2023b). For the adaptive clustering FL methods, we choose CFL (Sattler et al., 2020b), ICFL (Yan et al., 2023), and StoCFL (Zeng et al., 2023).

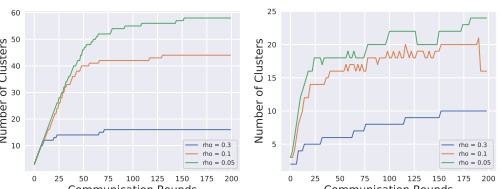

(a) HCFL$^+$ (FeSEM)  (b) HCFL$^+$ (FedEM)

Figure 3: **Number of clusters in HCFL$^+$ over communication rounds.** We illustrate changes in cluster numbers across communication rounds for various $\rho$ values using the CIFAR-10 dataset in our experiments.

**Experiment settings.** Unless specifically mentioned, we divide the datasets into 100 clients and execute all algorithms for 200 communication rounds. Additional settings are provided in Appendix E.2. We conducted all experiments using MobileNet-V2 (Sandler et al., 2018) and results on ResNet18 defer to Table 7 of Appendix E.

**Evaluation metrics.** We present the following metrics to evaluate the personalization and generalization abilities of the algorithms: (1) Validation Accuracy for evaluating personalization: The average accuracy on local validation datasets that match the distribution of local training sets. (2) Test Accuracy evaluating generalization: The average accuracy on global shared test datasets.

## 5.2 RESULTS ON DIVERSE DISTRIBUTION SHIFTS SCENARIOS

In this section, we compare the performance of HCFL$^+$ with other clustered FL methods. We also perform ablation studies to confirm the effectiveness of HCFL$^+$'s proposed components.

**HCFL$^+$ achieves better personalization-generalization trade-offs and comparable performance.** We highlight some key observations in Table 1. **A.** HCFL$^+$ consistently achieves superior test accuracy, with validation accuracy surpassing that of baseline methods with a similar number of clusters. This demonstrates improved efficiency and a better balance between personalization and generalization. **B.** Soft clustering methods like HCFL$^+$ (FedEM) and HCFL$^+$ (FedRC) outperform hard clustering methods in test accuracy, showcasing their superior generalization capabilities. **C.**

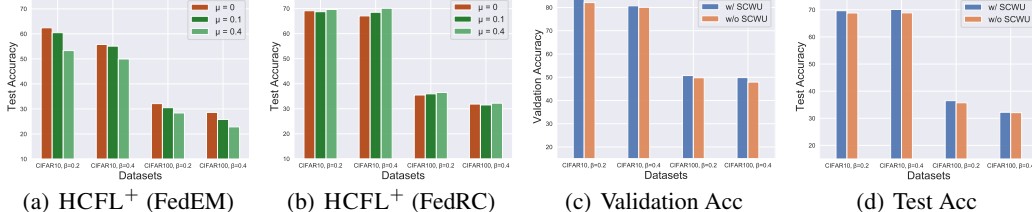

| (a) HCFL$^+$ (FedEM) | (b) HCFL$^+$ (FedRC) | (c) Validation Acc | (d) Test Acc |

Figure 4: **Ablation studies on Sections 4.2 and 4.3.** For Sec 4.2, we evaluated test accuracies of HCFL$^+$ using different backbones (FedEM and FedRC) and varying values of $\tilde{\mu}$, as shown in Figures 4(a) and 4(b). For Sec 4.3, we present the best Val and Test accuracy achieved by HCFL$^+$ with either FedEM or FedRC as backbones. "w/ SCWU" indicates the use of soft clustering weight updating mechanisms introduced in Section 4.3. More detailed results can be found in Tables 5 and 6 in Appendix E.

Table 2: **Ablation studies on Sec 4.4.** We conducted experiments on the CIFAR10 and CIFAR100 datasets, showcasing the highest test accuracies, the maximum number of clusters during training ($\max_t K^t$), and the final number of clusters ($K^T$) or each algorithm while maintaining a fixed value of $\rho = 0.3$. We used FedRC as the backbone, with 3 clusters identified as ideal by CSCP.

| Algorithm | CIFAR10, $\beta = 0.2$ | | | CIFAR10, $\beta = 0.4$ | | | CIFAR100, $\beta = 0.2$ | | | CIFAR100, $\beta = 0.4$ | | |
|---|---|---|---|---|---|---|---|---|---|---|---|---|
| | Test Acc | $\max_t K^t$ | $K^T$ | Test Acc | $\max_t K^t$ | $K^T$ | Test Acc | $\max_t K^t$ | $K^T$ | Test Acc | $\max_t K^t$ | $K^T$ |
| HCFL$^+$ | **69.67** ±1.27 | 4.5 | **3.0** | **70.13** ±0.42 | 7.0 | **6.0** | **36.50** ±0.28 | 3.5 | 3.5 | **32.22** ±0.20 | 5.0 | **4.0** |
| + ① | 67.83 ±1.70 | 9.5 | 7.0 | 64.53 ±0.23 | 10.5 | 10.0 | **36.77** ±0.67 | 9.5 | 8.5 | 31.33 ±2.12 | 11.0 | 7.5 |
| + ② | 56.14 ±8.11 | 10.5 | 5.5 | 50.87 ±2.26 | 12.5 | 8.5 | 34.11 ±1.58 | 10.0 | 6.0 | **32.75** ±0.67 | 7.5 | 6.5 |
| + ② | 68.52 ±0.64 | 8.0 | 6.0 | 69.47 ±0.15 | 11.0 | 8.5 | 34.65 ±1.16 | 8.5 | 5.5 | 31.61 ±0.54 | 11.0 | 7.5 |
| + ③ | 68.82 ±0.59 | 5.5 | 3.5 | 65.74 ±0.09 | **7.0** | 7.0 | 35.97 ±0.80 | 4.0 | **3.5** | 31.72 ±0.59 | **4.5** | **4.0** |

While baseline methods may achieve higher validation accuracy by separating every client into different clusters (namely when the value of $K^T$ is close to 100), these trained clusters tend to overfit local distributions, resulting in significantly lower test accuracy. **D.** The extended algorithms, namely HCFL$^+$ (FeSEM), HCFL$^+$ (FedEM), and HCFL$^+$ (FedRC), outperform the original methods that rely on fixed cluster numbers significantly. Additionally, these extended algorithms can automatically adjust the number of clusters, making the algorithms more practical, as we illustrated in Figure 3.

**Ablation studies on Sec 4.2.** We perform ablation studies on $\tilde{\mu}$, which control the distance between sample-wise weights $\omega_{i,j;k}$ and client-wise weights $\tilde{\omega}_{i;k}$ in Figures 4(a) and 4(b). A larger $\tilde{\mu}$ signifies a greater difference between $\omega_{i,j;k}$ and $\tilde{\omega}_{i;k}$. Our results show that HCFL$^+$ (FedEM) prefers smaller distance between $\omega_{i,j;k}$ and $\tilde{\omega}_{i;k}$. However, HCFL$^+$ (FedRC) prefers larger $\tilde{\mu}$ values, highlighting the necessity of different clustering weights among samples within the same clients.

**Ablation studies on Sec 4.3.** In Figures 4(c) and 4(d), we perform ablation studies on the soft clustering weight updating mechanism (w/ SCWU) introduced in Section 4.3. The term w/o SCWU refers to using the traditional clustering weight updating mechanism as described in Sattler et al. (2020b); Zeng et al. (2023). The results demonstrate that our proposed SCWU consistently achieves better performance in terms of both validation and test accuracies.

**Ablation studies on techniques in Sec 4.4.** We perform ablation studies to demonstrate the effectiveness of the designed distance metrics in Sec 4.4. The ablation studies include: ① Using gradient similarity, as in previous works (Sattler et al., 2020b; Yan et al., 2023), instead of distance on $\mathcal{P}(\mathbf{z}|x; \phi)$ and $\mathcal{P}(\mathbf{z}|x, y; \phi)$, as we proposed in Equation 11; ② Remove $\mathbb{E}_{D_i}[\tilde{\mathcal{L}}_k(\mathbf{z}, y; \boldsymbol{\theta}_k)]$ and $\mathbb{E}_{D_i}[\tilde{\mathcal{L}}_k(\mathbf{z}, y; \boldsymbol{\theta}_k)]$ in (11); ③ Using mean distances instead of maximum distances in (11). The results show that HCFL$^+$ consistently achieves the highest test accuracy and produces a number of clusters closer to the ideal number than other ablation studies.

## 6 CONCLUSION

In this paper, we introduce HCFL, a comprehensive clustered FL framework that unifies existing methods while enabling the integration of diverse algorithms to gather the advantages of various clustered FL approaches. Additionally, we identify persistent challenges unaddressed by current algorithms and propose HCFL$^+$ as a solution. The HCFL is flexible and can generate numerous clustered FL methods by altering techniques in each tier. Though we have chosen some typical components and demonstrated their effectiveness, conducting further performance verification with more choices in each tier would be beneficial.

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

CONTENTS OF APPENDIX

## A  PROOF OF OPTIMIZATION STEPS

**Theorem A.1.** *Given objective function* $\mathcal{L}(\phi, \boldsymbol{\Theta}, \boldsymbol{\Omega}, \tilde{\boldsymbol{\Omega}})$

$$
\mathcal{L}(\phi, \boldsymbol{\Theta}, \boldsymbol{\Omega}, \tilde{\boldsymbol{\Omega}}) = \frac{1}{N} \sum_{i=1}^{M} \sum_{j=1}^{N_i} \log \left( \sum_{k=1}^{K} \omega_{i,j;k} \mathcal{L}_k(\mathbf{x}_{i,j}, y_{ij}; \phi, \boldsymbol{\theta}_k) \right)
$$
$$
+ \sum_{i=1}^{M} \sum_{j=1}^{N_i} \lambda_{i,j} \left( \sum_{k=1}^{K} \omega_{i,j;k} - 1 \right)
$$
$$
- \mu \sum_{i=1}^{M} \sum_{j=1}^{N_i} \left( \sum_{k=1}^{K} \tilde{\omega}_{i;k} \log \frac{\tilde{\omega}_{i;k}}{\omega_{i,j;k}} \right) , \tag{12}
$$

*and we define* $\tilde{\boldsymbol{\Omega}} = \{\tilde{\omega}_{i;k} | \forall i, k\}$*, then* $\tilde{\boldsymbol{\Omega}}$ *is obtained by*

$$
\tilde{\boldsymbol{\Omega}} = \arg\min_{\tilde{\boldsymbol{\Omega}}} \left| \max_{\boldsymbol{\Omega}} \mathcal{L}(\phi, \boldsymbol{\Theta}, \boldsymbol{\Omega}, \boldsymbol{\Omega}) - \mathcal{L}(\phi, \boldsymbol{\Theta}, \tilde{\boldsymbol{\Omega}}, \tilde{\boldsymbol{\Omega}}) \right| . \tag{13}
$$

*Then E-M steps are obtained by maximizing* $\mathcal{L}(\phi, \boldsymbol{\Theta}, \boldsymbol{\Omega}, \tilde{\boldsymbol{\Omega}})$*.*

$$
\gamma_{i,j;k}^{t+1} = \frac{\omega_{i,j;k}^{t} \mathcal{L}_k(\mathbf{x}_{i,j}, y_{ij}; \boldsymbol{\theta}_k^t)}{\sum_{n=1}^{K} \omega_{i,j;n}^{t} \mathcal{L}_k(\mathbf{x}_{i,j}, y_{ij}; \boldsymbol{\theta}_n^t)} , \tag{14}
$$

$$
\tilde{\gamma}_{i,j;k}^{t+1} = \frac{\tilde{\omega}_{i;k}^{t} \mathcal{L}_k(\mathbf{x}_{i,j}, y_{ij}; \boldsymbol{\theta}_k^t)}{\sum_{n=1}^{K} \omega_{i;n}^{t} \mathcal{L}_k(\mathbf{x}_{i,j}, y_{ij}; \boldsymbol{\theta}_n^t)} , \tag{15}
$$

$$
\tilde{\omega}_{i;k}^{t+1} = \frac{1}{N_i} \sum_{j=1}^{N_i} \tilde{\gamma}_{i,j;k}^{t+1} , \tag{16}
$$

$$
\omega_{i,j;k}^{t+1} = \frac{\gamma_{i,j;k}^{t+1}}{1 + \mu N} + \frac{\mu N}{1 + \mu N} \tilde{\omega}_{i;k}^{t+1} , \tag{17}
$$

$$
\boldsymbol{\theta}_k^{t+1} = \boldsymbol{\theta}_k^t - \eta \sum_{i=1}^{M} \sum_{j=1}^{N_i} \frac{\gamma_{i,j;k}^{t+1}}{\mathcal{L}_k(\mathbf{x}_{ij}, y_{ij}, \phi^t, \boldsymbol{\theta}_k^t)} \nabla_{\boldsymbol{\theta}_k} \mathcal{L}_k(\mathbf{x}_{ij}, y_{ij}, \phi^t, \boldsymbol{\theta}_k^t) , \tag{18}
$$

$$
\phi^{t+1} = \phi^t - \eta \sum_{i=1}^{M} \sum_{j=1}^{N_i} \sum_{k=1}^{K} \frac{\gamma_{i,j;k}^{t+1}}{\mathcal{L}_k(\mathbf{x}_{ij}, y_{ij}, \phi^t, \boldsymbol{\theta}_k^t)} \nabla_{\phi} \mathcal{L}_k(\mathbf{x}_{ij}, y_{ij}, \phi^t, \boldsymbol{\theta}_k^{t+1}) \tag{19}
$$

*Proof.* Let's begin by assuming $\tilde{\omega}_{i;k}^{t+1}$ is given for each round $t$, then we will discuss how to compute $\omega_{i,j;k}^{t+1}$. Consider the objective function $\mathcal{L}(\phi, \Theta, \Omega, \tilde{\Omega})$, we have

$$\frac{\partial \mathcal{L}(\phi, \Theta, \Omega, \tilde{\Omega})}{\partial \omega_{i,j;k}} = \frac{1}{N} \frac{\mathcal{L}_k(\mathbf{x}_{i,j}, y_{ij}; \boldsymbol{\theta}_n)}{\sum_{n=1}^{K} \omega_{i,j;n} \mathcal{L}_n(\mathbf{x}_{i,j}, y_{ij}; \boldsymbol{\theta}_n)} + \lambda_{i,j} + \mu \frac{\tilde{\omega}_{i;k}}{\omega_{i,j;k}} . \tag{20}$$

Then define

$$\gamma_{i,j;k} = \frac{\omega_{i,j;k} \mathcal{L}_k(\mathbf{x}_{i,j}, y_{ij}; \boldsymbol{\theta}_n)}{\sum_{n=1}^{K} \omega_{i,j;n} \mathcal{L}_n(\mathbf{x}_{i,j}, y_{ij}; \boldsymbol{\theta}_n)} , \tag{21}$$

and take $\frac{\partial \mathcal{L}(\Theta, \Omega)}{\partial \omega_{i,j;k}} = 0$ we have

$$\frac{\gamma_{i,j;k}}{N} + \mu \tilde{\omega}_{i;k} = -\lambda_{i,j} \omega_{i,j;k} . \tag{22}$$

Then we have

$$\omega_{i,j;k} = -\frac{1}{\lambda_{i,j}} \left( \frac{\gamma_{i,j;k}}{N} + \mu \tilde{\omega}_{i;k} \right) . \tag{23}$$

Because we have $\sum_{k=1}^{K} \omega_{i,j;k} = 1$, we have

$$1 = -\frac{1}{\lambda_{i,j}} \left( \frac{1}{N} + \mu \right) \tag{24}$$

$$\lambda_{i,j} = -\frac{1 + \mu N}{N} . \tag{25}$$

Then we have

$$\omega_{i,j;k} = \frac{\gamma_{i,j;k}}{1 + \mu N} + \frac{\mu N}{1 + \mu N} \tilde{\omega}_{i;k} . \tag{26}$$

Then consider to optimize $\boldsymbol{\theta}_k$, we have,

$$\frac{\partial \mathcal{L}(\phi, \Theta, \Omega, \tilde{\Omega})}{\partial \boldsymbol{\theta}_k}$$

$$= \frac{1}{N} \sum_{i=1}^{M} \sum_{j=1}^{N_i} \frac{\omega_{i,j;k}}{\sum_{n=1}^{K} \omega_{i,j;n} \mathcal{L}_n(\mathbf{x}_{ij}, y_{ij}; \phi, \boldsymbol{\theta}_n)} \cdot \frac{\partial \mathcal{L}_k(\mathbf{x}_{ij}, y_{ij}; \phi, \boldsymbol{\theta}_k)}{\partial \boldsymbol{\theta}_k} , \tag{27}$$

$$= -\frac{1}{N} \sum_{i=1}^{M} \sum_{j=1}^{N_i} \frac{\gamma_{i,j;k}^{t+1}}{\mathcal{L}_k(\mathbf{x}_{ij}, y_{ij}; \phi, \boldsymbol{\theta}_k)} \nabla_{\boldsymbol{\theta}_k} \mathcal{L}_k(\mathbf{x}_{ij}, y_{ij}; \phi, \boldsymbol{\theta}_k) . \tag{28}$$

Finally, consider to optimize $\phi$, we have

$$\frac{\partial \mathcal{L}(\phi, \Theta, \Omega, \tilde{\Omega})}{\partial \phi}$$

$$= \frac{1}{N} \sum_{i=1}^{M} \sum_{j=1}^{N_i} \sum_{k=1}^{K} \frac{\omega_{i,j;k}}{\sum_{n=1}^{K} \omega_{i,j;n} \mathcal{L}_n(\mathbf{x}_{ij}, y_{ij}; \phi, \boldsymbol{\theta}_n)} \cdot \frac{\partial \mathcal{L}_k(\mathbf{x}_{ij}, y_{ij}; \phi, \boldsymbol{\theta}_k)}{\partial \phi} , \tag{29}$$

$$= -\frac{1}{N} \sum_{i=1}^{M} \sum_{j=1}^{N_i} \sum_{k=1}^{K} \frac{\gamma_{i,j;k}^{t+1}}{\mathcal{L}_k(\mathbf{x}_{ij}, y_{ij}; \phi, \boldsymbol{\theta}_k)} \nabla_{\phi} \mathcal{L}_k(\mathbf{x}_{ij}, y_{ij}; \phi, \boldsymbol{\theta}_k) . \tag{30}$$

Because if hard to find a close-form solution to $\frac{\partial \mathcal{L}(\phi, \Theta, \Omega, \tilde{\Omega})}{\partial \boldsymbol{\theta}_k} = 0$ when $\boldsymbol{\theta}_k$ is the parameter of deep neural networks, we use gradient ascent to optimize $\boldsymbol{\theta}_k$. The same method is used for feature extractors $\phi$.

Then the remaining thing is how to decide $\tilde{\omega}_{i;k}$. From the formulation of objective function, the $\tilde{\omega}_{i;k}$ is decided by

$$\tilde{\Omega} = \arg\min_{\tilde{\Omega}} \left| \max_{\Omega} \mathcal{L}(\phi, \Theta, \Omega, \Omega) - \mathcal{L}(\phi, \Theta, \tilde{\Omega}, \tilde{\Omega}) \right| . \tag{31}$$

To solve this problem, firstly, we can transform the definition of $\mathcal{L}(\phi, \Theta, \tilde{\Omega}, \tilde{\Omega})$ to

$$\mathcal{L}(\phi, \Theta, \tilde{\Omega}, \tilde{\Omega}) = \frac{1}{N} \sum_{i=1}^{M} \sum_{j=1}^{N_i} \log \left( \sum_{k=1}^{K} \omega_{i,j;k} \mathcal{L}_k(\mathbf{x}_{i,j}, y_{ij}; \phi, \boldsymbol{\theta}_k) \right)$$
$$+ \sum_{i=1}^{M} \sum_{j=1}^{N_i} \lambda_{i,j} \left( \sum_{k=1}^{K} \omega_{i,j;k} - 1 \right)$$
$$- \mu \sum_{i=1}^{M} \sum_{j=1}^{N_i} \left( \sum_{k=1}^{K} \tilde{\omega}_{i;k} \log \frac{\tilde{\omega}_{i;k}}{\omega_{i,j;k}} \right), \tag{32}$$

$$s.t. \ \omega_{i,j;k} = \tilde{\omega}_{i;k}, \ \forall i, j, k. \tag{33}$$

Then by removing the constrains, we can always find

$$\max_{\Omega} \mathcal{L}(\phi, \Theta, \Omega, \tilde{\Omega}) \geq \tilde{\mathcal{L}}(\phi, \Theta, \tilde{\Omega}, \tilde{\Omega}). \tag{34}$$

Then we have

$$\tilde{\Omega} = \arg\min_{\tilde{\Omega}} \left| \max_{\Omega} \mathcal{L}(\phi, \Theta, \Omega, \tilde{\Omega}) - \mathcal{L}(\phi, \Theta, \tilde{\Omega}, \tilde{\Omega}) \right| \tag{35}$$

$$= \arg\min_{\tilde{\Omega}} \left( \max_{\Omega} \mathcal{L}(\phi, \Theta, \Omega, \tilde{\Omega}) - \mathcal{L}(\phi, \Theta, \tilde{\Omega}, \tilde{\Omega}) \right) \tag{36}$$

$$= \arg\max_{\tilde{\Omega}} \mathcal{L}(\phi, \Theta, \tilde{\Omega}, \tilde{\Omega}) \tag{37}$$

Then we can obtain the results by directly use the proof in (Guo et al., 2023b) and (Marfoq et al., 2021). $\square$

## B  THEORETICAL STUDY ON LINEAR REPRESENTATION CASE

### B.1  DEFINITION AND ASSUMPTIONS

In this section, we target on the effectiveness of split-feature-classifier method. Therefore, we focus on a case study that clients are solving a linear representation learning problem, similar to the analysis in Collins et al. (2021); Tziotis et al. (2022), and assume the optimal $\omega_{i,j;k}$ is already given. In detail, we assume local data $\mathbf{x}_{i,j} \in \mathbb{R}^d$, and $\phi$ is a global shared projection onto a $c$-dimensional subspace $\mathbb{R}^d$, which is parameterized by matrix $\mathbf{B} \in \mathbb{R}^{d \times c}$. Besides, for each underlying distribution, we have $\boldsymbol{\theta}_k \in \mathbb{R}^c$, and the labels of each sample $\mathbf{x}_{i,j}$ belongs to the distribution $k$ is given by $y_{i,j} = \boldsymbol{\theta}_k^{*T} \mathbf{B}^{*T} x_{i,j} + z_k$, where $\boldsymbol{\theta}_k^*$ and $\mathbf{B}^*$ are ground truth parameters, and $z_k \sim \mathcal{N}(0, \sigma^2)$ is to capture the heterogeneous between $K$ underlying distributions. Under these assumptions, the global empirical risk is

$$\min_{\mathbf{B}, \Theta} \frac{1}{2N} \sum_{i=1}^{M} \sum_{j=1}^{N_i} \left( y_{i,j} - \sum_{k=1}^{K} \omega_{i,j;k} \boldsymbol{\theta}_k^T \mathbf{B}^T \mathbf{x}_{i,j} \right)^2 \tag{38}$$

The distance that measuring the distance between sub-spaces is defined as follows (Jain et al., 2013; Collins et al., 2021; Tziotis et al., 2022),

**Definition B.1.** *The principal angle distance between the column spaces of $\mathbf{B}_1, \mathbf{B}_2 \in \mathbb{R}^{d \times c}$ is given by,*

$$dist(\mathbf{B}_1, \mathbf{B}_2) = \left\| \hat{\mathbf{B}}_{1,\perp}^T \hat{\mathbf{B}}_2 \right\|_2, \tag{39}$$

*where $\hat{\mathbf{B}}_{1,\perp}$ and $\hat{\mathbf{B}}_2$ are orthogonal matrices satisfying $span(\hat{\mathbf{B}}_{1,\perp}) = span(\mathbf{B}_1)^\perp$, and $span(\hat{\mathbf{B}}_2) = span(\mathbf{B}_2)$.*

**Definition B.2** ($\|\mathbf{A}\|_2$-sub-Gaussian)**.** *For a random vector $\mathbf{x} \in \mathbb{R}^d$ and a fixed matrix $\mathbf{A} \in \mathbb{R}^{d \times c}$, the vector $\mathbf{A}^T \mathbf{x}$ is called $\|\mathbf{A}\|_2$-sub-Gaussian if $\mathbf{y}^T \mathbf{A}^T \mathbf{x}$ is sub-Gaussian with sub-Gaussian norm $\|\mathbf{A}\|_2 \|\mathbf{y}\|_2$ for all $\mathbf{y} \in \mathbb{R}^c$, i.e., $\mathbb{E} \left[ \exp(\mathbf{y}^T \mathbf{A}^T \mathbf{x}) \right] \leq \exp(\|\mathbf{A}\|_2^2 \|\mathbf{y}\|_2^2 / 2)$.*

**Assumption 1** (Sub-Gaussian design). *The samples $\mathbf{x}_{i,j} \in \mathbb{R}^d$ are i.i.d. with mean $\mathbf{0}$, covariance $\mathbf{I}_d$, and are $\mathbf{I}_d$-sub-Gaussian, i.e., $\mathbb{E}\left[e^{\mathbf{v}^T \mathbf{x}_{i,j}}\right] \le e^{\|\mathbf{v}\|_2^2/2}$ for all $\mathbf{v} \in \mathbb{R}^d$.*

**Assumption 2** (Underlying distribution diversity). *Let $\bar{\sigma}_{min,*}$ be the minimum singular value of any matrix $\bar{W} \in \mathbb{R}^{K \times c}$ with rows being a $K$-sized subset of ground-truth distribution-specific parameters $\{\boldsymbol{\theta}_1, \cdots, \boldsymbol{\theta}_K\}$. Then $\bar{\sigma}_{min,*} > 0$.*

**Assumption 3** (Client normalization). *The ground-truth distribution-specific parameters satisfy $\|\boldsymbol{\theta}_k^*\|_2 = \sqrt{c}$ for all $k \in [K]$, and $\mathbf{B}^*$ has orthogonal columns.*

**Assumption 4** (Binary weights). *We assume the value of $\omega_{i,j;k}$ is given in $\{0, 1\}$.*

Assumption 1, 2, 3 are widely used assumptions in the analysis of linear representation learning problem (Collins et al., 2021; Tziotis et al., 2022). We introduce Assumption 4 to simplify the analysis, and this assumption match the observation in our numerical experiments that the max weights $\max_k[\omega_{i,j;k}]$ usually close to 1 when converge.

### B.2 PRELIMINARY AND LEMMAS

We first introduce some important lemmas that we would like to use in the following parts of proof.

**Lemma B.3.** *Given orthogonal matrix $\hat{\mathbf{B}} \in \mathbb{R}^{d \times c}$, matrix $\mathbf{X} \in \mathbb{R}^{N \times d}$ that each row of $\mathbf{X}$ is sub-Gaussian (Assumption 1), and matrix $\bar{\mathbf{\Omega}}_k \in \mathbb{R}^{N \times d}$ valued by $\{0, 1\}$. Then let $\delta_k = \frac{12\mathcal{C}_{1;k}c^{3/2}\sqrt{\log(M)}}{\sqrt{\hat{N}_k}}$, we have*

$$\sigma_{\min}\left(\frac{1}{N}\hat{\mathbf{B}}^T\left(\bar{\mathbf{\Omega}}_k^T \odot \mathbf{X}^T\right)\left(\mathbf{X} \odot \bar{\mathbf{\Omega}}_k\right)\hat{\mathbf{B}}\right) \ge \frac{\hat{N}_k}{N}\left(1 - \delta_k\right), \tag{40}$$

*with probability at least $1 - \exp(121c^3 \log(M))$, and $\hat{N}_k = \sum_{i=1}^M \sum_{j=1}^{N_i} \mathbf{1}_{\omega_{i,j;k}=1}$.*

*Proof.* Firstly, we can rewrite

$$\frac{1}{N}\hat{\mathbf{B}}^T\left(\bar{\mathbf{\Omega}}_k^T \odot \mathbf{X}^T\right)\left(\mathbf{X} \odot \bar{\mathbf{\Omega}}_k\right)\hat{\mathbf{B}} = \frac{\hat{N}_k}{N}\sum_{i=1}^M\sum_{j=1}^{N_i}\left(\frac{\omega_{i,j;k}}{\sqrt{\hat{N}_k}}\hat{\mathbf{B}}^T\mathbf{x}_{i,j}\right)\left(\frac{\omega_{i,j;k}}{\sqrt{\hat{N}_k}}\hat{\mathbf{B}}^T\mathbf{x}_{i,j}\right)^T, \tag{41}$$

$$= \frac{\hat{N}_k}{N}\sum_{\omega_{i,j;k}=1}^N\left(\frac{1}{\sqrt{\hat{N}_k}}\hat{\mathbf{B}}^T\mathbf{x}_{i,j}\right)\left(\frac{1}{\sqrt{\hat{N}_k}}\hat{\mathbf{B}}^T\mathbf{x}_{i,j}\right)^T, \tag{42}$$

where $\hat{N}_k = \sum_{i=1}^M \sum_{j=1}^{N_i} \mathbf{1}_{\omega_{i,j;k}=1}$. Define $\mathbf{v}_{i,j;k} = \frac{\omega_{i,j;k}}{\sqrt{\hat{N}_k}}\hat{\mathbf{B}}^T\mathbf{x}_{i,j}$, by Definition B.2 and Assumption 1, we can observe that each $\mathbf{v}_{i,j;k}$ is either $\left\|\frac{1}{\sqrt{\hat{N}_k}}\hat{\mathbf{B}}^T\right\|_2$-sub-Gaussian or $\mathbf{v}_{i,j;k} = \mathbf{0}$. Then we can define $\mathbf{V}_k \in \mathbb{R}^{\hat{N}_k \times c}$, and each row of $\mathbf{V}_k$ is $\mathbf{v}_{i,j;k}$ that $\mathbf{v}_{i,j;k} \ne \mathbf{0}$. Then we have

$$\frac{1}{N}\hat{\mathbf{B}}^T\left(\bar{\mathbf{\Omega}}_k^T \odot \mathbf{X}^T\right)\left(\mathbf{X} \odot \bar{\mathbf{\Omega}}_k\right)\hat{\mathbf{B}} = \frac{\hat{N}_k}{N}\mathbf{V}_k^T\mathbf{V}_k. \tag{43}$$

Then based on Theorem 4.6.1, Equation (4.22) in Vershynin (2018), we have

$$\sigma_{min}\left(\frac{1}{N}\hat{\mathbf{B}}^T\left(\bar{\mathbf{\Omega}}_k^T \odot \mathbf{X}^T\right)\left(\mathbf{X} \odot \bar{\mathbf{\Omega}}_k\right)\hat{\mathbf{B}}\right) = \frac{\hat{N}_k}{N}\sigma_{min}\left(\mathbf{V}_k^T\mathbf{V}_k\right) \ge \frac{\hat{N}_k}{N}\left(1 - \delta_k\right) \tag{44}$$

, with probability at least $1 - \exp(-\left(\delta_k\sqrt{N_k}/\mathcal{C}_{1;k} - \sqrt{C}\right)^2)$ for constant $\mathcal{C}_{1;k}$. We then set $\delta_k = \frac{12\mathcal{C}_{1;k}c^{3/2}\sqrt{\log(M)}}{\sqrt{\hat{N}_k}}$, and we have

$$1 - \exp(-\left(\delta_k\sqrt{N_k}/\mathcal{C}_{1;k} - \sqrt{C}\right)^2) \ge 1 - \exp(121c^3\log(M)), \tag{45}$$

which finish the proof. □

**Lemma B.4.** *Given the objective function defined in Equation* (38)

$$\min_{\mathbf{B},\boldsymbol{\Theta}} \frac{1}{2N} \sum_{i=1}^{M} \sum_{j=1}^{N_i} \left( y_{i,j} - \sum_{k=1}^{K} \omega_{i,j;k} \boldsymbol{\theta}_k^T \mathbf{B}^T \mathbf{x}_{i,j} \right)^2, \tag{46}$$

*define the matrix form of* $\mathbf{x}_{i,j}$, $y_{i,j}$, *and* $\omega_{i,j;k}$ *by* $\mathbf{X} \in \mathbb{R}^{N \times d}$, $\mathbf{Y} \in \mathbb{R}^N$, $\boldsymbol{\Omega}_k \in \mathbb{R}^N$, *and define* $\bar{\boldsymbol{\Omega}}_k \in \mathbb{R}^{N \times d}$ *by repeat* $\boldsymbol{\Omega}_k$ *for* $d$ *times, we can have the following optimization steps to solve the above objective function*

$$\boldsymbol{\theta}_k^{t+1} = \left( \frac{1}{N} [\hat{\mathbf{B}}^t]^T \left( \bar{\boldsymbol{\Omega}}_k^T \odot \mathbf{X}^T \right) \left( \mathbf{X} \odot \bar{\boldsymbol{\Omega}}_k \right) \hat{\mathbf{B}}^t \right)^{-1} \left( \frac{1}{N} [\hat{\mathbf{B}}^t]^T \left( \bar{\boldsymbol{\Omega}}_k^T \odot \mathbf{X}^T \right) \left( \mathbf{Y} \odot \boldsymbol{\Omega}_k \right) \right), \tag{47}$$

$$\mathbf{B}^{t+1} = \hat{\mathbf{B}}^t - \sum_{k=1}^{K} \frac{\eta}{N} \left( \left( \bar{\boldsymbol{\Omega}}_k^T \odot \mathbf{X}^T \right) \left( \mathbf{X} \odot \bar{\boldsymbol{\Omega}}_k \right) \hat{\mathbf{B}} \boldsymbol{\theta}_k - \left( \bar{\boldsymbol{\Omega}}_k^T \odot \mathbf{X}^T \right) \left( \mathbf{Y} \odot \boldsymbol{\Omega}_k \right) \right) \boldsymbol{\theta}_k^T, \tag{48}$$

$$\hat{\mathbf{B}}^{t+1} = \mathbf{B}^{t+1} \left( \mathbf{R}^{t+1} \right)^{-1}, \tag{49}$$

*where* $\hat{\mathbf{B}}^{t+1} \mathbf{R}^{t+1}$ *is the QR factorization of* $\mathbf{B}^{t+1}$.

*Proof.* We first extend the empirical function via matrices $\mathbf{X}_i \in \mathbb{R}^{N_i \times d}$, $\mathbf{Y}_i \in \mathbb{R}^{N_i}$, and $\boldsymbol{\Omega}_{i;k} \in \mathbb{R}^{N_i}$, $\bar{\boldsymbol{\Omega}}_{i;k} \in \mathbb{R}^{N_i \times d}$ is repeat $\boldsymbol{\Omega}_{i;k} \in \mathbb{R}^{N_i}$ for $d$ times.

$$\mathcal{L}(\mathbf{B}, \boldsymbol{\Theta}) = \frac{1}{2N} \sum_{i=1}^{M} \left\| \sum_{k=1}^{K} \boldsymbol{\Omega}_{i;k} \odot \left( \mathbf{Y}_i - \mathbf{X}_i \hat{\mathbf{B}} \boldsymbol{\theta}_k \right) \right\|_2^2. \tag{50}$$

Define $\mathbf{X} \in \mathbb{R}^{N \times d} = [\mathbf{X}_1^T, \cdots, \mathbf{X}_M^T]^T$, $\bar{\boldsymbol{\Omega}}_k \in \mathbb{R}^{N \times d} = [\bar{\boldsymbol{\Omega}}_{1,k}^T, \cdots, \bar{\boldsymbol{\Omega}}_{M,k}^T]^T$, and $\boldsymbol{\Omega}_k \in \mathbb{R}^N = [\boldsymbol{\Omega}_{1,k}^T, \cdots, \boldsymbol{\Omega}_{M,k}^T]^T$, then compute the gradients of $\hat{B}$ and $\boldsymbol{\theta}_k$,

$$\frac{\partial \mathcal{L}(\mathbf{B}, \boldsymbol{\Theta})}{\partial \hat{B}} = -\frac{1}{N} \sum_{i=1}^{M} \sum_{k=1}^{K} \left( \bar{\boldsymbol{\Omega}}_{i;k} \odot \mathbf{X}_i \right)^T \left( \boldsymbol{\Omega}_{i;k} \odot \left( \mathbf{Y}_i - \mathbf{X}_i \hat{\mathbf{B}} \boldsymbol{\theta}_k \right) \right) \boldsymbol{\theta}_k^T, \tag{51}$$

$$= \frac{1}{N} \sum_{k=1}^{K} \left( \left( \bar{\boldsymbol{\Omega}}_k^T \odot \mathbf{X}^T \right) \left( \mathbf{X} \odot \bar{\boldsymbol{\Omega}}_k \right) \hat{\mathbf{B}} \boldsymbol{\theta}_k - \left( \bar{\boldsymbol{\Omega}}_k^T \odot \mathbf{X}^T \right) \left( \mathbf{Y} \odot \boldsymbol{\Omega}_k \right) \right) \boldsymbol{\theta}_k^T, \tag{52}$$

$$\frac{\partial \mathcal{L}(\mathbf{B}, \boldsymbol{\Theta})}{\partial \boldsymbol{\theta}_k} = -\frac{1}{N} \sum_{i=1}^{M} \left( \bar{\boldsymbol{\Omega}}_{i;k} \odot \mathbf{X}_i \hat{\mathbf{B}} \right)^T \left( \boldsymbol{\Omega}_{i;k} \odot \left( \mathbf{Y}_i - \mathbf{X}_i \hat{\mathbf{B}} \boldsymbol{\theta}_k \right) \right), \tag{53}$$

$$= \frac{1}{N} \left( \hat{\mathbf{B}}^T \left( \bar{\boldsymbol{\Omega}}_k^T \odot \mathbf{X}^T \right) \left( \mathbf{X} \odot \bar{\boldsymbol{\Omega}}_k \right) \hat{\mathbf{B}} \boldsymbol{\theta}_k - \hat{\mathbf{B}}^T \left( \bar{\boldsymbol{\Omega}}_k^T \odot \mathbf{X}^T \right) \left( \mathbf{Y} \odot \boldsymbol{\Omega}_k \right) \right). \tag{54}$$

Then we can define the following optimization steps based on the above gradients, similar to the analysis in Collins et al. (2021).

$$\boldsymbol{\theta}_k^{t+1} = \left( \frac{1}{N} [\hat{\mathbf{B}}^t]^T \left( \bar{\boldsymbol{\Omega}}_k^T \odot \mathbf{X}^T \right) \left( \mathbf{X} \odot \bar{\boldsymbol{\Omega}}_k \right) \hat{\mathbf{B}}^t \right)^{-1} \left( \frac{1}{N} [\hat{\mathbf{B}}^t]^T \left( \bar{\boldsymbol{\Omega}}_k^T \odot \mathbf{X}^T \right) \left( \mathbf{Y} \odot \boldsymbol{\Omega}_k \right) \right), \tag{55}$$

$$\mathbf{B}^{t+1} = \hat{\mathbf{B}}^t - \sum_{k=1}^{K} \frac{\eta}{N} \left( \left( \bar{\boldsymbol{\Omega}}_k^T \odot \mathbf{X}^T \right) \left( \mathbf{X} \odot \bar{\boldsymbol{\Omega}}_k \right) \hat{\mathbf{B}} \boldsymbol{\theta}_k - \left( \bar{\boldsymbol{\Omega}}_k^T \odot \mathbf{X}^T \right) \left( \mathbf{Y} \odot \boldsymbol{\Omega}_k \right) \right) \boldsymbol{\theta}_k^T, \tag{56}$$

$$\hat{\mathbf{B}}^{t+1} = \mathbf{B}^{t+1} \left( \mathbf{R}^{t+1} \right)^{-1}, \tag{57}$$

*where* $\hat{\mathbf{B}}^{t+1} \mathbf{R}^{t+1}$ *is the QR factorization of* $\mathbf{B}^{t+1}$. Then the $\hat{\mathbf{B}}$ will be orthogonal matrix at the end of each optimization step. From Lemma B.3 we know that $\left( \frac{1}{N} [\hat{\mathbf{B}}^t]^T \left( \bar{\boldsymbol{\Omega}}_k^T \odot \mathbf{X}^T \right) \left( \mathbf{X} \odot \bar{\boldsymbol{\Omega}}_k \right) \hat{\mathbf{B}}^t \right)$ is invertible with high probability, which indicates the feasibility of the above optimization steps. □

**Lemma B.5.** *The optimization step of* $\boldsymbol{\theta}_k$ *can be expressed by the following equation*

$$\boldsymbol{\theta}_k = \hat{\mathbf{B}}^T \hat{\mathbf{B}}^* \boldsymbol{\theta}_k^* + \mathbf{F}_k + \mathbf{G}_k, \tag{58}$$

*where* $\mathbf{F}_k, \mathbf{G}_k \in \mathbb{R}^c$ *are given in Equation* (64) *and* (65).

*Proof.* From Lemma B.4 we have

$$\boldsymbol{\theta}_k = \left(\frac{1}{N}\hat{\mathbf{B}}^T\left(\bar{\boldsymbol{\Omega}}_k^T \odot \mathbf{X}^T\right)\left(\mathbf{X} \odot \bar{\boldsymbol{\Omega}}_k\right)\hat{\mathbf{B}}\right)^{-1}\left(\frac{1}{N}\hat{\mathbf{B}}^T\left(\bar{\boldsymbol{\Omega}}_k^T \odot \mathbf{X}^T\right)\left(\mathbf{Y} \odot \boldsymbol{\Omega}_k\right)\right). \quad (59)$$

Based on the fact that $\mathbf{Y} = \sum_{k=1}^{K} \boldsymbol{\Omega}_k \odot \left(\mathbf{X}\hat{B}^*\boldsymbol{\theta}_k^* + \mathbf{Z}\right), \boldsymbol{\Omega}_k \odot \boldsymbol{\Omega}_k = \boldsymbol{\Omega}_k, \boldsymbol{\Omega}_k \odot \boldsymbol{\Omega}_{k'} = \mathbf{0}$ we have

$$\boldsymbol{\theta}_k = \left(\frac{1}{N}\hat{\mathbf{B}}^T\left(\bar{\boldsymbol{\Omega}}_k^T \odot \mathbf{X}^T\right)\left(\mathbf{X} \odot \bar{\boldsymbol{\Omega}}_k\right)\hat{\mathbf{B}}\right)^{-1}\left(\frac{1}{N}\hat{\mathbf{B}}^T\left(\bar{\boldsymbol{\Omega}}_k^T \odot \mathbf{X}^T\right)\left(\mathbf{Y} \odot \boldsymbol{\Omega}_k\right)\right), \quad (60)$$

$$= \left(\frac{1}{N}\hat{\mathbf{B}}^T\left(\bar{\boldsymbol{\Omega}}_k^T \odot \mathbf{X}^T\right)\left(\mathbf{X} \odot \bar{\boldsymbol{\Omega}}_k\right)\hat{\mathbf{B}}\right)^{-1}\left(\frac{1}{N}\hat{\mathbf{B}}^T\left(\bar{\boldsymbol{\Omega}}_k^T \odot \mathbf{X}^T\right)\left(\sum_{n=1}^{K} \boldsymbol{\Omega}_k \odot \boldsymbol{\Omega}_n \odot \left(\mathbf{X}\hat{\mathbf{B}}^*\boldsymbol{\theta}_k^* + \mathbf{Z}\right)\right)\right), \quad (61)$$

$$= \left(\frac{1}{N}\hat{\mathbf{B}}^T\left(\bar{\boldsymbol{\Omega}}_k^T \odot \mathbf{X}^T\right)\left(\mathbf{X} \odot \bar{\boldsymbol{\Omega}}_k\right)\hat{\mathbf{B}}\right)^{-1}\left(\frac{1}{N}\left(\hat{\mathbf{B}}^T\left(\bar{\boldsymbol{\Omega}}_k^T \odot \mathbf{X}^T\right)\left(\mathbf{X} \odot \bar{\boldsymbol{\Omega}}_k\right)\hat{\mathbf{B}}^*\boldsymbol{\theta}_k^*\right)\right)$$
$$+ \left(\frac{1}{N}\hat{\mathbf{B}}^T\left(\bar{\boldsymbol{\Omega}}_k^T \odot \mathbf{X}^T\right)\left(\mathbf{X} \odot \bar{\boldsymbol{\Omega}}_k\right)\hat{\mathbf{B}}\right)^{-1}\left(\frac{1}{N}\left(\hat{\mathbf{B}}^T\left(\bar{\boldsymbol{\Omega}}_k^T \odot \mathbf{X}^T\right)\left(\mathbf{Z} \odot \boldsymbol{\Omega}_k\right)\right)\right), \quad (62)$$

$$= \hat{\mathbf{B}}^T\hat{\mathbf{B}}^*\boldsymbol{\theta}_k^* + \left(\frac{1}{N}\hat{\mathbf{B}}^T\left(\bar{\boldsymbol{\Omega}}_k^T \odot \mathbf{X}^T\right)\left(\mathbf{X} \odot \bar{\boldsymbol{\Omega}}_k\right)\hat{\mathbf{B}}\right)^{-1}\left(\frac{1}{N}\left(\hat{\mathbf{B}}^T\left(\bar{\boldsymbol{\Omega}}_k^T \odot \mathbf{X}^T\right)\left(\mathbf{X} \odot \bar{\boldsymbol{\Omega}}_k\right)\hat{\mathbf{B}}^*\boldsymbol{\theta}_k^*\right)\right)$$
$$- \left(\frac{1}{N}\hat{\mathbf{B}}^T\left(\bar{\boldsymbol{\Omega}}_k^T \odot \mathbf{X}^T\right)\left(\mathbf{X} \odot \bar{\boldsymbol{\Omega}}_k\right)\hat{\mathbf{B}}\right)^{-1}\left(\frac{1}{N}\left(\hat{\mathbf{B}}^T\left(\bar{\boldsymbol{\Omega}}_k^T \odot \mathbf{X}^T\right)\left(\mathbf{X} \odot \bar{\boldsymbol{\Omega}}_k\right)\hat{\mathbf{B}}\hat{\mathbf{B}}^T\hat{\mathbf{B}}^*\boldsymbol{\theta}_k^*\right)\right)$$
$$+ \left(\frac{1}{N}\hat{\mathbf{B}}^T\left(\bar{\boldsymbol{\Omega}}_k^T \odot \mathbf{X}^T\right)\left(\mathbf{X} \odot \bar{\boldsymbol{\Omega}}_k\right)\hat{\mathbf{B}}\right)^{-1}\left(\frac{1}{N}\left(\hat{\mathbf{B}}^T\left(\bar{\boldsymbol{\Omega}}_k^T \odot \mathbf{X}^T\right)\left(\mathbf{Z} \odot \boldsymbol{\Omega}_k\right)\right)\right). \quad (63)$$

Then define

$$\mathbf{F}_k = \left(\frac{1}{N}\hat{\mathbf{B}}^T\left(\bar{\boldsymbol{\Omega}}_k^T \odot \mathbf{X}^T\right)\left(\mathbf{X} \odot \bar{\boldsymbol{\Omega}}_k\right)\hat{\mathbf{B}}\right)^{-1}\left(\frac{1}{N}\left(\hat{\mathbf{B}}^T\left(\bar{\boldsymbol{\Omega}}_k^T \odot \mathbf{X}^T\right)\left(\mathbf{X} \odot \bar{\boldsymbol{\Omega}}_k\right)\hat{\mathbf{B}}^*\boldsymbol{\theta}_k^*\right)\right)$$
$$- \left(\frac{1}{N}\hat{\mathbf{B}}^T\left(\bar{\boldsymbol{\Omega}}_k^T \odot \mathbf{X}^T\right)\left(\mathbf{X} \odot \bar{\boldsymbol{\Omega}}_k\right)\hat{\mathbf{B}}\right)^{-1}\left(\frac{1}{N}\left(\hat{\mathbf{B}}^T\left(\bar{\boldsymbol{\Omega}}_k^T \odot \mathbf{X}^T\right)\left(\mathbf{X} \odot \bar{\boldsymbol{\Omega}}_k\right)\hat{\mathbf{B}}\hat{\mathbf{B}}^T\hat{\mathbf{B}}^*\boldsymbol{\theta}_k^*\right)\right), \quad (64)$$

$$\mathbf{G}_k = \left(\frac{1}{N}\hat{\mathbf{B}}^T\left(\bar{\boldsymbol{\Omega}}_k^T \odot \mathbf{X}^T\right)\left(\mathbf{X} \odot \bar{\boldsymbol{\Omega}}_k\right)\hat{\mathbf{B}}\right)^{-1}\left(\frac{1}{N}\left(\hat{\mathbf{B}}^T\left(\bar{\boldsymbol{\Omega}}_k^T \odot \mathbf{X}^T\right)\left(\mathbf{Z} \odot \boldsymbol{\Omega}_k\right)\right)\right). \quad (65)$$

$\square$

**Corollary B.6.** *Define* $\boldsymbol{\Theta}, \mathbf{F}, \mathbf{G}$ *by the matrices that each row is* $\boldsymbol{\theta}_k, \mathbf{F}_k, \mathbf{G}_k$, *respectively, we have*

$$\boldsymbol{\Theta}^{t+1} = \boldsymbol{\Theta}^*[\hat{\mathbf{B}}^*]^T\hat{\mathbf{B}}^t + \mathbf{F}^t + \mathbf{G}^t. \quad (66)$$

*Proof.* We can easily obtain this result by Lemma B.5. $\square$

**Lemma B.7.** *Define*

$$\mathbf{H}_k = \left(\frac{1}{\sqrt{N}}\hat{\mathbf{B}}^T\left(\bar{\boldsymbol{\Omega}}_k^T \odot \mathbf{X}^T\right)\right)\frac{1}{\sqrt{N}}\left(\mathbf{X} \odot \bar{\boldsymbol{\Omega}}_k\right)\left(\mathbf{I}_d - \hat{\mathbf{B}}\hat{\mathbf{B}}^T\right)\hat{\mathbf{B}}^* \quad (67)$$

, $\hat{N}_k = \boldsymbol{\Omega}_k^T\boldsymbol{\Omega}_k$, *and* $\delta = \mathcal{C}\frac{c^{3/2}\sqrt{\log(M)}}{\sqrt{\min_k \hat{N}_k}}$ *for constant* $\mathcal{C}$, *we have*

$$\|\mathbf{H}_k\|_2 \le \frac{\delta\hat{N}_k}{\sqrt{c}N}dist\left(\hat{\mathbf{B}}, \hat{\mathbf{B}}^*\right), \quad (68)$$

$$\sum_{k=1}^{K}\|\mathbf{H}_k\boldsymbol{\theta}_k^*\|_2^2 \le \left(\frac{\sum_{k=1}^{K}\hat{N}_k^2}{KN^2}\right)\delta^2\|\boldsymbol{\Theta}^*\|_2^2 dist^2\left(\hat{\mathbf{B}}, \hat{\mathbf{B}}^*\right), \quad (69)$$

*with probability at least* $1 - \exp(-111c^2\log(M))$.

*Proof.* Because we have

$$\mathbf{H}_k = \sum_{i=1}^{M} \sum_{j=1}^{N_i} \left( \frac{\omega_{i,j;k}}{\sqrt{N}} \hat{\mathbf{B}}^T \mathbf{x}_{i,j} \right) \left( \frac{\omega_{i,j;k}}{\sqrt{N}} [\hat{\mathbf{B}}^*]^T \left( \mathbf{I}_d - \hat{\mathbf{B}} \hat{\mathbf{B}}^T \right) \mathbf{x}_{i,j} \right)^T , \tag{70}$$

$$= \sum_{i=1}^{M} \sum_{\omega_{i,j;k}=1}^{N_i} \left( \frac{1}{\sqrt{N}} \hat{\mathbf{B}}^T \mathbf{x}_{i,j} \right) \left( \frac{1}{\sqrt{N}} [\hat{\mathbf{B}}^*]^T \left( \mathbf{I}_d - \hat{\mathbf{B}} \hat{\mathbf{B}}^T \right) \mathbf{x}_{i,j} \right)^T , \tag{71}$$

$$= \left( \frac{1}{\sqrt{N}} \hat{\mathbf{B}}^T \hat{\mathbf{X}}_k^T \right) \frac{1}{\sqrt{N}} \hat{\mathbf{X}}_k \left( \mathbf{I}_d - \hat{\mathbf{B}} \hat{\mathbf{B}}^T \right) \hat{\mathbf{B}}^* , \tag{72}$$

$$= \frac{\hat{N}_k}{N} \left( \frac{1}{\sqrt{\hat{N}_k}} \hat{\mathbf{B}}^T \hat{\mathbf{X}}_k^T \right) \frac{1}{\sqrt{\hat{N}_k}} \hat{\mathbf{X}}_k \left( \mathbf{I}_d - \hat{\mathbf{B}} \hat{\mathbf{B}}^T \right) \hat{\mathbf{B}}^* \tag{73}$$

where $\hat{\mathbf{X}}_k \in \mathbb{R}^{\hat{N}_k \times d}$ with rows the concatenation of $\mathbf{x}_{i,j}$ that $\omega_{i,j;k} = 1$. Here we define $\hat{N}_k = \sum_{i=1}^{M} \sum_{j=1}^{N_i} \mathbf{1}_{\omega_{i,j;k}=1}$. Then directly use Lemma 4 of Collins et al. (2021), and define $\delta = \mathcal{C} \frac{c^{3/2} \sqrt{\log(M)}}{\sqrt{\min_k \hat{N}_k}}$ for constant $\mathcal{C}$, we have

$$\|\mathbf{H}_k\|_2 \leq \frac{\delta \hat{N}_k}{\sqrt{c} N} dist \left( \hat{\mathbf{B}}, \hat{\mathbf{B}}^* \right) . \tag{74}$$

with probability at least $1 - \exp(-111 c^2 \log(M))$. Then we have

$$\sum_{k=1}^{K} \|\mathbf{H}_k \boldsymbol{\theta}_k^*\|_2^2 \leq \sum_{k=1}^{K} \|\mathbf{H}_k\|_2^2 \|\boldsymbol{\theta}_k^*\|_2^2 , \tag{75}$$

$$\leq \frac{c}{K} \|\boldsymbol{\Theta}^*\|_2^2 \sum_{k=1}^{K} \|\mathbf{H}_k\|_2^2 , \tag{76}$$

$$\leq \left( \frac{\sum_{k=1}^{K} \hat{N}_k^2}{K N^2} \right) \delta^2 \|\boldsymbol{\Theta}^*\|_2^2 dist^2 \left( \hat{\mathbf{B}}, \hat{\mathbf{B}}^* \right) . \tag{77}$$

Then the proof finished. $\qquad \square$

**Lemma B.8.** *Given $\mathbf{F}$ defined in Corollary B.6, define $\delta = \mathcal{C} \frac{c^{3/2} \sqrt{\log(M)}}{\sqrt{\min_k \hat{N}_k}}$, we have*

$$\|\mathbf{F}_k\|_2 \leq \frac{\delta}{(1-\delta)\sqrt{c}} \|\boldsymbol{\theta}_k^*\|_2 dist \left( \hat{\mathbf{B}}, \hat{\mathbf{B}}^* \right) , \tag{78}$$

$$\|\mathbf{F}\|_F \leq \frac{\delta}{(1-\delta)\sqrt{K}} \|\boldsymbol{\Theta}^*\|_2 dist \left( \hat{\mathbf{B}}, \hat{\mathbf{B}}^* \right) , \tag{79}$$

*with probability at least $1 - \exp(-111 c^2 \log(M))$.*

*Proof.* From Lemma B.3, we have

$$\left\| \left( \frac{1}{N} \hat{\mathbf{B}}^T \left( \bar{\boldsymbol{\Omega}}_k^T \odot \mathbf{X}^T \right) \left( \mathbf{X} \odot \bar{\boldsymbol{\Omega}}_k \right) \hat{\mathbf{B}} \right)^{-1} \right\|_2^2 \leq \frac{N^2}{\hat{N}_k^2 (1-\delta)^2} . \tag{80}$$

Then we consider to bound $\mathbf{F}_k$ first, by Lemma B.7 we have

$$\|\mathbf{F}_k\|_2^2 \leq \left\| \left( \frac{1}{N} \hat{\mathbf{B}}^T \left( \bar{\boldsymbol{\Omega}}_k^T \odot \mathbf{X}^T \right) \left( \mathbf{X} \odot \bar{\boldsymbol{\Omega}}_k \right) \hat{\mathbf{B}} \right)^{-1} \right\|_2^2 \|\mathbf{H}_k\|_2^2 \|\boldsymbol{\theta}_k^*\|_2^2 , \tag{81}$$

$$\leq \frac{1}{(1-\delta)^2} \frac{\delta^2}{c} dist^2 \left( \hat{\mathbf{B}}, \hat{\mathbf{B}}^* \right) \|\boldsymbol{\theta}_k^*\|_2^2 , \tag{82}$$

for $\delta = \mathcal{C}\frac{c^{3/2}\sqrt{\log(M)}}{\sqrt{\min_k \hat{N}_k}}$ with probability at least $1 - \exp(-111c^2 \log(M))$. Then we consider to bound $\mathbf{F}$, and we have

$$\|\mathbf{F}\|_F^2 = \sum_{k=1}^{K} \|\mathbf{F}_i\|_2^2 \, , \tag{83}$$

$$\leq \sum_{k=1}^{K} \left\| \left( \frac{1}{N}\hat{\mathbf{B}}^T \left(\bar{\mathbf{\Omega}}_k^T \odot \mathbf{X}^T\right) \left(\mathbf{X} \odot \bar{\mathbf{\Omega}}_k\right) \hat{\mathbf{B}} \right)^{-1} \right\|_2^2 \|\mathbf{H}_k \boldsymbol{\theta}_k\|_2^2 \, , \tag{84}$$

$$\leq \frac{1}{(1-\delta)^2} \sum_{k=1}^{K} \frac{N^2}{\hat{N}_k^2} \|\mathbf{H}_k \boldsymbol{\theta}_k\|_2^2 \, , \tag{85}$$

$$\leq \frac{c}{(1-\delta)^2 K} \|\mathbf{\Theta}^*\|_2^2 \sum_{k=1}^{K} \frac{N^2}{\hat{N}_k^2} \|\mathbf{H}_k\|_2^2 \, , \tag{86}$$

$$\leq \frac{\delta^2}{(1-\delta)^2 K} \|\mathbf{\Theta}^*\|_2^2 \, dist^2 \left(\hat{\mathbf{B}}, \hat{\mathbf{B}}^*\right) \tag{87}$$

with probability at least $1 - \exp(-111c^2 \log(M))$. The last equation comes from Lemma B.7. $\quad\square$

**Lemma B.9.** *Given $\mathbf{G}_k$ defined by*

$$\mathbf{G}_k = \left( \frac{1}{N}\hat{\mathbf{B}}^T \left(\bar{\mathbf{\Omega}}_k^T \odot \mathbf{X}^T\right) \left(\mathbf{X} \odot \bar{\mathbf{\Omega}}_k\right) \hat{\mathbf{B}} \right)^{-1} \left( \frac{1}{N}\left( \hat{\mathbf{B}}^T \left(\bar{\mathbf{\Omega}}_k^T \odot \mathbf{X}^T\right) \left(\mathbf{Z} \odot \mathbf{\Omega}_k\right) \right) \right) \tag{88}$$

*and $\mathbf{G}$ defined in Corollary B.6. Define $\delta = \mathcal{C}\frac{c^{3/2}\sqrt{\log(M)}}{\sqrt{\min_k \hat{N}_k}}$, we have*

$$\|\mathbf{G}_k\|_2 \leq \frac{\delta}{1-\delta}\sigma^2 \, , \tag{89}$$

$$\|\mathbf{G}\|_F \leq \sqrt{K}\frac{\delta}{(1-\delta)}\sigma^2 \, , \tag{90}$$

*with probability at least $1 - \exp(-110c^2 \log(M))$.*

*Proof.* We can rewrite $\mathbf{G}_k$ by

$$\mathbf{G}_k = \left( \frac{1}{N}\hat{\mathbf{B}}^T \left(\bar{\mathbf{\Omega}}_k^T \odot \mathbf{X}^T\right) \left(\mathbf{X} \odot \bar{\mathbf{\Omega}}_k\right) \hat{\mathbf{B}} \right)^{-1} \frac{1}{N} \sum_{i=1}^{M} \sum_{j=1}^{N_i} \omega_{i,j;k} z_k \hat{\mathbf{B}}^T \mathbf{x}_{i,j} \, , \tag{91}$$

$$= \left( \frac{1}{N}\hat{\mathbf{B}}^T \left(\bar{\mathbf{\Omega}}_k^T \odot \mathbf{X}^T\right) \left(\mathbf{X} \odot \bar{\mathbf{\Omega}}_k\right) \hat{\mathbf{B}} \right)^{-1} \frac{1}{N} \sum_{\omega_{i,j;k}=1}^{N} z_k \hat{\mathbf{B}}^T \mathbf{x}_{i,j} \, , \tag{92}$$

$$= \left( \frac{1}{N}\hat{\mathbf{B}}^T \left(\bar{\mathbf{\Omega}}_k^T \odot \mathbf{X}^T\right) \left(\mathbf{X} \odot \bar{\mathbf{\Omega}}_k\right) \hat{\mathbf{B}} \right)^{-1} \frac{1}{N} \left( \hat{\mathbf{B}}^T \hat{\mathbf{X}}_k^T \hat{\mathbf{Z}}_k \right) \, . \tag{93}$$

where $\hat{\mathbf{X}}_k \in \mathbb{R}^{\hat{N}_k \times d}$ with rows the concatenation of $\mathbf{x}_{i,j}$ that $\omega_{i,j;k} = 1$. Here we define $\hat{N}_k = \sum_{i=1}^{M} \sum_{j=1}^{N_i} \mathbf{1}_{\omega_{i,j;k}=1}$. Then directly use Lemma A.7 of Tziotis et al. (2022), and define $\delta = \mathcal{C}\frac{c^{3/2}\sqrt{\log(M)}}{\sqrt{\min_k \hat{N}_k}}$, we have

$$\left\| \frac{1}{\hat{N}_k}\hat{\mathbf{B}}^T \hat{\mathbf{X}}_k^T \hat{\mathbf{Z}}_k \right\|_2 \leq \sigma^2 \delta \, . \tag{94}$$

with probability at least $1 - \exp(-113c^2 \log(M))$. Then consider $\mathbf{G}_k$, we have

$$\|\mathbf{G}_k\|_2 \leq \frac{\hat{N}_k}{N} \left\| \left( \frac{1}{N}\hat{\mathbf{B}}^T \left(\bar{\mathbf{\Omega}}_k^T \odot \mathbf{X}^T\right) \left(\mathbf{X} \odot \bar{\mathbf{\Omega}}_k\right) \hat{\mathbf{B}} \right)^{-1} \right\| \left\| \frac{1}{\hat{N}_k}\hat{\mathbf{B}}^T \hat{\mathbf{X}}_k^T \hat{\mathbf{Z}}_k \right\|_2 \, , \tag{95}$$

$$\leq \frac{\delta}{1-\delta}\sigma^2 \, . \tag{96}$$

Then consider $\mathbf{G}$, we have

$$\|\mathbf{G}\|_F^2 = \sum_{k=1}^{K} \|\mathbf{G}_k\|_2^2 \leq K \left( \frac{\delta}{(1-\delta)} \right)^2 \sigma^4 \,. \tag{97}$$

$\square$

**Lemma B.10.** *Define $\delta = \mathcal{C} \frac{c^{3/2} \sqrt{\log(M)}}{\sqrt{\min_k \hat{N}_k}}$, we have*

$$\|\boldsymbol{\theta}_k\|_2 \leq \sqrt{c} + \frac{\delta}{1-\delta} dist\left(\hat{\mathbf{B}}, \hat{\mathbf{B}}^*\right) + \frac{\delta}{1-\delta}\sigma^2 \,, \tag{98}$$

$$\left\|\hat{\mathbf{B}}\boldsymbol{\theta}_k - \hat{\mathbf{B}}^*\boldsymbol{\theta}_k^*\right\|_2 \leq \left(\sqrt{c} + \frac{\delta}{1-\delta}\right) dist\left(\hat{\mathbf{B}}, \hat{\mathbf{B}}^*\right) + \frac{\delta}{1-\delta}\sigma^2 \,, \tag{99}$$

$$\left\|\frac{1}{N} \sum_{k=1}^{K} \left(\bar{\boldsymbol{\Omega}}_k^T \odot \mathbf{X}^T\right)\left(\mathbf{Z} \odot \boldsymbol{\Omega}_k\right)\boldsymbol{\theta}_k^T\right\|_2 \leq \mathcal{C}_1 \sigma^2 \frac{\sqrt{d+c}}{\sqrt{N}} \left(\sqrt{c} + \frac{\delta}{1-\delta} dist\left(\hat{\mathbf{B}}, \hat{\mathbf{B}}^*\right) + \frac{\delta}{1-\delta}\sigma^2\right) \,, \tag{100}$$

*with probability at least $1 - \exp(-105(d+c)) - \exp(-105c^2 \log(M))$ for some constant $\mathcal{C}_1$.*

*Proof.* Define

$$\mathbf{q}_k = \left(\bar{\boldsymbol{\Omega}}_k^T \odot \mathbf{X}^T\right)\left(\mathbf{X} \odot \bar{\boldsymbol{\Omega}}_k\right)\hat{\mathbf{B}}\boldsymbol{\theta}_k - \left(\bar{\boldsymbol{\Omega}}_k^T \odot \mathbf{X}^T\right)\left(\mathbf{Y} \odot \boldsymbol{\Omega}_k\right) \,, \tag{101}$$

and $\mathbf{Q} \in \mathbb{R}^{d \times K}$ with rows the concatenation of $\mathbf{q}_k$. With the fact that $\mathbf{Y} = \sum_{k=1}^{K} \boldsymbol{\Omega}_k \odot \left(\mathbf{X}\hat{\mathbf{B}}^*\boldsymbol{\theta}_k^* + \mathbf{Z}\right)$, we have

$$\mathbf{q}_k = \left(\bar{\boldsymbol{\Omega}}_k^T \odot \mathbf{X}^T\right)\left(\mathbf{X} \odot \bar{\boldsymbol{\Omega}}_k\right)\hat{\mathbf{B}}\boldsymbol{\theta}_k - \left(\bar{\boldsymbol{\Omega}}_k^T \odot \mathbf{X}^T\right)\left(\mathbf{Y} \odot \boldsymbol{\Omega}_k\right) \,, \tag{102}$$

$$= \left(\bar{\boldsymbol{\Omega}}_k^T \odot \mathbf{X}^T\right)\left(\mathbf{X} \odot \bar{\boldsymbol{\Omega}}_k\right)\hat{\mathbf{B}}\boldsymbol{\theta}_k - \left(\bar{\boldsymbol{\Omega}}_k^T \odot \mathbf{X}^T\right)\left(\boldsymbol{\Omega}_k \odot \sum_{n=1}^{K} \boldsymbol{\Omega}_n \odot \left(\mathbf{X}\hat{\mathbf{B}}^*\boldsymbol{\theta}_n^* + \mathbf{Z}\right)\right) \,, \tag{103}$$

$$= \left(\bar{\boldsymbol{\Omega}}_k^T \odot \mathbf{X}^T\right)\left(\mathbf{X} \odot \bar{\boldsymbol{\Omega}}_k\right)\hat{\mathbf{B}}\boldsymbol{\theta}_k - \left(\bar{\boldsymbol{\Omega}}_k^T \odot \mathbf{X}^T\right)\left(\boldsymbol{\Omega}_k \odot \left(\mathbf{X}\hat{\mathbf{B}}^*\boldsymbol{\theta}_k^* + \mathbf{Z}\right)\right) \,, \tag{104}$$

$$= \left(\bar{\boldsymbol{\Omega}}_k^T \odot \mathbf{X}^T\right)\left(\mathbf{X} \odot \bar{\boldsymbol{\Omega}}_k\right)\hat{\mathbf{B}}\boldsymbol{\theta}_k - \left(\bar{\boldsymbol{\Omega}}_k^T \odot \mathbf{X}^T\right)\left(\mathbf{X} \odot \bar{\boldsymbol{\Omega}}_k\right)\hat{\mathbf{B}}^*\boldsymbol{\theta}_k^* - \left(\bar{\boldsymbol{\Omega}}_k^T \odot \mathbf{X}^T\right)\left(\mathbf{Z} \odot \boldsymbol{\Omega}_k\right) \,, \tag{105}$$

$$= \left(\bar{\boldsymbol{\Omega}}_k^T \odot \mathbf{X}^T\right)\left(\mathbf{X} \odot \bar{\boldsymbol{\Omega}}_k\right)\left(\hat{\mathbf{B}}\boldsymbol{\theta}_k - \hat{\mathbf{B}}^*\boldsymbol{\theta}_k^*\right) - \left(\bar{\boldsymbol{\Omega}}_k^T \odot \mathbf{X}^T\right)\left(\mathbf{Z} \odot \boldsymbol{\Omega}_k\right) \,. \tag{106}$$

Then we would like to consider the $\left(\hat{\mathbf{B}}\boldsymbol{\theta}_k - \hat{\mathbf{B}}^*\boldsymbol{\theta}_k^*\right)$ first. From Lemma B.5, we have

$$\boldsymbol{\theta}_k = \hat{\mathbf{B}}^T\hat{\mathbf{B}}^*\boldsymbol{\theta}_k^* + \mathbf{F}_k + \mathbf{G}_k \,. \tag{107}$$

Therefore we have

$$\left\|\hat{\mathbf{B}}\boldsymbol{\theta}_k - \hat{\mathbf{B}}^*\boldsymbol{\theta}_k^*\right\|_2 = \left\|\hat{\mathbf{B}}\hat{\mathbf{B}}^T\hat{\mathbf{B}}^*\boldsymbol{\theta}_k^* + \hat{\mathbf{B}}\mathbf{F}_k + \hat{\mathbf{B}}\mathbf{G}_k - \hat{\mathbf{B}}^*\boldsymbol{\theta}_k^*\right\|_2 \,, \tag{108}$$

$$\leq \left\|\left(\hat{\mathbf{B}}\hat{\mathbf{B}}^T - \mathbf{I}_d\right)\hat{\mathbf{B}}^*\boldsymbol{\theta}_k^*\right\|_2 + \left\|\hat{\mathbf{B}}\mathbf{F}_k\right\|_2 + \left\|\hat{\mathbf{B}}\mathbf{G}_k\right\|_2 \,, \tag{109}$$

$$\leq dist\left(\hat{\mathbf{B}}, \hat{\mathbf{B}}^*\right)\|\boldsymbol{\theta}_k^*\|_2 + \|\mathbf{F}_k\|_2 + \|\mathbf{G}_k\|_2 \,, \tag{110}$$

$$\leq \sqrt{c}\, dist\left(\hat{\mathbf{B}}, \hat{\mathbf{B}}^*\right) + \frac{\delta}{(1-\delta)\sqrt{c}}\|\boldsymbol{\theta}_k^*\|_2\, dist\left(\hat{\mathbf{B}}, \hat{\mathbf{B}}^*\right) + \frac{\delta}{1-\delta}\sigma^2 \,, \tag{111}$$

$$= \left(\sqrt{c} + \frac{\delta}{1-\delta}\right) dist\left(\hat{\mathbf{B}}, \hat{\mathbf{B}}^*\right) + \frac{\delta}{1-\delta}\sigma^2 \,, \tag{112}$$

with probability at least $1 - \exp(-110c^2 \log(M))$, and $\delta = \mathcal{C} \frac{c^{3/2}\sqrt{\log(M)}}{\sqrt{\min_k \hat{N}_k}}$.

Then we consider to bound $\boldsymbol{\theta}_k$, and we have

$$\|\boldsymbol{\theta}_k\|_2 = \left\|\hat{\mathbf{B}}^T\hat{\mathbf{B}}^*\boldsymbol{\theta}_k^* + \mathbf{F}_k + \mathbf{G}_k\right\|_2, \tag{113}$$

$$\leq \|\boldsymbol{\theta}_k^*\|_2 + \|F_k\|_2 + \|\mathbf{G}_k\|_2, \tag{114}$$

$$\leq \sqrt{c} + \frac{\delta}{1-\delta}dist\left(\hat{\mathbf{B}}, \hat{\mathbf{B}}^*\right) + \frac{\delta}{1-\delta}\sigma^2, \tag{115}$$

with probability at least $1 - \exp(-110c^2\log(M))$.

Then we consider to bound $\left(\bar{\boldsymbol{\Omega}}_k^T \odot \mathbf{X}^T\right)\left(\mathbf{Z} \odot \boldsymbol{\Omega}_k\right)$, and we have

$$\left(\bar{\boldsymbol{\Omega}}_k^T \odot \mathbf{X}^T\right)\left(\mathbf{Z} \odot \boldsymbol{\Omega}_k\right) = \sum_{\substack{N \\ \omega_{i,j;k}=1}} z_k\mathbf{x}_{i,j}, \tag{116}$$

$$= \hat{\mathbf{X}}_k^T\hat{\mathbf{Z}}_k, \tag{117}$$

where rows of $\hat{\mathbf{X}}_k \in \mathbb{R}^{\hat{N}_k \times d}$ and $\hat{\mathbf{Z}}_k \in \mathbb{R}^{\hat{N}_k}$ are $\mathbf{x}_{i,j}$ and $z_k$ subject to $\omega_{i,j;k} = 1$. Then let $\mathcal{S}^{d-1}, \mathcal{S}^{c-1}$ denote the unit spheres in $d$ and $c$ dimensions, and $\mathcal{N}_d, \mathcal{N}_k$ denote the $\frac{1}{4}$-nets of cardinality $9^d$ and $9^k$, respectively. Then by Equation 4.13 of Vershynin (2018), we have

$$\left\|\frac{1}{N}\sum_{k=1}^K \hat{\mathbf{X}}_k^T\hat{\mathbf{Z}}_k\boldsymbol{\theta}_k^T\right\|_2 \leq 2 \max_{\mathbf{p}\in\mathcal{N}_d, \mathbf{y}\in\mathcal{N}_k} \mathbf{p}^T\left(\frac{1}{N}\sum_{k=1}^K \hat{\mathbf{X}}_k^T\hat{\mathbf{Z}}_k\boldsymbol{\theta}_k^T\right)\mathbf{y}, \tag{118}$$

$$= 2 \max_{\mathbf{p}\in\mathcal{N}_d, \mathbf{y}\in\mathcal{N}_k} \sum_{k=1}^K \sum_{i,j}^{\hat{N}_k} \left(\frac{z_k}{N}\langle\mathbf{x}_{i,j}, \mathbf{p}\rangle\langle\boldsymbol{\theta}_k, \mathbf{y}\rangle\right). \tag{119}$$

Notice that for any fixed $\mathbf{p}, \mathbf{y}$, the random variables $\frac{z_k}{N}\langle\mathbf{x}_{i,j}, \mathbf{p}\rangle\langle\boldsymbol{\theta}_k, \mathbf{y}\rangle$ are i.i.d. zero-mean sub-exponentials with the norm at most $\mathcal{C}_1\frac{\sigma^2\|\boldsymbol{\theta}_k\|}{N}$ for some constant $\mathcal{C}$. Then consider the event

$$\mathcal{E} = \bigcap_{k=1}^K \left\{\|\boldsymbol{\theta}_k\|_2 \leq \sqrt{c} + \frac{\delta}{1-\delta}dist\left(\hat{\mathbf{B}}, \hat{\mathbf{B}}^*\right) + \frac{\delta}{1-\delta}\sigma^2\right\}, \tag{120}$$

which holds with probability at least $1 - \exp(-105c^2\log(M))$. Then use the Bernstein's inequality we have

$$\Pr\left(\sum_{k=1}^K \sum_{i,j}^{\hat{N}_k} \frac{z_k}{N}\langle\mathbf{x}_{i,j}, \mathbf{p}\rangle\langle\boldsymbol{\theta}_k, \mathbf{y}\rangle \geq s \mid \mathcal{E}\right)$$

$$\leq \exp\left(-\mathcal{C}_1'N\min\left\{\frac{s^2}{\sigma^4\left(\sqrt{c} + \frac{\delta}{1-\delta}dist\left(\hat{\mathbf{B}}, \hat{\mathbf{B}}^*\right) + \frac{\delta}{1-\delta}\sigma^2\right)^2}, \frac{s}{\sigma^2\left(\sqrt{c} + \frac{\delta}{1-\delta}dist\left(\hat{\mathbf{B}}, \hat{\mathbf{B}}^*\right) + \frac{\delta}{1-\delta}\sigma^2\right)}\right\}\right). \tag{121}$$

Setting

$$s = \mathcal{C}_2\frac{\sigma^2\sqrt{d+c}\left(\sqrt{c} + \frac{\delta}{1-\delta}dist\left(\hat{\mathbf{B}}, \hat{\mathbf{B}}^*\right) + \frac{\delta}{1-\delta}\sigma^2\right)}{\sqrt{N}}, \tag{122}$$

we have

$$\Pr\left(\sum_{k=1}^K \sum_{i,j}^{\hat{N}_k} \frac{z_k}{N}\langle\mathbf{x}_{i,j}, \mathbf{p}\rangle\langle\boldsymbol{\theta}_k, \mathbf{y}\rangle \geq \mathcal{C}_2\sigma^2\frac{\sqrt{d+c}}{\sqrt{N}}\left(\sqrt{c} + \frac{\delta}{1-\delta}dist\left(\hat{\mathbf{B}}, \hat{\mathbf{B}}^*\right) + \frac{\delta}{1-\delta}\sigma^2\right) \mid \mathcal{E}\right)$$

$$\leq \exp(-\mathcal{C}_1'\mathcal{C}_2d) \leq exp(-110(d+c)), \tag{123}$$

for $\mathcal{C}_2$ large enough. Taking the union bound over all points $\mathbf{p}, \mathbf{y}$ on the $\mathcal{N}_d, \mathcal{N}_k$, we have

$$\Pr\left(\left\|\frac{1}{N}\sum_{k=1}^K \hat{\mathbf{X}}_k^T\hat{\mathbf{Z}}_k\boldsymbol{\theta}_k^T\right\|_2 \geq 2\mathcal{C}_2\sigma^2\frac{\sqrt{d+c}}{\sqrt{N}}\left(\sqrt{c} + \frac{\delta}{1-\delta}dist\left(\hat{\mathbf{B}}, \hat{\mathbf{B}}^*\right) + \frac{\delta}{1-\delta}\sigma^2\right) \mid \mathcal{E}\right)$$

$$\leq 9^{d+c}\exp(-110(d+c)) \leq \exp(-105(d+c)). \tag{124}$$

Removing the conditional on $\mathcal{E}$, we have

$$\Pr\left(\left\|\frac{1}{N}\sum_{k=1}^{K}\hat{\mathbf{X}}_k^T\hat{\mathbf{Z}}_k\boldsymbol{\theta}_k^T\right\|_2 \geq 2\mathcal{C}_2\sigma^2\frac{\sqrt{d+c}}{\sqrt{N}}\left(\sqrt{c}+\frac{\delta}{1-\delta}dist\left(\hat{\mathbf{B}},\hat{\mathbf{B}}^*\right)+\frac{\delta}{1-\delta}\sigma^2\right)\right)$$

$$\leq \exp(-105(d+c))+\Pr(\mathcal{E}^C) \leq \exp(-105(d+c))+\exp(-105c^2\log(M)). \tag{125}$$

$\square$

**Lemma B.11.** *Define* $\delta = \mathcal{C}\dfrac{c^{3/2}\sqrt{\log(M)}}{\sqrt{\min_k \hat{N}_k}}$, *we have*

$$\left\|\sum_{k=1}^{K}\frac{1}{N}\left(\left(\bar{\boldsymbol{\Omega}}_k^T\odot\mathbf{X}^T\right)\left(\mathbf{X}\odot\bar{\boldsymbol{\Omega}}_k\right)\left(\hat{\mathbf{B}}\boldsymbol{\theta}_k-\hat{\mathbf{B}}^*\boldsymbol{\theta}_k^*\right)-\hat{N}_k\left(\hat{\mathbf{B}}\boldsymbol{\theta}_k-\hat{\mathbf{B}}^*\boldsymbol{\theta}_k^*\right)\right)\boldsymbol{\theta}_k^T\right\|_2 \leq \mathcal{C}\frac{\sqrt{d+c}}{\sqrt{N}}\epsilon,$$

$$\tag{126}$$

*where*

$$\epsilon = \left(\sqrt{c}\frac{\delta}{1-\delta}+\frac{\delta^2}{(1-\delta)^2}\right)dist^2\left(\hat{\mathbf{B}},\hat{\mathbf{B}}^*\right)$$

$$+\left(c+(\sigma^2+1)\sqrt{c}\frac{\delta}{1-\delta}+\frac{2\delta^2}{(1-\delta)^2}\sigma^2\right)dist\left(\hat{\mathbf{B}},\hat{\mathbf{B}}^*\right)+\sqrt{c}\frac{\delta}{1-\delta}\sigma^2+\frac{\delta^2}{(1-\delta)^2}\sigma^4, \tag{127}$$

*with probability at least* $1-\exp(-100(d+c))-\exp(-105c^2\log(M))$.

*Proof.* Define

$$\mathbf{q}_k = \frac{1}{N}\left(\bar{\boldsymbol{\Omega}}_k^T\odot\mathbf{X}^T\right)\left(\mathbf{X}\odot\bar{\boldsymbol{\Omega}}_k\right)\left(\hat{\mathbf{B}}\boldsymbol{\theta}_k-\hat{\mathbf{B}}^*\boldsymbol{\theta}_k^*\right)-\frac{\hat{N}_k}{N}\left(\hat{\mathbf{B}}\boldsymbol{\theta}_k-\hat{\mathbf{B}}^*\boldsymbol{\theta}_k^*\right), \tag{128}$$

$$= \frac{1}{N}\hat{X}_k^T\hat{X}_k\left(\hat{\mathbf{B}}\boldsymbol{\theta}_k-\hat{\mathbf{B}}^*\boldsymbol{\theta}_k^*\right)-\frac{\hat{N}_k}{N}\left(\hat{\mathbf{B}}\boldsymbol{\theta}_k-\hat{\mathbf{B}}^*\boldsymbol{\theta}_k^*\right), \tag{129}$$

where rows of $\hat{\mathbf{X}}_k\in\mathbb{R}^{\hat{N}_k\times d}$ are $\mathbf{x}_{i,j}$ subject to $\omega_{i,j;k}=1$. Then we would like to define the event

$$\mathcal{E} = \bigcap_{k=1}^{K}\left\{\mathcal{A}_k\bigcap\mathcal{B}_k\right\}, \tag{130}$$

$$\mathcal{A}_k = \left\{\|\boldsymbol{\theta}_k\|_2 \leq \sqrt{c}+\frac{\delta}{1-\delta}dist\left(\hat{\mathbf{B}},\hat{\mathbf{B}}^*\right)+\frac{\delta}{1-\delta}\sigma^2\right\}, \tag{131}$$

$$\mathcal{B}_k = \left\{\left\|\hat{\mathbf{B}}\boldsymbol{\theta}_k-\hat{\mathbf{B}}^*\boldsymbol{\theta}_k^*\right\|_2 \leq \left(\sqrt{c}+\frac{\delta}{1-\delta}\right)dist\left(\hat{\mathbf{B}},\hat{\mathbf{B}}^*\right)+\frac{\delta}{1-\delta}\sigma^2\right\}, \tag{132}$$

happens with the probability at least $1-\exp(-105(d+c))-\exp(-105c^2\log(M))$ by Lemma B.10. Define $\mathbf{g}_k=\hat{\mathbf{B}}\boldsymbol{\theta}_k-\hat{\mathbf{B}}^*\boldsymbol{\theta}_k^*$, we have

$$\sum_{k=1}^{K}\mathbf{q}_k\boldsymbol{\theta}_k^T = \frac{1}{N}\left(\sum_{k=1}^{K}\sum_{i,j}^{\hat{N}_k}\left(\langle\mathbf{x}_{i,j},\mathbf{g}_k\rangle\mathbf{x}_{i,j}\boldsymbol{\theta}_k^T-\mathbf{g}_k\boldsymbol{\theta}_k^T\right)\right). \tag{133}$$

Let $\mathcal{S}^{d-1}, \mathcal{S}^{c-1}$ denote the unit spheres in $d$ and $c$ dimensions and $\mathcal{N}_d, \mathcal{N}_k$ the $\frac{1}{4}$-nets of cardinality $9^d$ and $9^k$, respectively. By Equation 4.13 in Vershynin (2018), we have

$$\left\|\sum_{k=1}^{K}\mathbf{q}_k\boldsymbol{\theta}_k^T\right\|_2 \leq \frac{2}{N}\max_{\mathbf{p}\in\mathcal{N}_d,\mathbf{y}\in\mathcal{N}_k}\mathbf{p}^T\left(\sum_{k=1}^{K}\sum_{i,j}^{\hat{N}_k}\langle\mathbf{x}_{i,j},\mathbf{g}_k\rangle\mathbf{x}_{i,j}\boldsymbol{\theta}_k^T-\sum_{k=1}^{K}\mathbf{g}_k\boldsymbol{\theta}_k^T\right)\mathbf{y}, \tag{134}$$

$$= \frac{2}{N}\max_{\mathbf{p}\in\mathcal{N}_d,\mathbf{y}\in\mathcal{N}_k}\sum_{k=1}^{K}\sum_{i,j}^{\hat{N}_k}\left(\langle\mathbf{x}_{i,j},\mathbf{g}_k\rangle\langle\mathbf{p},\mathbf{x}_{i,j}\rangle\langle\boldsymbol{\theta}_k,\mathbf{y}\rangle-\langle\mathbf{p},\mathbf{g}_k\rangle\langle\boldsymbol{\theta}_k,\mathbf{y}\rangle\right). \tag{135}$$

Then for any $\mathbf{p}, \mathbf{y}$, the inner products $\langle \mathbf{x}_{i,j}, \mathbf{g}_k \rangle, \langle \mathbf{p}, \mathbf{x}_{i,j} \rangle$ are sub-gaussians with norm at most $\mathcal{C}_1 \|\mathbf{g}_i\|_2$ and $\mathcal{C}_2 \|\mathbf{p}\|_2 = \mathcal{C}_2$, respectively for some constants $\mathcal{C}_1, \mathcal{C}_2$. Then under the condition that $\mathcal{E}$ holds we have $\frac{1}{N} \langle \mathbf{x}_{i,j}, \mathbf{g}_k \rangle \langle \mathbf{p}, \mathbf{x}_{i,j} \rangle \langle \boldsymbol{\theta}_k, \mathbf{y} \rangle$ is sub-exponential with norm at most

$$
\frac{\mathcal{C}_3}{N} \epsilon = \frac{\mathcal{C}_3}{N} \left( \sqrt{c} \frac{\delta}{1-\delta} + \frac{\delta^2}{(1-\delta)^2} \right) dist^2 \left( \hat{\mathbf{B}}, \hat{\mathbf{B}}^* \right)
$$
$$
+ \frac{\mathcal{C}_3}{N} \left( \left( c + (\sigma^2 + 1) \sqrt{c} \frac{\delta}{1-\delta} + \frac{2\delta^2}{(1-\delta)^2} \sigma^2 \right) dist \left( \hat{\mathbf{B}}, \hat{\mathbf{B}}^* \right) + \sqrt{c} \frac{\delta}{1-\delta} \sigma^2 + \frac{\delta^2}{(1-\delta)^2} \sigma^4 \right) . \tag{136}
$$

The same thing can be observed for $\langle \mathbf{p}, \mathbf{g}_k \rangle \langle \boldsymbol{\theta}_k, \mathbf{y} \rangle$. Besides, we can observe that

$$
\mathbb{E} \left[ \langle \mathbf{x}_{i,j}, \mathbf{g}_k \rangle \langle \mathbf{p}, \mathbf{x}_{i,j} \rangle \langle \boldsymbol{\theta}_k, \mathbf{y} \rangle - \langle \mathbf{p}, \mathbf{g}_k \rangle \langle \boldsymbol{\theta}_k, \mathbf{y} \rangle \right] = 0 . \tag{137}
$$

Then we are dealing with $N$ zero-mean, sub-exponential random variables. Using Bernstein's inequality we have

$$
\Pr \left( \frac{1}{N} \sum_{k=1}^{K} \sum_{i,j}^{\hat{N}_k} \langle \mathbf{x}_{i,j}, \mathbf{g}_k \rangle \langle \mathbf{p}, \mathbf{x}_{i,j} \rangle \langle \boldsymbol{\theta}_k, \mathbf{y} \rangle - \langle \mathbf{p}, \mathbf{g}_k \rangle \langle \boldsymbol{\theta}_k, \mathbf{y} \rangle \geq s \mid \mathcal{E} \right) \leq \exp \left( -\mathcal{C}_4 N \min \left\{ \frac{s^2}{\epsilon^2}, \frac{s}{\epsilon} \right\} \right) . \tag{138}
$$

Setting $s = \frac{\sqrt{\mathcal{C}_5 (d+c)} \epsilon}{\sqrt{N}}$, for constant $\mathcal{C}_5$ that satisfy $\mathcal{C}_5 \leq \frac{N}{d+c}$, and taking union bound over all $\mathbf{p}, \mathbf{y}$, we have

$$
\Pr \left( \left\| \sum_{k=1}^{K} \mathbf{q}_k \boldsymbol{\theta}_k^T \right\|_2 \geq \frac{2\sqrt{\mathcal{C}_5 (d+c)} \epsilon}{\sqrt{N}} \mid \mathcal{E} \right) \leq 9^{d+c} \exp \left( -\mathcal{C}_4 \mathcal{C}_5 (d+c) \right) \leq \exp(-105(d+c)) . \tag{139}
$$

Then by removing the conditional on $\mathcal{E}$, we have

$$
\left\| \sum_{k=1}^{K} \mathbf{q}_k \boldsymbol{\theta}_k^T \right\|_2 \leq \mathcal{C} \frac{\sqrt{d+c} \epsilon}{\sqrt{N}} , \tag{140}
$$

with probability at least $1 - \exp(-100(d+c)) - \exp(-105c^2 \log(M))$. $\qquad\square$

## B.3 MAIN RESULTS

**Theorem B.12.** *Under Assumption 1- 4, when we have* $N \geq \frac{K^2}{d+c}$, *and* $\min_k \hat{N}_k \geq \mathcal{C} \frac{c^3 (1+\sigma^2)^4 \log(M)}{E_0^2} \min \left\{ \frac{1}{\kappa^2}, \bar{\sigma}_{\min}^2 \right\}$ *for some constant $\mathcal{C}$, we have*

$$
dist(\hat{\mathbf{B}}^{t+1}, \hat{\mathbf{B}}^*) \leq dist(\hat{\mathbf{B}}^t, \hat{\mathbf{B}}^*) \left( 1 - c_{min} + \frac{57}{200} c_{max} \right) \left( 1 - \frac{1}{2} c_{max} \right)^{-1/2}
$$
$$
+ \left( \frac{7}{100} c_{max} \right) \left( 1 - \frac{1}{2} c_{max} \right)^{-1/2} , \tag{141}
$$

*with the probability at least* $1 - \exp(-90(d + c)) - \exp(-90c^2 \log(M))$. *Here* $\hat{N}_k = \sum_{i=1}^{M} \sum_{j=1}^{N_i} \omega_{i,j;k}$, $E_0 = 1 - dist^2(\hat{\mathbf{B}}^0, \hat{\mathbf{B}}^*)$, $c_{min} = \eta K \frac{\min_k \hat{N}_k}{N} \bar{\sigma}_{\min, *}^2 E_0$, *and* $c_{max} = \eta K \frac{\max_k \hat{N}_k}{N} \bar{\sigma}_{\min, *}^2 E_0$.

*Proof.* From the optimization steps we have

$$\mathbf{B}^{t+1} = \hat{\mathbf{B}}^t - \sum_{k=1}^{K} \frac{\eta}{N} \left( \left( \bar{\boldsymbol{\Omega}}_k^T \odot \mathbf{X}^T \right) \left( \mathbf{X} \odot \bar{\boldsymbol{\Omega}}_k \right) \hat{\mathbf{B}}^t \boldsymbol{\theta}_k^t - \left( \bar{\boldsymbol{\Omega}}_k^T \odot \mathbf{X}^T \right) \left( \mathbf{Y} \odot \boldsymbol{\Omega}_k \right) \right) (\boldsymbol{\theta}_k^t)^T, \quad (142)$$

$$= \hat{\mathbf{B}}^t - \sum_{k=1}^{K} \frac{\eta}{N} \left( \left( \bar{\boldsymbol{\Omega}}_k^T \odot \mathbf{X}^T \right) \left( \mathbf{X} \odot \bar{\boldsymbol{\Omega}}_k \right) \left( \hat{\mathbf{B}}^t \boldsymbol{\theta}_k^t - \hat{\mathbf{B}}^* \boldsymbol{\theta}_k^* \right) - \left( \bar{\boldsymbol{\Omega}}_k^T \odot \mathbf{X}^T \right) \left( \mathbf{Z} \odot \boldsymbol{\Omega}_k \right) \right) (\boldsymbol{\theta}_k^t)^T,$$
$$(143)$$

$$= \hat{\mathbf{B}}^t - \eta \left( \sum_{k=1}^{K} \left( \frac{1}{N} \left( \bar{\boldsymbol{\Omega}}_k^T \odot \mathbf{X}^T \right) \left( \mathbf{X} \odot \bar{\boldsymbol{\Omega}}_k \right) \left( \hat{\mathbf{B}}^t \boldsymbol{\theta}_k^t - \hat{\mathbf{B}}^* \boldsymbol{\theta}_k^* \right) - \frac{\hat{N}_k}{N} \left( \hat{\mathbf{B}}^t \boldsymbol{\theta}_k^t - \hat{\mathbf{B}}^* \boldsymbol{\theta}_k^* \right) \right) (\boldsymbol{\theta}_k^t)^T \right)$$

$$- \frac{\eta}{N} \sum_{k=1}^{K} \hat{N}_k \left( \hat{\mathbf{B}}^t \boldsymbol{\theta}_k^t - \hat{\mathbf{B}}^* \boldsymbol{\theta}_k^* \right) (\boldsymbol{\theta}_k^t)^T + \frac{\eta}{N} \sum_{k=1}^{K} \left( \bar{\boldsymbol{\Omega}}_k^T \odot \mathbf{X}^T \right) \left( \mathbf{Z} \odot \boldsymbol{\Omega}_k \right) (\boldsymbol{\theta}_k^t)^T. \quad (144)$$

Multiplying both sides by $(\hat{\mathbf{B}}_\perp^*)^T$, we have

$$(\hat{\mathbf{B}}_\perp^*)^T \mathbf{B}^{t+1} = (\hat{\mathbf{B}}_\perp^*)^T \hat{\mathbf{B}}^t$$

$$- \eta (\hat{\mathbf{B}}_\perp^*)^T \left( \sum_{k=1}^{K} \left( \frac{1}{N} \left( \bar{\boldsymbol{\Omega}}_k^T \odot \mathbf{X}^T \right) \left( \mathbf{X} \odot \bar{\boldsymbol{\Omega}}_k \right) \left( \hat{\mathbf{B}}^t \boldsymbol{\theta}_k^t - \hat{\mathbf{B}}^* \boldsymbol{\theta}_k^* \right) - \frac{\hat{N}_k}{N} \left( \hat{\mathbf{B}}^t \boldsymbol{\theta}_k^t - \hat{\mathbf{B}}^* \boldsymbol{\theta}_k^* \right) \right) (\boldsymbol{\theta}_k^t)^T \right)$$

$$- \frac{\eta}{N} \sum_{k=1}^{K} \hat{N}_k \left( (\hat{\mathbf{B}}_\perp^*)^T \hat{\mathbf{B}}^t \boldsymbol{\theta}_k^t - (\hat{\mathbf{B}}_\perp^*)^T \hat{\mathbf{B}}^* \boldsymbol{\theta}_k^* \right) (\boldsymbol{\theta}_k^t)^T + \frac{\eta}{N} \sum_{k=1}^{K} (\hat{\mathbf{B}}_\perp^*)^T \left( \bar{\boldsymbol{\Omega}}_k^T \odot \mathbf{X}^T \right) \left( \mathbf{Z} \odot \boldsymbol{\Omega}_k \right) (\boldsymbol{\theta}_k^t)^T,$$
$$(145)$$

$$= (\hat{\mathbf{B}}_\perp^*)^T \hat{\mathbf{B}}^t \left( \mathbf{I}_c - \frac{\eta}{N} \sum_{k=1}^{K} \hat{N}_k \boldsymbol{\theta}_k^t (\boldsymbol{\theta}_k^t)^T \right) + \frac{\eta}{N} \sum_{k=1}^{K} (\hat{\mathbf{B}}_\perp^*)^T \left( \bar{\boldsymbol{\Omega}}_k^T \odot \mathbf{X}^T \right) \left( \mathbf{Z} \odot \boldsymbol{\Omega}_k \right) (\boldsymbol{\theta}_k^t)^T$$

$$- \frac{\eta}{N} (\hat{\mathbf{B}}_\perp^*)^T \left( \sum_{k=1}^{K} \left( \left( \bar{\boldsymbol{\Omega}}_k^T \odot \mathbf{X}^T \right) \left( \mathbf{X} \odot \bar{\boldsymbol{\Omega}}_k \right) \left( \hat{\mathbf{B}}^t \boldsymbol{\theta}_k^t - \hat{\mathbf{B}}^* \boldsymbol{\theta}_k^* \right) - \hat{N}_k \left( \hat{\mathbf{B}}^t \boldsymbol{\theta}_k^t - \hat{\mathbf{B}}^* \boldsymbol{\theta}_k^* \right) \right) (\boldsymbol{\theta}_k^t)^T \right). \quad (146)$$

Because we have $\hat{\mathbf{B}}^{t+1} = \mathbf{B}^{t+1} \left( \mathbf{R}^{t+1} \right)^{-1}$, multiplying both sides by $\left( \mathbf{R}^{t+1} \right)^{-1}$ we have

$$dist \left( \hat{\mathbf{B}}^{t+1}, \hat{\mathbf{B}}^* \right)$$

$$\leq dist \left( \hat{\mathbf{B}}^t, \hat{\mathbf{B}}^* \right) \left\| \mathbf{I}_c - \eta \sum_{k=1}^{K} \boldsymbol{\theta}_k^t (\boldsymbol{\theta}_k^t)^T \right\|_2 \left\| (\mathbf{R}^{t+1})^{-1} \right\|_2$$

$$+ \left\| \frac{\eta}{N} \sum_{k=1}^{K} (\hat{\mathbf{B}}_\perp^*)^T \left( \bar{\boldsymbol{\Omega}}_k^T \odot \mathbf{X}^T \right) \left( \mathbf{Z} \odot \boldsymbol{\Omega}_k \right) (\boldsymbol{\theta}_k^t)^T \right\|_2 \left\| (\mathbf{R}^{t+1})^{-1} \right\|_2$$

$$+ \left\| \frac{\eta}{N} (\hat{\mathbf{B}}_\perp^*)^T \left( \sum_{k=1}^{K} \left( \left( \bar{\boldsymbol{\Omega}}_k^T \odot \mathbf{X}^T \right) \left( \mathbf{X} \odot \bar{\boldsymbol{\Omega}}_k \right) \left( \hat{\mathbf{B}}^t \boldsymbol{\theta}_k^t - \hat{\mathbf{B}}^* \boldsymbol{\theta}_k^* \right) - \hat{N}_k \left( \hat{\mathbf{B}}^t \boldsymbol{\theta}_k^t - \hat{\mathbf{B}}^* \boldsymbol{\theta}_k^* \right) \right) (\boldsymbol{\theta}_k^t)^T \right) \right\|_2$$

$$\left\| (\mathbf{R}^{t+1})^{-1} \right\|_2. \quad (147)$$

Then we can define

$$A_1 = dist \left( \hat{\mathbf{B}}^t, \hat{\mathbf{B}}^* \right) \left\| \mathbf{I}_c - \frac{\eta}{N} \sum_{k=1}^{K} \hat{N}_k \boldsymbol{\theta}_k^t (\boldsymbol{\theta}_k^t)^T \right\|_2, \quad (148)$$

$$A_2 = \left\| \frac{\eta}{N} \sum_{k=1}^{K} (\hat{\mathbf{B}}_\perp^*)^T \left( \bar{\boldsymbol{\Omega}}_k^T \odot \mathbf{X}^T \right) \left( \mathbf{Z} \odot \boldsymbol{\Omega}_k \right) (\boldsymbol{\theta}_k^t)^T \right\|_2, \quad (149)$$

$$A_3 = \left\| \frac{\eta}{N} (\hat{\mathbf{B}}_\perp^*)^T \left( \sum_{k=1}^{K} \left( \left( \bar{\boldsymbol{\Omega}}_k^T \odot \mathbf{X}^T \right) \left( \mathbf{X} \odot \bar{\boldsymbol{\Omega}}_k \right) \left( \hat{\mathbf{B}}^t \boldsymbol{\theta}_k^t - \hat{\mathbf{B}}^* \boldsymbol{\theta}_k^* \right) - \hat{N}_k \left( \hat{\mathbf{B}}^t \boldsymbol{\theta}_k^t - \hat{\mathbf{B}}^* \boldsymbol{\theta}_k^* \right) \right) (\boldsymbol{\theta}_k^t)^T \right) \right\|_2. \quad (150)$$

Then the inequality become

$$dist \left( \hat{\mathbf{B}}^{t+1}, \hat{\mathbf{B}}^* \right) \leq (A_1 + A_2 + A_3) \left\| (\mathbf{R}^{t+1})^{-1} \right\|_2. \quad (151)$$

For the following parts of the proof, we consider the following events hold

$$\mathcal{E}_1 = \bigcap_{k=1}^{K} \left\{ \mathcal{A}_k \bigcap \mathcal{B}_k \right\} , \tag{152}$$

$$\mathcal{E}_2 = \left\{ \|\mathbf{F}^t\|_F \leq \frac{\delta}{(1-\delta)\sqrt{K}} \|\mathbf{\Theta}^*\|_2 \, dist\left(\hat{\mathbf{B}}^t, \hat{\mathbf{B}}^*\right) \bigcap \|\mathbf{G}^t\|_F \leq \sqrt{K} \frac{\delta}{(1-\delta)} \sigma^2 \right\} , \tag{153}$$

$$\mathcal{E}_3 = \left\{ \left\| \frac{1}{N} \sum_{k=1}^{K} \left( \bar{\mathbf{\Omega}}_k^T \odot \mathbf{X}^T \right) (\mathbf{Z} \odot \mathbf{\Omega}_k) (\boldsymbol{\theta}_k^t)^T \right\|_2 \leq \mathcal{C}_1 \sigma^2 \frac{\sqrt{d+c}}{\sqrt{N}} \left( \sqrt{c} + \frac{\delta}{1-\delta} dist\left(\hat{\mathbf{B}}, \hat{\mathbf{B}}^*\right) + \frac{\delta}{1-\delta} \sigma^2 \right) \right\} , \tag{154}$$

$$\mathcal{E}_4 = \left\{ \left\| \sum_{k=1}^{K} \frac{1}{N} \left( \left( \bar{\mathbf{\Omega}}_k^T \odot \mathbf{X}^T \right) (\mathbf{X} \odot \bar{\mathbf{\Omega}}_k) \left( \hat{\mathbf{B}} \boldsymbol{\theta}_k^t - \hat{\mathbf{B}}^* \boldsymbol{\theta}_k^* \right) - \hat{N}_k \left( \hat{\mathbf{B}} \boldsymbol{\theta}_k^t - \hat{\mathbf{B}}^* \boldsymbol{\theta}_k^* \right) \right) (\boldsymbol{\theta}_k^t)^T \right\|_2 \leq \mathcal{C}_2 \frac{\sqrt{d+c}}{\sqrt{N}} \epsilon \right\} , \tag{155}$$

$$\tag{156}$$

where

$$\mathcal{A}_k = \left\{ \|\boldsymbol{\theta}_k^t\|_2 \leq \sqrt{c} + \frac{\delta}{1-\delta} dist\left(\hat{\mathbf{B}}, \hat{\mathbf{B}}^*\right) + \frac{\delta}{1-\delta} \sigma^2 \right\} , \tag{157}$$

$$\mathcal{B}_k = \left\{ \left\| \hat{\mathbf{B}} \boldsymbol{\theta}_k^t - \hat{\mathbf{B}}^* \boldsymbol{\theta}_k^* \right\|_2 \leq \left( \sqrt{c} + \frac{\delta}{1-\delta} \right) dist\left(\hat{\mathbf{B}}^t, \hat{\mathbf{B}}^*\right) + \frac{\delta}{1-\delta} \sigma^2 \right\} , \tag{158}$$

$$\epsilon = \left( \sqrt{c} \frac{\delta}{1-\delta} + \frac{\delta^2}{(1-\delta)^2} \right) dist^2\left(\hat{\mathbf{B}}, \hat{\mathbf{B}}^*\right)$$
$$+ \left( c + (\sigma^2 + 1)\sqrt{c} \frac{\delta}{1-\delta} + \frac{2\delta^2}{(1-\delta)^2} \sigma^2 \right) dist\left(\hat{\mathbf{B}}, \hat{\mathbf{B}}^*\right) + \sqrt{c} \frac{\delta}{1-\delta} \sigma^2 + \frac{\delta^2}{(1-\delta)^2} \sigma^4 . \tag{159}$$

which hold with probability at least $1 - \exp(-90(d+c)) - \exp(-90c^2 \log(M))$ for some constants $\mathcal{C}_1, \mathcal{C}_2$ by Lemma B.8, B.9, B.10, B.11. Then we consider to bound $A_1, A_2, A_3$, respectively. Then we consider to bound $A_1$ first, and we have

$$\lambda_{\max}((\mathbf{\Theta}^t)^T \mathbf{\Theta}^t) = \left\| \mathbf{\Theta}^t \right\|_2^2 = \left\| \mathbf{\Theta}^* (\hat{\mathbf{B}}^*)^T \hat{\mathbf{B}}^t + \mathbf{F}^t + \mathbf{G}^t \right\|_2^2 , \tag{160}$$

$$\leq 2 \left\| \mathbf{\Theta}^* \right\|_2^2 + 2 \left\| \mathbf{F}^t \right\|_2^2 + 2 \left\| \mathbf{G}^t \right\|_2^2 , \tag{161}$$

$$\leq 2 \left\| \mathbf{\Theta}^* \right\|_2^2 + \frac{2\delta^2}{(1-\delta)^2 K} \left\| \mathbf{\Theta}^* \right\|_2^2 dist^2\left(\hat{\mathbf{B}}^t, \hat{\mathbf{B}}^*\right) + 2K \frac{\delta}{(1-\delta)^2} \sigma^4 , \tag{162}$$

$$\leq \left( 2 + \frac{2\delta^2}{(1-\delta)^2 K} \right) \left\| \mathbf{\Theta}^* \right\|_2^2 + 2K \frac{\delta}{(1-\delta)^2} \sigma^4 , \tag{163}$$

$$\leq \left( 2K + \frac{2\delta^2}{(1-\delta)^2} \right) \bar{\sigma}_{\max,*}^2 + 2K \frac{\delta}{(1-\delta)^2} \sigma^4 . \tag{164}$$

Then when $\eta$ is small enough, it's simple to promise that $\mathbf{I}_c - \frac{\eta}{N}\sum_{k=1}^{K}\boldsymbol{\theta}_k\boldsymbol{\theta}_k^T$ is positive definite. Define diagonal matrix $\mathbf{W} \in \mathbb{R}^{K \times K}$, and $\mathbf{W}_{k,k} = \frac{\hat{N}_k}{N}$. Then we have

$$\left\|\mathbf{I}_c - \frac{\eta}{N}\sum_{k=1}^{K}\hat{N}_k\boldsymbol{\theta}_k^t(\boldsymbol{\theta}_k^t)^T\right\|_2 = \left\|\mathbf{I}_c - \eta(\mathbf{W}\boldsymbol{\Theta}^t)^T\boldsymbol{\Theta}^t\right\|_2 \tag{165}$$

$$\leq 1 - \eta\lambda_{\min}\left((\mathbf{W}\boldsymbol{\Theta}^t)^T\boldsymbol{\Theta}^t\right), \tag{166}$$

$$\leq 1 - \eta\left(\sigma_{\min}\left(\mathbf{W}\right)\sigma_{\min}^2\left(\boldsymbol{\Theta}^*(\hat{\mathbf{B}}^*)^T\hat{\mathbf{B}}^t\right) - \sigma_{\min}((\mathbf{W}\mathbf{F}^t)^T\mathbf{F}^t) - \sigma_{\min}((\mathbf{W}\mathbf{G}^t)^T\mathbf{G}^t)\right)$$
$$+ 2\eta\left(\sigma_{\max}\left((\mathbf{W}\mathbf{F}^t)^T\boldsymbol{\Theta}^*(\hat{\mathbf{B}}^*)^T\hat{\mathbf{B}}^t\right) + \sigma_{\max}\left((\mathbf{W}\mathbf{F}^t)^T\mathbf{G}\right) + \sigma_{\max}\left((\mathbf{W}\mathbf{G}^t)^T\boldsymbol{\Theta}^*(\hat{\mathbf{B}}^*)^T\hat{\mathbf{B}}^t\right)\right), \tag{167}$$

$$\leq 1 - \eta\left(\frac{\min_k \hat{N}_k}{N}\right)\sigma_{\min}^2(\boldsymbol{\Theta}^*)\sigma_{\min}^2\left((\hat{\mathbf{B}}^*)^T\hat{\mathbf{B}}^t\right) + \eta\left(\frac{\max_k \hat{N}_k}{N}\right)\sigma_{\min}^2(\mathbf{F}^t)$$

$$+ \eta\left(\frac{\max_k \hat{N}_k}{N}\right)\sigma_{\min}^2(\mathbf{G}^t)$$

$$+ \frac{2\eta}{N}\left(\frac{\max_k \hat{N}_k}{N}\right)\left(\sigma_{\max}\left((\mathbf{F}^t)^T\boldsymbol{\Theta}^*\right) + \sigma_{\max}\left((\mathbf{G}^t)^T\boldsymbol{\Theta}^*\right)\right) + 2\eta\left(\frac{\max_k \hat{N}_k}{N}\right)\sigma_{\max}(\mathbf{F}^t)\sigma_{\max}(\mathbf{G}^t), \tag{168}$$

$$\leq 1 - \eta K\left(\frac{\min_k \hat{N}_k}{N}\right)\bar{\sigma}_{\min,*}^2\sigma_{\min}^2\left((\hat{\mathbf{B}}^*)^T\hat{\mathbf{B}}^t\right) + 2\eta\left(\frac{\max_k \hat{N}_k}{N}\right)\left(\|\mathbf{F}^t\|_2 + \|\mathbf{G}^t\|_2\right)\|\boldsymbol{\Theta}^*\|_2$$

$$+ 2\eta\left(\frac{\max_k \hat{N}_k}{N}\right)\|\mathbf{F}^t\|_2\|\mathbf{G}^t\|_2 + \eta\left(\frac{\max_k \hat{N}_k}{N}\right)\|\mathbf{F}^t\|_2^2 + \eta\left(\frac{\max_k \hat{N}_k}{N}\right)\|\mathbf{G}^t\|_2^2, \tag{169}$$

$$= 1 - \eta K\left(\frac{\min_k \hat{N}_k}{N}\right)\bar{\sigma}_{\min,*}^2\sigma_{\min}^2\left((\hat{\mathbf{B}}^*)^T\hat{\mathbf{B}}^t\right) + 2\eta\left(\frac{\max_k \hat{N}_k}{N}\right)\left(\|\mathbf{F}^t\|_2 + \|\mathbf{G}^t\|_2\right)\|\boldsymbol{\Theta}^*\|_2$$

$$+ \eta\left(\frac{\max_k \hat{N}_k}{N}\right)\left(\|\mathbf{F}^t\|_2 + \|\mathbf{G}^t\|_2\right)^2. \tag{170}$$

Then under condition $\mathcal{E}_1, \mathcal{E}_2, \mathcal{E}_3, \mathcal{E}_4$, we have

$$\left\| \mathbf{I}_c - \frac{\eta}{N} \sum_{k=1}^{K} \hat{N}_k \boldsymbol{\theta}_k^t (\boldsymbol{\theta}_k^t)^T \right\|_2$$

$$\leq 1 - \eta K \left( \frac{\min_k \hat{N}_k}{N} \right) \bar{\sigma}_{\min,*}^2 \sigma_{\min}^2 \left( (\hat{\mathbf{B}}^*)^T \hat{\mathbf{B}}^t \right)$$

$$+ 2\eta \left( \frac{\max_k \hat{N}_k}{N} \right) \left( \frac{\delta}{(1-\delta)\sqrt{K}} \|\boldsymbol{\Theta}^*\|_2 \, dist \left( \hat{\mathbf{B}}^t, \hat{\mathbf{B}}^* \right) + \sqrt{K} \frac{\delta}{1-\delta} \sigma^2 \right) \|\boldsymbol{\Theta}^*\|_2$$

$$+ \eta \left( \frac{\max_k \hat{N}_k}{N} \right) \left( \frac{\delta}{(1-\delta)\sqrt{K}} \|\boldsymbol{\Theta}^*\|_2 \, dist \left( \hat{\mathbf{B}}^t, \hat{\mathbf{B}}^* \right) + \sqrt{K} \frac{\delta}{1-\delta} \sigma^2 \right)^2 , \tag{171}$$

$$\leq 1 - \eta K \left( \frac{\min_k \hat{N}_k}{N} \right) \bar{\sigma}_{\min,*}^2 \sigma_{\min}^2 \left( (\hat{\mathbf{B}}^*)^T \hat{\mathbf{B}}^t \right)$$

$$+ 2\eta \left( \frac{\max_k \hat{N}_k}{N} \right) \left( \frac{\delta\sqrt{K}}{1-\delta} \bar{\sigma}_{\max,*}^2 + \frac{\delta K \sigma^2}{1-\delta} \bar{\sigma}_{\max,*} \right)$$

$$+ \eta \left( \frac{\max_k \hat{N}_k}{N} \right) \frac{\delta^2}{(1-\delta)^2} \left( 2\bar{\sigma}_{\max,*}^2 + 2K\sigma^4 \right) , \tag{172}$$

$$\leq 1 - \eta K \left( \frac{\min_k \hat{N}_k}{N} \right) \bar{\sigma}_{\min,*}^2 E_0 + \eta K \left( \frac{\max_k \hat{N}_k}{N} \right) \frac{\delta}{1-\delta} \left( 2\bar{\sigma}_{\max,*} + \frac{1}{2}\sigma^2 \right)^2 , \tag{173}$$

$$\leq 1 - \eta K \left( \frac{\min_k \hat{N}_k}{N} \right) \bar{\sigma}_{\min,*}^2 E_0 + \frac{\eta K}{5} \left( \frac{\max_k \hat{N}_k}{N} \right) \bar{\sigma}_{\min,*}^2 E_0 + \frac{\eta K}{20} \left( \frac{\max_k \hat{N}_k}{N} \right) \bar{\sigma}_{\min,*}^2 E_0 . \tag{174}$$

where $E_0 = 1 - dist^2 \left( \hat{\mathbf{B}}^0, \hat{\mathbf{B}}^* \right) \leq \sigma_{\min}^2 \left( (\hat{\mathbf{B}}^*)^T \hat{\mathbf{B}}^t \right)$. The Equation (173) holds when $\hat{N}_k$ is large enough that makes the following equation holds

$$\frac{\delta}{1-\delta} \leq 2\delta \leq \frac{E_0}{20(1+\sigma^2)^2} \max \left\{ \frac{1}{\kappa^2}, \bar{\sigma}_{\min}^2 \right\} , \tag{175}$$

where $\kappa = \frac{\bar{\sigma}_{\max,*}}{\bar{\sigma}_{\min,*}}$, and the equation holds when $\hat{N}_k$ satisfy

$$\min_k \hat{N}_k \geq \mathcal{C}_0 \frac{c^3 (1+\sigma^2)^4 \log(M)}{E_0^2} \min \left\{ \kappa^4, \frac{1}{\bar{\sigma}_{\min}^4} \right\} . \tag{176}$$

Then consider $A_1$, we will have

$$A_1 \leq dist \left( \hat{\mathbf{B}}^t, \hat{\mathbf{B}}^* \right) \left( 1 - \left( \left( \frac{\min_k \hat{N}_k}{N} \right) - \frac{1}{4} \left( \frac{\max_k \hat{N}_k}{N} \right) \right) \eta K \bar{\sigma}_{\min,*}^2 E_0 \right) . \tag{177}$$

The consider $A_2$, because $\left\| \hat{\mathbf{B}}_\perp^* \right\|_2 = 1$, and $\mathcal{E}_3$ holds, we have

$$A_2 \leq \eta \mathcal{C}_1 \sigma^2 \frac{\sqrt{d+c}}{\sqrt{N}} \left( \sqrt{c} + \frac{\delta}{1-\delta} dist \left( \hat{\mathbf{B}}, \hat{\mathbf{B}}^* \right) + \frac{\delta}{1-\delta} \sigma^2 \right) , \tag{178}$$

$$\leq \eta \mathcal{C}_1 \sigma^2 \frac{\sqrt{d+c}}{\sqrt{N}} \left( \sqrt{c} + \frac{1}{10} \right) . \tag{179}$$

Similarly, for $A_3$, we have

$$A_3 \leq \eta \mathcal{C}_2 \frac{\sqrt{d+c}}{\sqrt{N}} \epsilon , \tag{180}$$

$$\leq \eta \mathcal{C}_2 \frac{\sqrt{d+c}}{\sqrt{N}} \left( \left( \sqrt{c} + \frac{1}{\sqrt{10}} \right)^2 dist \left( \hat{\mathbf{B}}^t, \hat{\mathbf{B}}^* \right) + \left( \frac{1}{400} + \frac{\sqrt{c}}{20} \right) (\sigma^2 + 1) \right) . \tag{181}$$

Combining Equation (151), (177), (179), and (181), and choose $\mathcal{C} = \max\{\mathcal{C}_1, \mathcal{C}_2, \mathcal{C}_1\mathcal{C}_2 + \mathcal{C}_2\}$, we have

$$dist\left(\hat{\mathbf{B}}^{t+1}, \hat{\mathbf{B}}^*\right) \leq dist\left(\hat{\mathbf{B}}^t, \hat{\mathbf{B}}^*\right)$$

$$\left(1 - \eta K\left(\left(\frac{\min_k \hat{N}_k}{N}\right)^2 - \frac{1}{4}\left(\frac{\max_k \hat{N}_k}{N}\right)^2\right)\right)\bar{\sigma}_{\min,*}^2 E_0 + \eta\mathcal{C}\frac{\sqrt{d+c}}{\sqrt{N}}\left(\sqrt{c} + \frac{1}{\sqrt{10}}\right)^2 \left\|(\mathbf{R}^{t+1})^{-1}\right\|_2$$

$$+ \eta\mathcal{C}\frac{\sqrt{d+c}}{\sqrt{N}}(\sigma^2 + 1)\left(2\sqrt{c} + \frac{1}{5}\right)\left\|(\mathbf{R}^{t+1})^{-1}\right\|_2 . \tag{182}$$

Then the remaining thing is to bound $(\mathbf{R}^{t+1})^{-1}$. Firstly we define

$$\mathbf{S}^t = \sum_{k=1}^{K}\left(\bar{\mathbf{\Omega}}_k^T \odot \mathbf{X}^T\right)\left(\mathbf{X} \odot \bar{\mathbf{\Omega}}_k\right)\left(\hat{\mathbf{B}}^t\boldsymbol{\theta}_k^t - \hat{\mathbf{B}}^*\boldsymbol{\theta}_k^*\right)(\boldsymbol{\theta}_k^t)^T , \tag{183}$$

$$\mathbf{E}^t = \sum_{k=1}^{K}\left(\bar{\mathbf{\Omega}}_k^T \odot \mathbf{X}^T\right)\left(\mathbf{Z} \odot \mathbf{\Omega}_k\right)(\boldsymbol{\theta}_k^t)^T . \tag{184}$$

Then we have

$$\mathbf{B}^{t+1} = \hat{\mathbf{B}}^t - \frac{\eta}{N}\mathbf{S}^t + \frac{\eta}{N}\mathbf{E}^t . \tag{185}$$

Because $(\mathbf{R}^{t+1})^T\mathbf{R}^{t+1} = (\mathbf{B}^{t+1})^T\mathbf{B}^{t+1}$, and we have

$$(\mathbf{B}^{t+1})^T\mathbf{B}^{t+1} = (\hat{\mathbf{B}}^t)^T\hat{\mathbf{B}}^t - \frac{\eta}{N}\left((\hat{\mathbf{B}}^t)^T\mathbf{S}^t + (\mathbf{S}^t)^T\hat{\mathbf{B}}^t\right) + \frac{\eta}{N}\left((\hat{\mathbf{B}}^t)^T\mathbf{E}^t + (\mathbf{E}^t)^T\hat{\mathbf{B}}^t\right)$$

$$+ \frac{\eta^2}{N^2}(\mathbf{S}^t)^T\mathbf{S}^t - \frac{\eta^2}{N^2}\left((\mathbf{E}^t)^T\mathbf{S}^t + (\mathbf{S}^t)^T\mathbf{E}^t\right) + \frac{\eta^2}{N^2}(\mathbf{E}^t)^T\mathbf{E}^t , \tag{186}$$

$$= \mathbf{I}_c - \frac{\eta}{N}\left((\hat{\mathbf{B}}^t)^T\mathbf{S}^t + (\mathbf{S}^t)^T\hat{\mathbf{B}}^t\right) + \frac{\eta}{N}\left((\hat{\mathbf{B}}^t)^T\mathbf{E}^t + (\mathbf{E}^t)^T\hat{\mathbf{B}}^t\right)$$

$$+ \frac{\eta^2}{N^2}(\mathbf{S}^t)^T\mathbf{S}^t - \frac{\eta^2}{N^2}\left((\mathbf{E}^t)^T\mathbf{S}^t + (\mathbf{S}^t)^T\mathbf{E}^t\right) + \frac{\eta^2}{N^2}(\mathbf{E}^t)^T\mathbf{E}^t . \tag{187}$$

By Weyl's inequality, we have

$$\sigma_{\min}^2\left(\mathbf{R}^{t+1}\right)$$

$$\geq 1 - \frac{\eta}{N}\lambda_{\max}\left((\hat{\mathbf{B}}^t)^T\mathbf{S}^t + (\mathbf{S}^t)^T\hat{\mathbf{B}}^t\right)$$

$$- \frac{\eta}{N}\lambda_{\max}\left((\hat{\mathbf{B}}^t)^T\mathbf{E}^t + (\mathbf{E}^t)^T\hat{\mathbf{B}}^t\right) - \frac{\eta^2}{N^2}\lambda_{\max}\left((\mathbf{E}^t)^T\mathbf{S}^t + (\mathbf{S}^t)^T\mathbf{E}^t\right) . \tag{188}$$

Then we can define

$$R_1 = \frac{\eta}{N}\lambda_{\max}\left((\hat{\mathbf{B}}^t)^T\mathbf{S}^t + (\mathbf{S}^t)^T\hat{\mathbf{B}}^t\right) , \tag{189}$$

$$R_2 = \frac{\eta}{N}\lambda_{\max}\left((\hat{\mathbf{B}}^t)^T\mathbf{E}^t + (\mathbf{E}^t)^T\hat{\mathbf{B}}^t\right) , \tag{190}$$

$$R_3 = \frac{\eta^2}{N^2}\lambda_{\max}\left((\mathbf{E}^t)^T\mathbf{S}^t + (\mathbf{S}^t)^T\mathbf{E}^t\right) . \tag{191}$$

Then we have

$$\sigma_{\min}^2\left(\mathbf{R}^{t+1}\right) \geq 1 - R_1 - R_2 - R_3 . \tag{192}$$

Then we consider to bound $R_1, R_2$, and $R_3$, respectively. Consider $R_1$ first, and we have

$$R_1$$

$$= \frac{2\eta}{N}\max_{\|\mathbf{p}\|_2=1}\mathbf{p}^T(\hat{\mathbf{B}}^t)^T\mathbf{S}^t\mathbf{p} , \tag{193}$$

$$= \frac{2\eta}{N}\max_{\|\mathbf{p}\|_2=1}\mathbf{p}^T(\hat{\mathbf{B}}^t)^T\left(\sum_{k=1}^{K}\left(\bar{\mathbf{\Omega}}_k^T \odot \mathbf{X}^T\right)\left(\mathbf{X} \odot \bar{\mathbf{\Omega}}_k\right)\left(\hat{\mathbf{B}}^t\boldsymbol{\theta}_k^t - \hat{\mathbf{B}}^*\boldsymbol{\theta}_k^*\right)(\boldsymbol{\theta}_k^t)^T\right)\mathbf{p} , \tag{194}$$

$$\leq \frac{2\eta}{N}\max_{\|\mathbf{p}\|_2=1}\mathbf{p}^T(\hat{\mathbf{B}}^t)^T\left(\sum_{k=1}^{K}\left(\left(\bar{\mathbf{\Omega}}_k^T \odot \mathbf{X}^T\right)\left(\mathbf{X} \odot \bar{\mathbf{\Omega}}_k\right)\left(\hat{\mathbf{B}}^t\boldsymbol{\theta}_k^t - \hat{\mathbf{B}}^*\boldsymbol{\theta}_k^*\right) - \hat{N}_k\left(\hat{\mathbf{B}}^t\boldsymbol{\theta}_k^t - \hat{\mathbf{B}}^*\boldsymbol{\theta}_k^*\right)\right)(\boldsymbol{\theta}_k^t)^T\right)\mathbf{p}$$

$$+ \frac{2\eta}{N}\max_{\|\mathbf{p}\|_2=1}\mathbf{p}^T(\hat{\mathbf{B}}^t)^T\left(\sum_{k=1}^{K}\hat{N}_k\left(\hat{\mathbf{B}}^t\boldsymbol{\theta}_k^t - \hat{\mathbf{B}}^*\boldsymbol{\theta}_k^*\right)(\boldsymbol{\theta}_k^t)^T\right)\mathbf{p} . \tag{195}$$

Under the condition $\mathcal{E}_4$, we have

$$R_1 \leq 2\eta \mathcal{C}_2 \left\| \hat{\mathbf{B}}^t \right\|_2 \frac{\sqrt{d+c}}{\sqrt{N}} \epsilon + \frac{2\eta}{N} \max_{\|\mathbf{p}\|_2=1} \mathbf{p}^T (\hat{\mathbf{B}}^t)^T \left( \sum_{k=1}^K \hat{N}_k \left( \hat{\mathbf{B}}^t \boldsymbol{\theta}_k^t - \hat{\mathbf{B}}^* \boldsymbol{\theta}_k^* \right) (\boldsymbol{\theta}_k^t)^T \right) \mathbf{p}, \quad (196)$$

$$\leq \left( 2\eta \mathcal{C}_2 \frac{\sqrt{d+c}}{\sqrt{N}} \right) \left( \left( \sqrt{c} + \frac{1}{\sqrt{10}} \right)^2 dist \left( \hat{\mathbf{B}}^t, \hat{\mathbf{B}}^* \right) + \left( \frac{1}{400} + \frac{\sqrt{c}}{20} \right) (\sigma^2 + 1) \right)$$

$$+ \frac{2\eta}{N} \max_{\|\mathbf{p}\|_2=1} \mathbf{p}^T (\hat{\mathbf{B}}^t)^T \left( \sum_{k=1}^K \hat{N}_k \left( \hat{\mathbf{B}}^t \boldsymbol{\theta}_k^t - \hat{\mathbf{B}}^* \boldsymbol{\theta}_k^* \right) (\boldsymbol{\theta}_k^t)^T \right) \mathbf{p}, \quad (197)$$

$$\leq 4\eta \left( \mathcal{C}_2 \frac{\sqrt{d+c}}{\sqrt{N}} \right) \left( (\sigma^2 + 1) \left( \sqrt{c} + \frac{1}{\sqrt{10}} \right)^2 \right)$$

$$+ \frac{2\eta}{N} \max_{\|\mathbf{p}\|_2=1} \mathbf{p}^T (\hat{\mathbf{B}}^t)^T \left( \sum_{k=1}^K \hat{N}_k \left( \hat{\mathbf{B}}^t \boldsymbol{\theta}_k^t - \hat{\mathbf{B}}^* \boldsymbol{\theta}_k^* \right) (\boldsymbol{\theta}_k^t)^T \right) \mathbf{p} \quad (198)$$

The remaining thing is to bound $\frac{2\eta}{N} \mathbf{p}^T (\hat{\mathbf{B}}^t)^T \left( \sum_{k=1}^K \hat{N}_k \left( \hat{\mathbf{B}}^t \boldsymbol{\theta}_k^t - \hat{\mathbf{B}}^* \boldsymbol{\theta}_k^* \right) (\boldsymbol{\theta}_k^t)^T \right) \mathbf{p}$, and we have

$$\frac{2\eta}{N} \mathbf{p}^T (\hat{\mathbf{B}}^t)^T \left( \sum_{k=1}^K \hat{N}_k \left( \hat{\mathbf{B}}^t \boldsymbol{\theta}_k^t - \hat{\mathbf{B}}^* \boldsymbol{\theta}_k^* \right) (\boldsymbol{\theta}_k^t)^T \right) \mathbf{p}$$

$$= \frac{2\eta}{N} tr \left[ \sum_{k=1}^K \hat{N}_k \left( \hat{\mathbf{B}}^t \boldsymbol{\theta}_k^t - \hat{\mathbf{B}}^* \boldsymbol{\theta}_k^* \right) (\boldsymbol{\theta}_k^t)^T \mathbf{p} \mathbf{p}^T (\hat{\mathbf{B}}^t)^T \right], \quad (199)$$

$$= \frac{2\eta}{N} tr \left[ \sum_{k=1}^K \hat{N}_k \left( \hat{\mathbf{B}}^t \boldsymbol{\theta}_k^t - \hat{\mathbf{B}}^* \boldsymbol{\theta}_k^* \right) \left( (\hat{\mathbf{B}}^t)^T \hat{\mathbf{B}}^* \boldsymbol{\theta}_k^* + \mathbf{F}_k^t + \mathbf{G}_k^t \right)^T \mathbf{p} \mathbf{p}^T (\hat{\mathbf{B}}^t)^T \right]. \quad (200)$$

Then we can define

$$T_1 = \frac{2\eta}{N} tr \left[ \sum_{k=1}^K \hat{N}_k \left( \hat{\mathbf{B}}^t \boldsymbol{\theta}_k^t - \hat{\mathbf{B}}^* \boldsymbol{\theta}_k^* \right) (\boldsymbol{\theta}_k^*)^T (\hat{\mathbf{B}}^*)^T \hat{\mathbf{B}}^t \mathbf{p} \mathbf{p}^T (\hat{\mathbf{B}}^t)^T \right], \quad (201)$$

$$T_2 = \frac{2\eta}{N} tr \left[ \sum_{k=1}^K \hat{N}_k \left( \hat{\mathbf{B}}^t \boldsymbol{\theta}_k^t - \hat{\mathbf{B}}^* \boldsymbol{\theta}_k^* \right) (\mathbf{F}_k^t)^T \mathbf{p} \mathbf{p}^T (\hat{\mathbf{B}}^t)^T \right], \quad (202)$$

$$T_3 = \frac{2\eta}{N} tr \left[ \sum_{k=1}^K \hat{N}_k \left( \hat{\mathbf{B}}^t \boldsymbol{\theta}_k^t - \hat{\mathbf{B}}^* \boldsymbol{\theta}_k^* \right) (\mathbf{G}_k^t)^T \mathbf{p} \mathbf{p}^T (\hat{\mathbf{B}}^t)^T \right]. \quad (203)$$

Consider $T_1$ first, we have

$$T_1 = \frac{2\eta}{N} tr \left[ \sum_{k=1}^K \hat{N}_k \left( \hat{\mathbf{B}}^t (\hat{\mathbf{B}}^t)^T \hat{\mathbf{B}}^* \boldsymbol{\theta}_k^* + \hat{\mathbf{B}}^t \mathbf{F}_k^t + \hat{\mathbf{B}}^t \mathbf{G}_k^t - \hat{\mathbf{B}}^* \boldsymbol{\theta}_k^* \right) (\boldsymbol{\theta}_k^*)^T (\hat{\mathbf{B}}^*)^T \hat{\mathbf{B}}^t \mathbf{p} \mathbf{p}^T (\hat{\mathbf{B}}^t)^T \right], \quad (204)$$

$$= \frac{2\eta}{N} tr \left[ \left( \hat{\mathbf{B}}^t (\hat{\mathbf{B}}^t)^T - \mathbf{I}_d \right) \sum_{k=1}^K \hat{N}_k \hat{\mathbf{B}}^* \boldsymbol{\theta}_k^* (\boldsymbol{\theta}_k^*)^T (\hat{\mathbf{B}}^*)^T \hat{\mathbf{B}}^t \mathbf{p} \mathbf{p}^T (\hat{\mathbf{B}}^t)^T \right]$$

$$+ \frac{2\eta}{N} tr \left[ \sum_{k=1}^K \hat{N}_k \left( \hat{\mathbf{B}}^t \mathbf{F}_k^t (\boldsymbol{\theta}_k^*)^T \right) (\hat{\mathbf{B}}^*)^T \hat{\mathbf{B}}^t \mathbf{p} \mathbf{p}^T (\hat{\mathbf{B}}^t)^T \right]$$

$$+ \frac{2\eta}{N} tr \left[ \sum_{k=1}^K \hat{N}_k \left( \hat{\mathbf{B}}^t \mathbf{G}_k^t (\boldsymbol{\theta}_k^*)^T \right) (\hat{\mathbf{B}}^*)^T \hat{\mathbf{B}}^t \mathbf{p} \mathbf{p}^T (\hat{\mathbf{B}}^t)^T \right], \quad (205)$$

$$= 2\eta tr \left[ (\hat{\mathbf{B}}^t)^T \hat{\mathbf{B}}^t \left( (\mathbf{F}^t)^T + (\mathbf{G}^t)^T \right) \mathbf{W} \boldsymbol{\Theta}^* (\hat{\mathbf{B}}^*)^T \hat{\mathbf{B}}^t \mathbf{p} \mathbf{p}^T \right], \quad (206)$$

$$\leq 2\eta \frac{\max_k N_k}{N} \left( \|\mathbf{F}^t\|_F + \|\mathbf{G}^t\|_F \right) \|\boldsymbol{\Theta}^*\|_2. \quad (207)$$

Because $\mathcal{E}_2$ holds, we have

$$T_1 \leq 2\eta \frac{\max_k N_k}{N} \left( \frac{\delta}{(1-\delta)\sqrt{K}} \|\mathbf{\Theta}^*\|_2^2 + \sqrt{K}\frac{\delta}{1-\delta}\sigma^2 \|\mathbf{\Theta}^*\|_2 \right), \tag{208}$$

$$\leq 2\eta K \frac{\max_k N_k}{N} \left( \frac{\delta}{1-\delta}\bar{\sigma}_{\max,*}^2 + \frac{\delta}{1-\delta}\sigma^2\bar{\sigma}_{\max,*} \right), \tag{209}$$

$$\leq \frac{\eta K}{5} \frac{\max_k N_k}{N} E_0 \bar{\sigma}_{\min,*}^2. \tag{210}$$

Then consider $T_2$, we have

$$T_2 = \frac{2\eta}{N} tr\left[ \sum_{k=1}^{K} \hat{N}_k \left( \hat{\mathbf{B}}^t \boldsymbol{\theta}_k^t - \hat{\mathbf{B}}^* \boldsymbol{\theta}_k^* \right) (\mathbf{F}_k^t)^T \mathbf{pp}^T (\hat{\mathbf{B}}^t)^T \right], \tag{211}$$

$$= \frac{2\eta}{N} tr\left[ \sum_{k=1}^{K} \hat{N}_k \left( \hat{\mathbf{B}}^t (\hat{\mathbf{B}}^t)^T \hat{\mathbf{B}}^* \boldsymbol{\theta}_k^* + \hat{\mathbf{B}}^t \mathbf{F}_k^t + \hat{\mathbf{B}}^t \mathbf{G}^t - \hat{\mathbf{B}}^* \boldsymbol{\theta}_k^* \right) (\mathbf{F}_k^t)^T \mathbf{pp}^T (\hat{\mathbf{B}}^t)^T \right], \tag{212}$$

$$= 2\eta tr\left[ (\hat{\mathbf{B}}^t)^T \hat{\mathbf{B}}^t \left( (\mathbf{F}^t)^T \mathbf{F}^t + (\mathbf{G}^t)^T \mathbf{F}^t \right) \mathbf{Wpp}^T \right], \tag{213}$$

$$\leq 2\eta \frac{\max_k N_k}{N} \left( \|\mathbf{F}^t\|_F^2 + \|\mathbf{F}^t\|_F \|\mathbf{G}^t\|_F \right), \tag{214}$$

$$\leq 2\eta \frac{\max_k N_k}{N} \left( \frac{\delta^2}{(1-\delta)^2}\bar{\sigma}_{\max,*}^2 + \frac{\delta^2}{(1-\delta)^2}\sqrt{K}\bar{\sigma}_{\max,*}\sigma^2 \right), \tag{215}$$

$$\leq \frac{\eta K}{100} \frac{\max_k N_k}{N} E_0 \bar{\sigma}_{\min,*}^2. \tag{216}$$

Finally, consider $T_3$, we have

$$T_3 = \frac{2\eta}{N} tr\left[ \sum_{k=1}^{K} \hat{N}_k \left( \hat{\mathbf{B}}^t \boldsymbol{\theta}_k^t - \hat{\mathbf{B}}^* \boldsymbol{\theta}_k^* \right) (\mathbf{G}_k^t)^T \mathbf{pp}^T (\hat{\mathbf{B}}^t)^T \right], \tag{217}$$

$$= \frac{2\eta}{N} tr\left[ \sum_{k=1}^{K} \hat{N}_k \left( \hat{\mathbf{B}}^t (\hat{\mathbf{B}}^t)^T \hat{\mathbf{B}}^* \boldsymbol{\theta}_k^* + \hat{\mathbf{B}}^t \mathbf{F}_k^t + \hat{\mathbf{B}}^t \mathbf{G}^t - \hat{\mathbf{B}}^* \boldsymbol{\theta}_k^* \right) (\mathbf{G}_k^t)^T \mathbf{pp}^T (\hat{\mathbf{B}}^t)^T \right], \tag{218}$$

$$= 2\eta tr\left[ (\hat{\mathbf{B}}^t)^T \hat{\mathbf{B}}^t \left( (\mathbf{G}^t)^T \mathbf{G}^t + (\mathbf{F}^t)^T \mathbf{G}^t \right) \mathbf{Wpp}^T \right], \tag{219}$$

$$\leq 2\eta \frac{\max_k N_k}{N} \left( \|\mathbf{G}^t\|_F^2 + \|\mathbf{F}^t\|_F \|\mathbf{G}^t\|_F \right), \tag{220}$$

$$\leq 2\eta \frac{\max_k N_k}{N} \left( \frac{\delta^2}{(1-\delta)^2}K\sigma^4 + \frac{\delta^2}{(1-\delta)^2}\sqrt{K}\bar{\sigma}_{\max,*}\sigma^2 \right), \tag{221}$$

$$\leq \frac{\eta K}{100} \frac{\max_k N_k}{N} E_0 \bar{\sigma}_{\min,*}^2. \tag{222}$$

Combining Equation (210), (216), and (222), we have

$$R_1 \leq 4\eta \left( \mathcal{C}_2 \frac{\sqrt{d+c}}{\sqrt{N}} \right) \left( (\sigma^2 + 1) \left( \sqrt{c} + \frac{1}{\sqrt{10}} \right)^2 \right) + \frac{11\eta K}{50} \frac{\max_k N_k}{N} E_0 \bar{\sigma}_{\min,*}^2. \tag{223}$$

Then consider $R_2$, because condition $\mathcal{E}_3$ holds, we have

$$R_2 = \frac{2\eta}{N} \max_{\|\mathbf{p}\|_2=1} \mathbf{p}^T (\hat{\mathbf{B}}^t)^T \mathbf{E}^t \mathbf{p}, \tag{224}$$

$$= \frac{2\eta}{N} \max_{\|\mathbf{p}\|_2=1} \mathbf{p}^T (\hat{\mathbf{B}}^t)^T \left( \sum_{k=1}^{K} (\bar{\mathbf{\Omega}}_k^T \odot \mathbf{X}^T)(\mathbf{Z} \odot \mathbf{\Omega}_k)(\boldsymbol{\theta}_k^t)^T \right) \mathbf{p}, \tag{225}$$

$$\leq 2\eta \left\| \frac{1}{N} \sum_{k=1}^{K} (\bar{\mathbf{\Omega}}_k^T \odot \mathbf{X}^T)(\mathbf{Z} \odot \mathbf{\Omega}_k)(\boldsymbol{\theta}_k^t)^T \right\|_2, \tag{226}$$

$$\leq 2\eta \mathcal{C}_1 \sigma^2 \frac{\sqrt{d+c}}{\sqrt{N}} \left( \sqrt{c} + \frac{1}{10} \right). \tag{227}$$

Then consider $R_3$, when condition $\mathcal{E}_3, \mathcal{E}_4, \mathcal{E}_5$, and $\mathcal{E}_6$ hold, we have

$$R_3 = \frac{\eta^2}{N^2} \max_{\|\mathbf{p}\|_2 = 1} \mathbf{p}^T (\mathbf{E}^t)^T \mathbf{S}^t \mathbf{p}, \tag{228}$$

$$\leq \frac{2\eta^2}{N} \left( \mathcal{C}_1 \sigma^2 \frac{\sqrt{d+c}}{\sqrt{N}} \left( \sqrt{c} + \frac{1}{10} \right) \right) \left( \mathcal{C}_2 \frac{\sqrt{d+c}}{\sqrt{N}} + 1 \right) \left( (\sigma^2 + 1) \left( \sqrt{c} + \frac{1}{\sqrt{10}} \right)^2 \right), \tag{229}$$

$$\leq \mathcal{C} \frac{2\eta^2 \sqrt{d+c}}{N^{3/2}} (\sigma^2 + 1)^2 \left( \sqrt{c} + \frac{1}{\sqrt{10}} \right)^3. \tag{230}$$

The last equation holds when $N$ satisfy $N \geq \frac{K^2}{d+c}$. Then combine Equation (192), (198), (227), and (230), because $N \geq (\sqrt{c} + 1)(\sigma^2 + 1)$ holds, we have

$$\sigma_{\min}^2(\mathbf{R}^{t+1}) \geq 1 - R_1 - R_2 - R_3, \tag{231}$$

$$\geq 1 - 8\eta \mathcal{C} \left( \frac{\sqrt{d+c}}{\sqrt{N}} \right) (\sigma^2 + 1) \left( \sqrt{c} + \frac{1}{\sqrt{10}} \right)^2 - \frac{11\eta K}{50} \frac{\max_k N_k}{N} E_0 \bar{\sigma}_{\min,*}^2. \tag{232}$$

Combining Equation (182) and (232), we have

$$dist\left(\hat{\mathbf{B}}^{t+1}, \hat{\mathbf{B}}^*\right) \leq dist\left(\hat{\mathbf{B}}^t, \hat{\mathbf{B}}^*\right)$$

$$\left( 1 - \eta K \left( \left( \frac{\min_k \hat{N}_k}{N} \right) - \frac{1}{4} \left( \frac{\max_k \hat{N}_k}{N} \right) \right) \bar{\sigma}_{\min,*}^2 E_0 + \eta \mathcal{C} \frac{\sqrt{d+c}}{\sqrt{N}} \left( \sqrt{c} + \frac{1}{\sqrt{10}} \right)^2 \right) \sigma_{\min}(\mathbf{R}^{t+1})^{-1}$$

$$+ \eta \mathcal{C} \frac{\sqrt{d+c}}{\sqrt{N}} (\sigma^2 + 1) \left( 2\sqrt{c} + \frac{1}{5} \right) \sigma_{\min}(\mathbf{R}^{t+1})^{-1}. \tag{233}$$

When $\frac{\max_k \hat{N}_k}{N}$ satisfy

$$\frac{\max_k \hat{N}_k}{N} \geq \frac{200}{7} \mathcal{C} \frac{\sqrt{d+c}}{\sqrt{N}} \frac{(\sigma^2 + 1)(\sqrt{c} + 1)^2}{K E_0 \bar{\sigma}_{\min,*}^2}, \tag{234}$$

we will have

$$\sigma_{\min}^2(\mathbf{R}^{t+1}) \geq 1 - \frac{1}{2} \eta K \left( \frac{\max_k \hat{N}_k}{N} \right) \bar{\sigma}_{\min,*}^2 E_0, \tag{235}$$

$$\eta \mathcal{C} \frac{\sqrt{d+c}}{\sqrt{N}} \left( \sqrt{c} + \frac{1}{\sqrt{10}} \right)^2 \leq \frac{7}{200} \eta K \left( \frac{\max_k \hat{N}_k}{N} \right) \bar{\sigma}_{\min,*}^2 E_0. \tag{236}$$

Then we have

$$dist\left(\hat{\mathbf{B}}^{t+1}, \hat{\mathbf{B}}^*\right) \leq dist\left(\hat{\mathbf{B}}^t, \hat{\mathbf{B}}^*\right)$$

$$\left( 1 - \eta K \left( \left( \frac{\min_k \hat{N}_k}{N} \right) - \frac{57}{200} \left( \frac{\max_k \hat{N}_k}{N} \right) \right) \bar{\sigma}_{\min,*}^2 E_0 \right) \left( 1 - \frac{1}{2} \eta K \left( \frac{\max_k \hat{N}_k}{N} \right) \bar{\sigma}_{\min,*}^2 E_0 \right)^{-1/2}$$

$$+ \left( \frac{7}{100} \eta K \left( \frac{\max_k \hat{N}_k}{N} \right) \bar{\sigma}_{\min,*}^2 E_0 \right) \left( 1 - \frac{1}{2} \eta K \left( \frac{\max_k \hat{N}_k}{N} \right) \bar{\sigma}_{\min,*}^2 E_0 \right)^{-1/2}. \tag{237}$$

$\square$

**Remark B.13.** *From Theorem B.12, we see that* $\mathrm{HCFL}^+$*'s convergence is strongly influenced by* $\max_k \hat{N}_k$ *and* $\min_k \hat{N}_k$*. In other words,* $\mathrm{HCFL}^+$ *converges faster as* $\frac{\min_k \hat{N}_k}{\max_k \hat{N}_k}$ *increases. This suggests that* $\mathrm{HCFL}^+$ *performs better when the number of samples is evenly distributed among all* $K$ *underlying clusters.*

## C  RELATED WORKS

**Federated Learning.**  As the de-facto algorithm in FL, FedAvg employs local SGD (McMahan et al., 2016; Lin et al., 2020) to reduce communication costs and protect client privacy. However, distribution shifts among clients pose a significant challenge in FL and hinder the performance of FL algorithms (Li et al., 2018; Wang et al., 2020; Karimireddy et al., 2020; Jiang & Lin, 2023; Guo et al., 2021). Traditional FL methods primarily aim to improve the convergence speed of global models and incorporate bias reduction techniques (Tang et al., 2022; Guo et al., 2023a; Li et al., 2021; 2018). At the same time, some studies investigate feature distribution shifts using domain generalization techniques (Peng et al., 2019; Wang et al., 2022a; Shen et al., 2021; Sun et al., 2022; Gan et al., 2021). However, single-model approaches are inadequate for handling heterogeneous data distributions, especially when dealing with concept shifts (Ke et al., 2022; Guo et al., 2023b; Jothimurugesan et al., 2023). To tackle these challenges, clustered FL algorithms are introduced to enhance FL algorithm performance.

**Clustered FL with fixed cluster numbers.**  Clustered FL groups clients based on their local data distribution, tackling the distribution shift problem. Most methods employ hard clustering with a fixed number of clusters, grouping clients by various similarity metrics, such as local loss values (Ghosh et al., 2020), local model parameter differences (Long et al., 2023), communication time/local calculation time (Wang et al., 2022b), and fuzzy $c$-Means (Stallmann & Wilbik, 2022). However, hard clustering may not capture complex relationships between local distributions adequately, and soft clustering paradigms have been proposed to address this issue. For instance, FedEM (Marfoq et al., 2021) employs Expectation-Maximization techniques to maximize likelihood functions. FedG-MMcitepwu2023personalized suggests using joint distributions instead of conditional distributions. FedRC(Guo et al., 2023b) introduces Robust Clustering, assigning clients with concept shifts to different clusters to enhance model generalization. FedSoft (Ruan & Joe-Wong, 2022) calculates weights based on the distances between clients' local model parameters and cluster model parameters, with smaller distances indicating larger weights for that cluster. In this paper, we propose a generalized formulation for clustered FL that encompasses the current methods and improves them by addressing issues related to intra-client inconsistency and efficiency.

**Clustered FL with adaptive clustering numbers.**  Another line of research focuses on automatically determining the number of clusters. Current methods utilize hierarchical clustering, which measures client dissimilarity using model parameters or local gradient distances. Most current methods modify cluster numbers by splitting them when client distances within clusters are large (Sattler et al., 2020b;a; Zhao et al., 2020; Briggs et al., 2020; Duan et al., 2021a;b). Recently, StoCFL (Zeng et al., 2023) suggests initially setting cluster numbers equal to the client count and merging clusters with small distances. In addition to model parameter distances, some papers employ alternative distance metrics for improved performance. For instance, Yan et al. (2023) employ principal eigenvectors of model parameters. Vahidian et al. (2023) use truncated singular value decomposition (SVD) to obtain a reduced set of principal vectors for distance measurement. Meanwhile, Wei & Huang (2023) focus on the distance of normalized local features. FEDCOLLAB (Bao et al., 2023) focuses on cross-silo scenarios with a limited number of clients and quantifies client similarity by training client discriminators. However, the need for discriminators between every pair of clients in FEDCOLLAB makes it challenging to expand to cross-device scenarios with numerous clients. In this paper, we concentrate on cross-device settings, introducing a holistic adaptive clustering framework enabling cluster splitting and merging. We also present enhanced weight updating for soft clustering and finer distance metrics for various clustering principles.

## D  ALGORITHMS

**Details of the HCFL$^+$.**  In Algorithm 2, we present a concise summary of the comprehensive algorithm that integrates all the enhanced components of HCFL$^+$, as introduced in Section 4. Specifically, during each communication round, the algorithm performs the following steps: (1) Randomly selects a subset of clients. (2) Calculates prototypes using Equations (241) and (242). (3) Performs local updates using Algorithm 3. (4) The server aggregates local updates, updates cluster model parameters, and computes client distance metrics using Equation (11) for each cluster $k$. (5) Identifies $k_{max}$ as the cluster with the highest average distance. (6) Checks if the maximum distance within $k_{max}$ significantly exceeds the average distance in this cluster. (7) If the following condition

is met, splits the clusters using Algorithm 4.

$$\max(D_{k_{max}}^t) - \text{mean}(D_{k_{max}}^t) \geq \rho\,. \tag{238}$$

(8) Mark and remove the empty clusters no clients will assign large clustering weights to using Algorithm 5.

**Intuitions on the distance metrics design.** From the objective function (Eq. (2)), we should assign higher clustering weights $\omega_{i,j;k}$ to clusters with greater $\mathcal{L}_k(\mathbf{x}_{i,j}, y_{i,j}, \boldsymbol{\phi}, \boldsymbol{\theta}_k)$ to maximize the objective function. Because the ultimate goals of the clustering algorithms are solving the objective functions, we analyse the $\mathcal{L}_k(\mathbf{x}_{i,j}, y_{i,j}, \boldsymbol{\phi}, \boldsymbol{\theta}_k)$ to check the key factors influencing the value of $\mathcal{L}_k(\mathbf{x}_{i,j}, y_{i,j}, \boldsymbol{\phi}, \boldsymbol{\theta}_k)$, and the relationships between these factors and the clustering principles.
We use the following algorithms as examples. For FedEM (Marfoq et al., 2021) and IFCA (Ghosh et al., 2020), $\mathcal{L}_k(\mathbf{x}, y, \boldsymbol{\phi}, \boldsymbol{\theta}_k) = \mathcal{P}_{\boldsymbol{\phi}, \boldsymbol{\theta}_k}(y|\mathbf{x})$; For FedRC (Guo et al., 2023b), $\mathcal{L}_k(\mathbf{x}, y, \boldsymbol{\phi}, \boldsymbol{\theta}_k) = \frac{\mathcal{P}_{\boldsymbol{\phi}, \boldsymbol{\theta}_k}(\mathbf{x}, y)}{\mathcal{P}_{\boldsymbol{\phi}, \boldsymbol{\theta}_k}(\mathbf{x})\mathcal{P}_{\boldsymbol{\phi}, \boldsymbol{\theta}_k}(y)}$. Defining $\mathbf{z} = g(\mathbf{x}; \boldsymbol{\phi})$ as the local features extracted by $\boldsymbol{\phi}$, assuming a $\mathbf{x} \to \mathbf{z} \to y$ Probabilistic Graphical Model (with $\mathbf{x}$ and $y$ being independent given $\mathbf{z}$), we obtain:

$$\mathcal{L}_k(\mathbf{x}, y, \boldsymbol{\phi}, \boldsymbol{\theta}_k) = \left\{ \begin{array}{ll} \mathcal{P}(y|\mathbf{x}; \boldsymbol{\phi}, \boldsymbol{\theta}_k) = \frac{\mathcal{P}(y|\mathbf{z}; \boldsymbol{\theta}_k)\mathcal{P}(\mathbf{z}|\mathbf{x}; \boldsymbol{\phi})}{\mathcal{P}(\mathbf{z}|\mathbf{x}, y; \boldsymbol{\phi})} & \text{(FedEM, IFCA)} \\ \frac{\mathcal{P}(\mathbf{x}, y; \boldsymbol{\phi}, \boldsymbol{\theta}_k)}{\mathcal{P}(y; \boldsymbol{\phi}, \boldsymbol{\theta}_k)\mathcal{P}(\mathbf{x}; \boldsymbol{\phi}, \boldsymbol{\theta}_k)} = \frac{\mathcal{P}(y|\mathbf{z}; \boldsymbol{\theta}_k)\mathcal{P}(\mathbf{z}|\mathbf{x}; \boldsymbol{\phi})}{\mathcal{P}(y; \boldsymbol{\phi}, \boldsymbol{\theta}_k)\mathcal{P}(\mathbf{z}|\mathbf{x}, y; \boldsymbol{\phi})} & \text{(FedRC)} \end{array} \right\} = \frac{\tilde{\mathcal{L}}_k(\mathbf{z}, y; \boldsymbol{\theta}_k)\mathcal{P}(\mathbf{z}|\mathbf{x}; \boldsymbol{\phi})}{\mathcal{P}(\mathbf{z}|\mathbf{x}, y; \boldsymbol{\phi})}\,.$$

Then we aim to give the following explanations of the three terms $\mathcal{P}(\mathbf{z}|\mathbf{x}; \boldsymbol{\phi})$, $\mathcal{P}(\mathbf{z}|\mathbf{x}; \boldsymbol{\phi})$, and $\tilde{\mathcal{L}}_k(\mathbf{z}, y; \boldsymbol{\theta}_k)$, which align with the terms considered in Sec 4.4.

- $\mathcal{P}(\mathbf{z}|\mathbf{x}; \boldsymbol{\phi})$ **for feature and label shifts**. Feature shifts introduce significant distances in $\mathbf{x}$. Additionally, $\mathbf{x}$ with different $y$ values generally exhibit substantial distances in the feature space. Without this, classifiers cannot distinguish samples with different labels. Hence, we employ $\mathcal{P}(\mathbf{z}|\mathbf{x}; \boldsymbol{\phi})$ to assess both feature and label shifts.
- $\mathcal{P}(\mathbf{z}|\mathbf{x}, y; \boldsymbol{\phi})$ **for concept shifts.** Concept shifts signify alerted $\mathbf{x} - y$ correlations. Hence, samples with concept shifts but have the same $y$ should exhibit a significant difference in $\mathcal{P}(\mathbf{z}|\mathbf{x}, y; \boldsymbol{\phi})$.
- $\tilde{\mathcal{L}}_k(\mathbf{z}, y; \boldsymbol{\theta}_k)$ **for the quality of clustering.** The $\tilde{\mathcal{L}}_k(\mathbf{z}, y; \boldsymbol{\theta}_k)$ is defined using features $\mathbf{z} = g(\mathbf{x}; \boldsymbol{\phi})$ instead of data $\mathbf{x}$ in $\mathcal{L}_k(\mathbf{x}, y, \boldsymbol{\phi}, \boldsymbol{\theta}_k)$. This term evaluates if features $\mathbf{z}$ can be correctly assigned to clusters given the current $\boldsymbol{\Theta}$; otherwise, the objectives in (2) cannot be achieved.

Finally, we propose the following distance metric:

$$\mathbf{D}_{i,j}^k = \left\{ \begin{array}{ll} \max\{d_c, d_{lf}\}\, \mathbb{E}_{D_i}\left[\tilde{\mathcal{L}}_k(\mathbf{z}, y; \boldsymbol{\theta}_k)\right] \mathbb{E}_{D_j}\left[\tilde{\mathcal{L}}_k(\mathbf{z}, y; \boldsymbol{\theta}_k)\right]\,, & \text{ASCP} \\ d_c \mathbb{E}_{D_i}\left[\tilde{\mathcal{L}}_k(\mathbf{z}, y; \boldsymbol{\theta}_k)\right] \mathbb{E}_{D_j}\left[\tilde{\mathcal{L}}_k(\mathbf{z}, y; \boldsymbol{\theta}_k)\right]\,, & \text{CSCP} \end{array} \right. \tag{239}$$

where dist is the cos-similarity in this paper, $d_c = \max_y \left\{ \text{dist}\left(\mathbb{E}_{D_i}\left[\mathcal{P}(\mathbf{z}|\mathbf{x}, y; \boldsymbol{\phi})\right], \mathbb{E}_{D_j}\left[\mathcal{P}(\mathbf{z}|\mathbf{x}, y; \boldsymbol{\phi})\right]\right) \right\}$, and $d_{lf} = \text{dist}\left(\mathbb{E}_{D_i}\left[\mathcal{P}(\mathbf{z}|\mathbf{x}; \boldsymbol{\phi})\right], \mathbb{E}_{D_j}\left[\mathcal{P}(\mathbf{z}|\mathbf{x}; \boldsymbol{\phi})\right]\right)$. The distances above become large only when the following conditions occur together: (1) Large values of $d_c$ indicate concept shifts between clients $i$ and $j$; (2) Large $d_{lf}$ indicate significant feature and label distribution differences. (2) Large values of $\mathbb{E}_{D_i}\left[\tilde{\mathcal{L}}_k(\mathbf{z}, y; \boldsymbol{\theta}_k)\right] \mathbb{E}_{D_j}\left[\tilde{\mathcal{L}}_k(\mathbf{z}, y; \boldsymbol{\theta}_k)\right]$ indicate incorrect clustering weights with high confidence.

**Approximation of the distance metrics in practice.** When calculating the distance metrics (Equation (11)) in practice, to avoid training extra generative networks and transmitting more data between servers and clients, we substitute $\tilde{\omega}_{i;k}$ for $\tilde{\mathcal{L}}_k(\mathbf{z}, y; \boldsymbol{\theta}_k)$ since $\tilde{\omega}_{i;k}$ is positively correlated with $\tilde{\mathcal{L}}_k(\mathbf{z}, y; \boldsymbol{\theta}_k)$ (Marfoq et al., 2021; Guo et al., 2023b). Additionally, we approximate $\mathbb{E}_{D_i}\left[\mathcal{P}(\mathbf{z}|\mathbf{x}, y; \boldsymbol{\phi})\right]$ and $\mathbb{E}_{D_i}\left[\mathcal{P}(\mathbf{z}|\mathbf{x}; \boldsymbol{\phi})\right]$ using feature prototypes. The prototypes are defined by the following equation:

$$\tilde{d}_c = Dist(\mathbf{P}_{c,i}, \mathbf{P}_{c,j})\,, \tilde{d}_{lf} = Dist(\mathbf{P}_{lf,i}, \mathbf{P}_{lf,j})\,, \tag{240}$$

where

$$\mathbf{P}_{c,i} \in \mathbb{R}^{d \times C} = [\frac{1}{N_{i,1}}\sum_{j=1}^{N_i}\mathbf{1}_{y_{i,j}=1}g(\mathbf{x}_{i,j}, \boldsymbol{\phi}), \cdots, \frac{1}{N_{i,C}}\sum_{j=1}^{N_i}\mathbf{1}_{y_{i,j}=C}g(\mathbf{x}_{i,j}, \boldsymbol{\phi})]\,, \tag{241}$$

$$\mathbf{P}_{lf,i} \in \mathbb{R}^d = \frac{1}{N_i}\sum_{j=1}^{N_i}g(\mathbf{x}_{i,j}, \boldsymbol{\phi})\,, \tag{242}$$

$N_{i,c} = \sum_{j=1}^{N_i}\mathbf{1}_{y_{i,j}=c}$, $g(\mathbf{x}_{i,j}, \boldsymbol{\phi})$ is the function parameterized by $\boldsymbol{\phi}$, $Dist$ is a function to measure the distance between prototypes, which we use the cosine similarity as an example in this paper.

---

**Algorithm 2** Algorithm Framework of HCFL$^+$

---

**Require:** Local datasets $D_1, \ldots, D_N$, number of local iterations $\mathcal{T}$, number of communication rounds $T$, number of clients chosen in each round $S$, initial number of clusters $K^0$, number of classes $C$, and hyper-parameter $\rho$.

**Ensure:** Trained global feature extractor $\phi^T$, final number of clusters $K^T$, and cluster-specific predictors $\Theta^T = [\theta_1^T, \cdots, \theta_{K^T}^T]$.

1: Initialize $\phi^0, \Theta^0 = [\theta_1^0, \cdots, \theta_{K^0}^0]$.
2: **for** $t = 0, \ldots, T-1$ **do**
3:    Choose a subset of clients $\mathcal{S}^t$, where $|\mathcal{S}^t| = S$.
4:    **for** chosen client $i \in \mathcal{S}^t$ **do**
5:       Calculate client prototypes $\mathbf{P}_i^t$ by Equation (241)- (242).
6:       $\mathcal{F}_i^{t+1}, \tilde{\omega}_{i;k}^{t+1}, \phi_i^{\mathcal{T}}, \theta_{k,i}^{\mathcal{T}} \leftarrow$ Local updates by Algorithm 3.
7:       Send $\mathbf{P}_i^t, \mathcal{F}_i^{t+1}$, and $\tilde{\omega}_{i;k}^{t+1}, \phi_i^{\mathcal{T}}, \theta_{k,i}^{\mathcal{T}}, \forall k \leq K^t$ to the server.
8:    $\mathcal{F}_i^{latest} \leftarrow \mathcal{F}_i^{t+1}$.
9:    $\phi^{t+1} = \frac{1}{\sum_{i \in \mathcal{S}^t} N_i} \sum_{i \in \mathcal{S}^t} N_i \phi_i^{\mathcal{T}}$.
10:    $\theta_k^{t+1} = \frac{1}{\sum_{i \in \mathcal{S}^t} N_i} \sum_{i \in \mathcal{S}^t} N_i \theta_{k,i}^{\mathcal{T}}, \forall k \leq K^t$.
11:    $\mathcal{F}_g^{t+1} \leftarrow [\mathcal{F}_{g,1}^{t+1}, \cdots, \mathcal{F}_{g,K^t}^{t+1}]$, where $\mathcal{F}_{g,k}^{t+1} \leftarrow [\sum_i \mathcal{F}_{i,k,1}^{latest}, \cdots, \sum_i \mathcal{F}_{i,k,C}^{latest}]$.
12:    Initialize $\mathcal{C}_k^t = \emptyset, \forall k \leq K^t$.
13:    **for** all client $i$ **do**
14:       $c_i \leftarrow \arg\max_k \tilde{\omega}_{i;k}$.
15:       $\mathcal{C}_{c_i}^t \leftarrow \mathcal{C}_{c_i}^t \cup i$.
16:    $\mathcal{R}^t \leftarrow \emptyset$.
17:    **for** $k \leq K^t$ **do**
18:       **if** $\mathcal{C}_k^t$ is empty **then**
19:          $\mathcal{R}^t \leftarrow \mathcal{R}^t \cup k$.
20:       Get the cluster-specific distance matrix $\mathbf{D}_k \in \mathbb{R}^{|\tilde{\mathcal{S}}_k^t| \times |\tilde{\mathcal{S}}_k^t|}, \forall k \leq K^t$ by Equation (11).
21:    $k_{min} \leftarrow \arg\max_k \max(\mathbf{D}_k^t)$.
22:    **if** $\max(\mathbf{D}_{k_{min}}^t) - mean(\mathbf{D}_{k_{min}}^t) \geq \rho$ **then**
23:       Split $\mathcal{C}_{k_{min}}^t$ into two clusters $\mathcal{C}_{k_{min},1}^t$ and $\mathcal{C}_{k_{min},2}^t$.
24:       $\theta_{k_{min}}^{t+1} = \frac{1}{\sum_{i \in \mathcal{C}_{k_{min},1}^t} N_i} \sum_{i \in \mathcal{C}_{k_{min},1}^t} N_i \theta_{k_{min},i}^{\mathcal{T}}$.
25:       Add new cluster and update $\mathcal{F}_g$ by server side of Algorithm 4.
26:       $K^{t+1} \leftarrow K^t + 1$.
27:    **else**
28:       $K^{t+1} \leftarrow K^t$.
29:    **for** cluster $k_r \in \mathcal{R}^t$ **do**
30:       Remove cluster $k_r$ and update $\mathcal{F}_g$ by server side of Algorithm 5.
31:       $K^{t+1} \leftarrow K^{t+1} - 1$.
32:    Send $\phi^{t+1}, \Theta^{t+1} = [\theta_1^{t+1}, \cdots, \theta_{K^{t+1}}^{t+1}]$, and information about add/remove cluster to clients.

---

**Algorithm 3** Local Updates of HCFL$^+$

---

**Require:** Number of local iterations $\mathcal{T}$, current number of clusters $K^t$, number of classes $C$, local dataset $D_i$, global feature extractor $\phi^t$, cluster-specific predictors $\Theta^t = [\theta_1^t, \cdots, \theta_{K^t}^t]$.

**Ensure:** Trained feature extractor $\phi_i^{\mathcal{T}}$, predictors $\Theta_i^{\mathcal{T}} = [\theta_{i,1}^{\mathcal{T}}, \cdots, \theta_{i,K^t}^{\mathcal{T}}]$, $\tilde{\Omega}_i^{t+1} = [\tilde{\omega}_{i;k}, \cdots, \tilde{\omega}_{i;K^t}]$, and $\mathcal{F}_i^{t+1} = [\mathcal{F}_{i,1}^{t+1}, \cdots, \mathcal{F}_{i,K^t}^{t+1}]$, where $\mathcal{F}_{i,k}^{t+1} = [\mathcal{F}_{i,k,1}^{t+1}, \cdots, \mathcal{F}_{i,k,C}^{t+1}]$.

1: Update $\gamma_{i,j;k}^{t+1}, \tilde{\gamma}_{i,j;k}^{t+1}, \omega_{i,j;k}^{t+1}, \tilde{\omega}_{i;k}^{t+1}$ by Equations (4)-(5), $\forall j \leq N_i, k \leq K^t$.      ▷ Tier 2
2: **for** $\tau = 1, \ldots, \mathcal{T}$ **do**      ▷ Tier 1
3:    Update $\theta_{k,i}^{\tau}$ by Equation (6), $\forall k \leq K^t$.
4:    Update $\phi_i^{\tau}$ by Equation (7).
5: $\mathcal{F}_{i,k,c}^{t+1} \leftarrow \sum_{j=1}^{N_i} \mathbf{1}_{y_{i,j}=c} \gamma_{i,j;k}^{t+1}$.

---

---

**Algorithm 4** Cluster Adding of HCFL$^+$

---

**Require:** $k_{min}$, set of clients $\mathcal{C}^t_{k_{min},2}$, the corresponding $\boldsymbol{\theta}^{\mathcal{T}}_{k_{min},i}$ for each client $i \in \mathcal{C}^t_{k_{min},2}$, and $\mathcal{F}_g$.
**Ensure:** New $\mathcal{F}^{t+1}_g$, and predictor of the new cluster $\boldsymbol{\theta}^{t+1}_{K^t+1}$.
1: **Server Side:**
2: $\boldsymbol{\theta}^{t+1}_{K^t+1} = \frac{1}{\sum_{i \in \mathcal{C}^t_{k_{min},2}}} \sum_{i \in \mathcal{C}^t_{k_{min},2}} N_i \boldsymbol{\theta}^{\mathcal{T}}_{k_{min},i}$.
3: Add $\mathcal{F}_{g,K^t+1} \leftarrow \mathcal{F}_{g,k_{min}}$ to $\mathcal{F}^{t+1}_g$.
4: Add $\mathcal{F}^{latest}_{i,K^t+1} \leftarrow \mathcal{F}^{latest}_{i,k_{min}}$ to $\mathcal{F}^{latest}_i, \forall i$.
5: **Client Side:**
6: $\omega_{i,j;K^t+1} \leftarrow \omega_{i,j;k_{min}}/2, \forall j \leq N_i$.
7: $\omega_{i,j;k_{min}} \leftarrow \omega_{i,j;k_{min}}/2, \forall j \leq N_i$.
8: $\tilde{\omega}_{i;K^t+1} \leftarrow \tilde{\omega}_{i;k_{m}in}/2$.
9: $\tilde{\omega}_{i;k_{min}} \leftarrow \tilde{\omega}_{i;k_{m}in}/2$.

---

**Algorithm 5** Cluster Removing of HCFL$^+$

---

**Require:** The cluster needs to be removed $k_r$, and $\mathcal{F}^{t+1}_g$.
1: **Server Side:**
2: Remove $\mathcal{F}^{t+1}_{g,k_r}$ from $\mathcal{F}^{t+1}_g$.
3: Remove $\mathcal{F}^{latest}_{i,k_r}$ from $\mathcal{F}^{latest}_i, \forall i$.
4: **Client Side:**
5: $\omega_{i,j;k} \leftarrow \frac{\omega_{i,j;k}}{\sum_{n \neq k_r} \omega_{i,j;n}}, \forall j \leq N_i, k \neq k_r$.
6: $\tilde{\omega}_{i;k} \leftarrow \frac{\omega_{i;k}}{\sum_{n \neq k_r} \omega_{i;n}}, \forall k \neq k_r$.
7: Remove $\gamma_{i,j;k_r}, \tilde{\gamma}_{i,j;k_r}, \omega_{i,j;k_r}, \tilde{\omega}_{i,j;k_r}, \forall j \leq N_i$.

---

# E  EXPERIMENT RESULTS

## E.1  DATASETS AND MODELS

**Diverse distribution shifts scenarios.**    Similar to previous work (Guo et al., 2023b), the diverse distribution shift scenario construct clients with three types of distribution shifts with each other:
- **Label Distribution Shifts:** We use the idea introduced Yoshida et al. (2019); Hsu et al. (2019); Reddi et al. (2021), where we leverage the Latent Dirichlet Allocation (LDA) with $\alpha = 1.0$. We split datasets to 100 clients by default.
- **Feature Distribution Shifts:** We utilize the idea of constructing CIFAR10-C, CIFAR100-C, and ImageNet-C (Hendrycks & Dietterich, 2019). In detail, we apply random augmentations to client samples, selecting from 20 types, including 'Original', 'Gaussian Noise', 'Shot Noise', 'Impulse Noise', 'Defocus Blur', 'Glass Blur', 'Motion Blur', 'Zoom Blur', 'Snow', 'Frost', 'Fog', 'Brightness', 'Contrast', 'Elastic', 'Pixelate', 'JPEG', 'Speckle Noise', 'Gaussian Blur', 'Spatter', and 'Saturate'. Augmentation types remain consistent within each client.
- **Concept Shifts:** For label $y \le C_\beta$, it becomes $y$, $(1 + y)\% C_\beta$, and $(2 + y)\% C_\beta$ across concepts, where $C_\beta = \lfloor C * \beta \rfloor$, and $C$ is the number of classes.

**Noisy label scenarios.**    We follow the methodology of previous works (Fang & Ye, 2022; Ke et al., 2022) to construct noisy label scenarios. Our approach involves two types of noisy labels: symmetric flip and pair flip. Symmetric flip entails randomly flipping the original class label to any wrong class label with equal probability. Pair flip involves flipping the original class label only to a very similar wrong category. We use the parameter $\chi$ to control the noisy rate, where $\chi = 0.1$ indicates that $10\%$ of the data have wrong labels.

## E.2  BASELINES AND HYPER-PARAMETER SETTINGS

**Detailed implementations and hyper-parameter settings for all the algorithms**    Unless special mentioned, we split each dataset to 100 clients with 3 concepts. The learning rates are chosen in $\{0.03, 0.06, 1.0\}$, and we report the best results for each algorithm. We run the algorithms for 200 communication rounds and set the number of local epochs to 1.

**Detailed implementations and hyper-parameter settings of baseline algorithms.**    The details of the settings and hyper-parameters we used for the baseline methods a summarized below. We exclude the algorithms that do not require additional hyper-parameters here.
- **CFL** (Sattler et al., 2020b). We use the public code provided by (Marfoq et al., 2021) for the CFL algorithm. The hyper-parameters $tol_1$ and $tol_2$ are tuned, and we report how the hyper-parameters affect the results of the algorithm in Table 1.
- **ICFL** (Yan et al., 2023). Follow the same setting as the original paper, we set the hyper-parameter $\alpha * (0)$ to $\{0.85, 0.98\}$, and $\epsilon_1 = 4.0$.
- **stoCFL** (Zeng et al., 2023). We choose $\tau = \{0, 0.05, 0.1, 0.15\}$ to control the trade-off between personalization and generalization as suggested by the original paper. In addition, we choose $\lambda = 0.5$, which always achieve the best performance as reported in the original paper.
- **HCFL$^+$ (FedRC)** (Guo et al., 2023b). We set $\tilde{\mu} = 0.4$, and choose $\rho = \{0.05, 0.1, 0.3\}$. The distance between clients are calculated by Equation (**??**).
- **HCFL$^+$ (FedEM)** (Marfoq et al., 2021). We set $\tilde{\mu} = 0.4$, and choose $\rho = \{0.05, 0.1, 0.3\}$. The distance between clients are calculated by Equation (11).
- **HCFL$^+$ (FeSEM)** (Long et al., 2023). We choose $\rho = \{0.05, 0.1, 0.3\}$. Follow the original paper, we use hard clustering paradigms that does not require the hyper-parameter $\tilde{\mu}$. The model splitting process is the same as Sattler et al. (2020b) that designed for hard clustering paradigms. The distance between clients are calculated by Equation (11).

## E.3  ADDITIONAL EXPERIMENT RESULTS

**Results on noisy data scenarios**    In Table 3, we show the performance of clustered FL emthods on noisy data scenarios. Results show that HCFL$^+$ consistently outperform other methods by a large margin.

**Additional results on diverse distribution shift scenarios.**    In Table 4, we show the performance of algorithms with $\beta = 0.4$. Results show HCFL$^+$ always achieve the best test accuracy, and achieve a good local-global balance.

Table 3: **Performance of algorithms on noisy data scenarios.** We evaluated the performance of algorithms using the CIFAR10 dataset split into 100 clients. For each algorithm, we report the best test accuracy for all 200 communication rounds.

| Algorithm | CIFAR10 (MobileNetV2) | | | |
| --- | --- | --- | --- | --- |
| | *Pairflip*, $\chi = 0.1$ | *Pairflip*, $\chi = 0.2$ | *Symflip*, $\chi = 0.2$ | *Symflip*, $\chi = 0.4$ |
| FedAvg | 54.75 $\pm 1.45$ | 52.35 $\pm 1.65$ | 52.60 $\pm 0.50$ | 41.80 $\pm 0.50$ |
| FeSEM | 32.60 $\pm 1.30$ | 35.25 $\pm 2.95$ | 32.40 $\pm 2.80$ | 29.70 $\pm 0.01$ |
| IFCA | 24.95 $\pm 7.05$ | 20.55 $\pm 4.65$ | 30.35 $\pm 2.05$ | 36.05 $\pm 4.45$ |
| FedEM | 64.40 $\pm 1.10$ | 57.55 $\pm 2.95$ | 53.00 $\pm 1.90$ | 43.10 $\pm 0.20$ |
| FedRC | 67.90 $\pm 1.00$ | 59.95 $\pm 1.05$ | 55.25 $\pm 2.05$ | 42.00 $\pm 0.40$ |
| HCFL$^+$ | **66.70** $\pm 0.40$ | **62.70** $\pm 0.30$ | **59.95** $\pm 1.15$ | **47.20** $\pm 0.20$ |

Table 4: **Performance of the adaptive clustering methods.** We evaluated algorithm performance on CIFAR10 and CIFAR100 datasets, employing 100 clients. For each algorithm, we present the highest validation and test accuracies across 200 communication rounds, and the final number of clusters during training denoted as $K^T$. All experiments utilized MobileNet-V2 (Sandler et al., 2018).

| Algorithm | CIFAR10, $\beta = 0.4$ | | | CIFAR100, $\beta = 0.4$ | | |
| --- | --- | --- | --- | --- | --- | --- |
| | Val | Test | $K^T$ | Val | Test | $K^T$ |
| FedAvg | 48.16 $\pm 1.64$ | 49.93 $\pm 0.80$ | 3.0 | 22.77 $\pm 0.01$ | 24.62 $\pm 0.55$ | 3.0 |
| FeSEM | 46.08 $\pm 4.54$ | 35.99 $\pm 4.59$ | 3.0 | 23.56 $\pm 1.52$ | 22.31 $\pm 1.08$ | 3.0 |
| IFCA | 36.15 $\pm 3.45$ | 24.79 $\pm 1.18$ | 3.0 | 27.72 $\pm 0.82$ | 21.37 $\pm 1.33$ | 3.0 |
| FedEM | 60.26 $\pm 1.10$ | 54.44 $\pm 0.04$ | 3.0 | 25.80 $\pm 0.20$ | 22.88 $\pm 0.19$ | 3.0 |
| FedRC | 57.99 $\pm 0.29$ | 56.75 $\pm 0.38$ | 3.0 | 30.94 $\pm 0.88$ | 31.63 $\pm 0.20$ | 3.0 |
| CFL | | | | | | |
| $\text{tol}_1 = 0.4, \text{tol}_2 = 1.6$ | 61.86 $\pm 5.29$ | 51.15 $\pm 0.82$ | 6.0 | 34.11 $\pm 6.35$ | 21.04 $\pm 2.21$ | 5.0 |
| $\text{tol}_1 = 0.4, \text{tol}_2 = 0.8$ | 60.42 $\pm 0.31$ | 41.59 $\pm 2.14$ | 8.0 | 36.23 $\pm 3.58$ | 16.03 $\pm 2.69$ | 6.0 |
| $\text{tol}_1 = 0.2, \text{tol}_2 = 0.8$ | 49.14 $\pm 6.11$ | 49.88 $\pm 4.21$ | 3.0 | 34.20 $\pm 7.13$ | 26.42 $\pm 0.73$ | 2.5 |
| ICFL | | | | | | |
| $\alpha^*(0) = 0.85$ | 77.73 $\pm 0.47$ | 52.03 $\pm 0.10$ | 100.0 | 49.71 $\pm 0.55$ | 28.55 $\pm 0.03$ | 100.0 |
| $\alpha^*(0) = 0.98$ | 63.69 $\pm 3.58$ | 54.02 $\pm 1.11$ | 81.5 | 45.72 $\pm 1.10$ | 28.45 $\pm 0.82$ | 70.0 |
| StoCFL | | | | | | |
| $\tau = 0.00$ | 48.55 $\pm 0.95$ | 51.25 $\pm 1.16$ | 1.5 | 24.50 $\pm 0.03$ | 25.70 $\pm 1.51$ | 1.0 |
| $\tau = 0.05$ | 57.84 $\pm 2.26$ | 50.42 $\pm 0.97$ | 20.5 | 26.24 $\pm 1.46$ | 26.60 $\pm 1.17$ | 4.0 |
| $\tau = 0.10$ | 72.91 $\pm 2.25$ | 47.84 $\pm 2.60$ | 59.0 | 67.67 $\pm 1.68$ | 9.89 $\pm 0.45$ | 86.0 |
| $\tau = 0.15$ | 77.19 $\pm 2.31$ | 41.49 $\pm 0.97$ | 92.0 | 70.13 $\pm 0.27$ | 7.77 $\pm 0.23$ | 94.0 |
| HCFL$^+$ (FeSEM) | | | | | | |
| $\rho = 0.1$ | 85.30 $\pm 1.05$ | 45.20 $\pm 0.28$ | 47.0 | 58.61 $\pm 4.14$ | 18.29 $\pm 2.38$ | 35.5 |
| $\rho = 0.3$ | 80.34 $\pm 1.33$ | 48.25 $\pm 2.72$ | 20.5 | 44.65 $\pm 0.35$ | 21.73 $\pm 1.27$ | 12.0 |
| HCFL$^+$ (FedEM) | | | | | | |
| $\rho = 0.05$ | 80.31 $\pm 1.60$ | 53.62 $\pm 4.36$ | 18.5 | 62.19 $\pm 1.54$ | 21.15 $\pm 0.88$ | 44.5 |
| $\rho = 0.1$ | 82.89 $\pm 0.92$ | 56.27 $\pm 1.08$ | 26.5 | 59.08 $\pm 0.06$ | 21.29 $\pm 0.87$ | 31.5 |
| $\rho = 0.3$ | 80.72 $\pm 1.90$ | 55.77 $\pm 1.93$ | 10.0 | 49.84 $\pm 6.85$ | 28.62 $\pm 0.78$ | 11.0 |
| HCFL$^+$ (FedRC) | | | | | | |
| $\rho = 0.05$ | 68.48 $\pm 0.25$ | 66.77 $\pm 0.28$ | 9.5 | 38.75 $\pm 0.98$ | 30.45 $\pm 0.07$ | 10.0 |
| $\rho = 0.1$ | 68.56 $\pm 3.56$ | 65.75 $\pm 5.40$ | 6.0 | 40.30 $\pm 1.19$ | 30.23 $\pm 0.85$ | 11.0 |
| $\rho = 0.3$ | 70.86 $\pm 0.31$ | 70.13 $\pm 0.42$ | 5.5 | 39.62 $\pm 0.34$ | 32.22 $\pm 0.20$ | 5.0 |

Table 5: **Ablation studies on techniques in Sec 4.2.** We evaluated algorithm performance on CIFAR10 and CIFAR100 datasets, showcasing the top Validation and Test accuracies for each. We kept $\rho = 0.3$ consistent across all algorithms and varied $\tilde{\mu}$ to adjust the penalty term's strength in the objective function. The best results in each block are highlighted.

| Algorithm | CIFAR10, $\beta = 0.2$ | | CIFAR10, $\beta = 0.4$ | | CIFAR100, $\beta = 0.2$ | | CIFAR100, $\beta = 0.4$ | |
|---|---|---|---|---|---|---|---|---|
| | Val | Test | Val | Test | Val | Test | Val | Test |
| HCFL$^+$ (FedEM) | | | | | | | | |
| $\tilde{\mu} = 0.0$ | **83.67** $\pm 0.72$ | **62.43** $\pm 0.71$ | **80.72** $\pm 1.90$ | **55.77** $\pm 1.93$ | **50.72** $\pm 2.97$ | **32.13** $\pm 0.18$ | **49.84** $\pm 6.85$ | **28.62** $\pm 0.78$ |
| $\tilde{\mu} = 0.1$ | 81.60 $\pm 0.59$ | 60.48 $\pm 0.50$ | 80.36 $\pm 2.40$ | 55.10 $\pm 1.75$ | 48.78 $\pm 0.62$ | 30.50 $\pm 0.33$ | 48.56 $\pm 1.10$ | 25.80 $\pm 1.17$ |
| $\tilde{\mu} = 0.4$ | 79.52 $\pm 0.11$ | 53.33 $\pm 2.97$ | 76.50 $\pm 0.34$ | 49.97 $\pm 2.26$ | 44.85 $\pm 0.48$ | 28.39 $\pm 0.12$ | 41.52 $\pm 0.08$ | 22.83 $\pm 0.42$ |
| HCFL$^+$ (FedRC) | | | | | | | | |
| $\tilde{\mu} = 0.0$ | **70.82** $\pm 0.25$ | 69.15 $\pm 0.35$ | 69.95 $\pm 1.99$ | 67.09 $\pm 1.01$ | 39.55 $\pm 1.29$ | 35.49 $\pm 0.16$ | 38.77 $\pm 1.20$ | 31.87 $\pm 1.13$ |
| $\tilde{\mu} = 0.1$ | 69.91 $\pm 0.16$ | 68.77 $\pm 1.56$ | 69.53 $\pm 0.21$ | 68.54 $\pm 1.08$ | 39.38 $\pm 0.40$ | 35.95 $\pm 0.59$ | **39.77** $\pm 2.33$ | 31.52 $\pm 0.45$ |
| $\tilde{\mu} = 0.4$ | 69.33 $\pm 0.24$ | **69.67** $\pm 1.27$ | **70.86** $\pm 0.31$ | **70.13** $\pm 0.42$ | **39.97** $\pm 0.21$ | **36.50** $\pm 0.28$ | 39.62 $\pm 0.34$ | **32.22** $\pm 0.20$ |

Table 6: **Ablation studies on techniques in Sec 4.3.** We evaluated algorithm performance on CIFAR10 and CIFAR100 datasets, displaying their highest Validation and Test accuracies. We kept $\rho$ consistent at 0.3 for all algorithms. "w/ SCWU" denotes the use of soft clustering weight updating mechanisms designed in Section 4.3.

| Algorithm | CIFAR10, $\beta = 0.2$ | | CIFAR10, $\beta = 0.4$ | | CIFAR100, $\beta = 0.2$ | | CIFAR100, $\beta = 0.4$ | |
|---|---|---|---|---|---|---|---|---|
| | Val | Test | Val | Test | Val | Test | Val | Test |
| HCFL$^+$ (FedEM) | | | | | | | | |
| w/ SCWU | **83.67** $\pm 0.72$ | 62.43 $\pm 0.71$ | **80.72** $\pm 1.90$ | 55.77 $\pm 1.93$ | **50.72** $\pm 2.97$ | 32.13 $\pm 0.18$ | **49.84** $\pm 6.85$ | 28.62 $\pm 0.78$ |
| w/o SCWU | 82.11 $\pm 2.39$ | 63.84 $\pm 0.19$ | 80.08 $\pm 0.99$ | 58.83 $\pm 2.12$ | 49.77 $\pm 1.93$ | 32.90 $\pm 1.11$ | 47.91 $\pm 2.67$ | 27.40 $\pm 1.17$ |
| HCFL$^+$ (FedRC) | | | | | | | | |
| w/ SCWU | 69.33 $\pm 0.24$ | **69.67** $\pm 1.27$ | 70.86 $\pm 0.31$ | **70.13** $\pm 0.42$ | 39.97 $\pm 0.21$ | **36.50** $\pm 0.28$ | **39.62** $\pm 0.34$ | **32.22** $\pm 0.20$ |
| w/o SCWU | 69.88 $\pm 0.30$ | 68.83 $\pm 0.71$ | 70.77 $\pm 0.47$ | 68.87 $\pm 0.23$ | 40.96 $\pm 1.24$ | 35.72 $\pm 1.01$ | 39.18 $\pm 0.13$ | 32.08 $\pm 0.78$ |

Table 7: **Performance of algorithms with Resnet18.** We evaluated algorithm performance on CIFAR10 datasets with $\beta = 0.2$, displaying their highest Validation and Test accuracies. All algorithms utilize ResNet18 and run for 200 communication rounds.

| Algorithm | Val | Test |
|---|---|---|
| CFL | | |
| $\text{tol}_1 = 0.4, \text{tol}_2 = 0.6$ | 63.07 $\pm 7.42$ | 53.65 $\pm 2.33$ |
| $\text{tol}_1 = 0.4, \text{tol}_2 = 0.8$ | 61.14 $\pm 1.87$ | 54.87 $\pm 1.32$ |
| ICFL | | |
| $\alpha^*(0) = 0.85$ | 80.46 $\pm 0.99$ | 45.28 $\pm 6.56$ |
| $\alpha^*(0) = 0.98$ | 82.34 $\pm 0.28$ | 44.08 $\pm 0.40$ |
| StoCFL | | |
| $\tau = 0.1$ | 57.41 $\pm 6.69$ | 48.95 $\pm 1.95$ |
| $\tau = 0.15$ | 66.54 $\pm 1.05$ | 47.77 $\pm 0.14$ |
| HCFL$^+$ (FeSEM) | | |
| $\rho = 0.05$ | 86.90 $\pm 0.20$ | 50.34 $\pm 5.99$ |
| $\rho = 0.1$ | 85.55 $\pm 0.24$ | 49.38 $\pm 6.15$ |
| HCFL$^+$ (FedEM) | | |
| $\rho = 0.05$ | 83.88 $\pm 0.25$ | 58.92 $\pm 1.11$ |
| $\rho = 0.1$ | 83.83 $\pm 0.01$ | 60.27 $\pm 3.11$ |
| HCFL$^+$ (FedRC) | | |
| $\rho = 0.05$ | 67.72 $\pm 1.30$ | 64.13 $\pm 0.37$ |
| $\rho = 0.1$ | 67.51 $\pm 0.24$ | 63.15 $\pm 0.78$ |

