# OpenReview forum: "Find Your Optimal Assignments On-the-fly: A Holistic Framework for Clustered Federated Learning"
_ICLR.cc/2024/Conference — ICLR 2024 Conference Withdrawn Submission_

### Official Review · Reviewer_fHet · 2023-10-30

**Soundness:** 3 good
**Presentation:** 3 good
**Contribution:** 2 fair
**Rating:** 3
**Confidence:** 5

**Summary:**

This paper focuses on a comprehensive investigation into current clustered federated learning methods and proposes a four-tier framework,  to encompass and extend existing approaches. Based on this method, the authors identify the remaining challenges associated with current clustering methods in each tier and propose an enhanced clustering method to address these problems.

**Strengths:**

1. The paper is well-written and easy to follow.

2. The algorithm derivation looks correct for the proposed method.

**Weaknesses:**

1. The proposed method doesn't really have the convergence guarantee. The paper only discusses a simple case in proof.

2. The proposed method could be sensitive the initialization. The clustering and EM algorithms could easily stuck at bad local solution.

3. The experiments only compared the proposed method to other clustered federated learning approaches. But many other models have been proposed to address the heterogeneity problem.

**Questions:**

Please address the issues listed in weakness.

---

### Official Review · Reviewer_8sHW · 2023-10-30

**Soundness:** 2 fair
**Presentation:** 2 fair
**Contribution:** 2 fair
**Rating:** 3
**Confidence:** 4

**Summary:**

This paper presents a generic algorithm, termed HCFL, which is based on combining specific design choices for four tiers. The authors then discuss challenges associated with HFCL and proposes an improvement termed HCFL+.

**Strengths:**

Authors propose a generic formulation that unifies many different clustering methods. I also appreciate the discussion of specific challenges for the basic framework.

**Weaknesses:**

* The significance/novelty of the proposed 4 tier framework is unclear. The definition of the four tiers seems somewhat arbitrary. Can you show that HCFL or HCFL+ is optimal in some relevant settings. One way to verrify optimality is e.g. by comparing the estimation error with minimax bounds (see [Ref1] for an implementation of this technique for studying optimality of dictionary learning methods)

* There is little analysis of the computational and statistical properties of the proposed methods. How quickly do the iterations of Algorithm 1 converge to solutions of Eq. (1). Under shich conditions are the learnt model parameters clustered according to a ground-truth partition (see [Ref2] for an analysis of the clustering structure delivered by Total Variation minimization methods)

[Ref1] A. Jung, Y. C. Eldar and N. Görtz, "On the Minimax Risk of Dictionary Learning," in IEEE Transactions on Information Theory, vol. 62, no. 3, pp. 1501-1515, March 2016, doi: 10.1109/TIT.2016.2517006.

[Ref2] Y. SarcheshmehPour, Y. Tian, L. Zhang and A. Jung, "Clustered Federated Learning via Generalized Total Variation Minimization," in IEEE Transactions on Signal Processing, doi: 10.1109/TSP.2023.3322848.

**Questions:**

* "...we aim to introduce a more interpretable approach here.." why is this approach more interpretable ?

* Would it be possible to formulate precise probabilistic models for the observed data such that the challenges in Sec. 4.1. are present ?

* Does the optimization formulation (1) also include total variation based approaches to clustered federated learning, such as [Ref2] and

[Ref3] David Hallac, Jure Leskovec, and Stephen Boyd. 2015. Network Lasso: Clustering and Optimization in Large Graphs. In Proceedings of the 21th ACM SIGKDD International Conference on Knowledge Discovery and Data Mining (KDD '15). Association for Computing Machinery, New York, NY, USA, 387–396. https://doi.org/10.1145/2783258.2783313

---

### Official Review · Reviewer_PAju · 2023-11-05

**Soundness:** 2 fair
**Presentation:** 2 fair
**Contribution:** 2 fair
**Rating:** 5
**Confidence:** 3

**Summary:**

The paper combines and applies various existing clustering methods for federated learning under non-iid setting. The two main objectives are to determine (1) (hard and soft) cluster assignments for clients, and (2) number of clusters. The paper uses the terminology *tiers* to organize these two main objectives.

**Strengths:**

1. Explores the idea of clustering for federated learning, which is a viable approach in non-iid or concept drift data situations.
2. Provides some ablation studies.

**Weaknesses:**

1. Novelty is incremental (combines various existing clustering methods with additional parameters to optimize)
2. Not clear on how the proposed method scales.
3. Paper needs better organization. Notation not clearly explained.
4. Experiments in paper limited to vision datasets. More impact if evaluated on others such as Reddit or Stackoverflow.

**Questions:**

The paper mentions using 100 clients for experiments. However, it is not clear on what the client participation percentage rate is. How many trials were used to determine $\pm$ information? Roughly, how long (running time) does each communication round take?